# LLMPhy: Complex Physical Reasoning Using Large Language Models and World Models

## Abstract

Physical reasoning is an important skill needed for robotic agents when operating in the real world. However, solving such reasoning problems often involves hypothesizing and reflecting over complex multi-body interactions under the effect of a multitude of physical forces and thus learning all such interactions poses a significant hurdle for state-of-the-art machine learning frameworks, including large language models (LLMs). To study this problem, we propose a new physical reasoning task and a dataset, dubbed *TraySim*. Our task involves predicting the dynamics of several objects on a tray that is given an external impact – the domino effect of the ensued object interactions and their dynamics thus offering a challenging yet controlled setup, with the goal of reasoning being to infer the stability of the objects after the impact. To solve this complex physical reasoning task, we present LLMPhy, a zero-shot black-box optimization framework that leverages the physics knowledge and program synthesis abilities of LLMs, and synergizes these abilities with the world models built into modern physics engines. Specifically, LLMPhy uses an LLM to generate code to iteratively estimate the physical hyperparameters of the system (friction, damping, layout, etc.) via an implicit analysis-by-synthesis approach using a (non-differentiable) simulator in the loop and uses the inferred parameters to imagine the dynamics of the scene towards solving the reasoning task. To show the effectiveness of LLMPhy, we present experiments on our TraySim dataset to predict the steady-state poses of the objects. Our results show that the combination of the LLM and the physics engine leads to state-of-the-art zero-shot physical reasoning performance, while demonstrating superior convergence against standard black-box optimization methods and better estimation of the physical parameters. Further, we show that LLMPhy is capable of solving both continuous and discrete black-box optimization problems.

## 1 Introduction

Many recent Large Language models (LLMs) appear to demonstrate the capacity to effectively capture knowledge from vast amounts of multimodal training data and their generative capabilities allow humans to naturally interact with them towards extracting this knowledge for solving challenging real-world problems. This powerful paradigm of LLM-powered problem solving has manifested in a dramatic shift in the manner of scientific pursuit towards modeling research problems attuned to a form that can leverage this condensed knowledge of the LLMs. A few notable such efforts include, but not limited to the use of LLMs for robotic planning (Song et al., 2023; Kim et al., 2024), complex code generation (Tang et al., 2024; Jin et al., 2023), solving optimization problems (Yang et al., 2024; Hao et al., 2024), conduct sophisticated mathematical reasoning (Trinh et al., 2024), or even making scientific discoveries (Romera-Paredes et al., 2024).

While current LLMs seem to possess the knowledge of the physical world and may be able to provide a plan for solving a physical reasoning task (Singh et al., 2023; Kim et al., 2024) when crafted in a suitable multimodal format (prompt), their inability to interact with the real-world or measure unobservable attributes of the world model, hinders their capacity in solving complex physical reasoning problems (Wang et al., 2023; Bakhtin et al., 2019; Riochet et al., 2021; Harter et al., 2020; Xue et al., 2021). Consider for example the scene in Figure 1, where the LLM is provided as input the first image and is asked to answer: *which of the objects will remain standing on the tray when impacted by the pusher if the pusher collides with the tray with a velocity of 4.8 m/s?*. To answer this

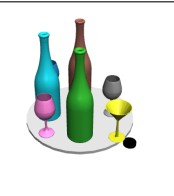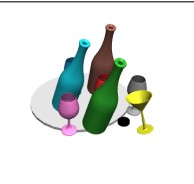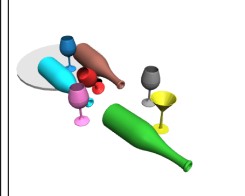

Figure 1: Frames from an example dynamical sequence in our TraySim dataset. The left-most frame shows the first frame of the scene with many objects on the tray and is going to be impacted by a black pusher (right-bottom). The subsequent frames show the state of the system at the 25-th, 50-th, and the 200-th time step (each step is 0.01s). Our task is for the LLM to reason through the dynamics of the system and predict the stability of each object on the tray at the end of the episode, in a zero-shot manner.

question, the LLM must know the various physical attributes of the system, including the masses, friction coefficients, and forces, among others. While, a sophisticated LLM may be able to give an educated guess based on the intuitive physics of the system extracted from its training data, a useful solution would demand a more intricate reasoning path in estimating the real-world physics and dynamics of the given system; such complex dynamics may be difficult or even impossible to be learned solely from training data. Conversely, advancements in graphics hardware and software have led to the development of advanced physics engines capable of simulating realistic world models. Thus, rather than having the LLM to learn the world physics, our key idea is to consider using a physics engine in tandem with the LLM, where the LLM may use its world knowledge for generating scene-based reasoning hypotheses while the simulator is used to verify them within the physical world model.

To study this problem, we consider the novel task of predicting the dynamics of objects and their stability under the influence of an impact – an important problem for a variety of robotic applications (Gasparetto et al., 2015; Ahmed et al., 2020). In this paper, we consider this problem in a challenging setting using our new dataset, *TraySim*, in which the impact is caused by a pusher colliding to a tray that holds several objects of varied sizes, masses, and centers of gravity, with the goal of predicting the dynamics of each of the object instances. We cast this task as that of answering physical reasoning questions. Specifically, as illustrated in Figure 1, TraySim includes simulated video sequences consisting of a tray with an arbitrary number of objects on it and given the first video frame of a given scene, the task of the reasoning model is to infer which of the objects on the tray will remain upright after the impact when the system has stabilized. As is clear from Figure 1, solving this task will require the model to derive details regarding the physical properties of each of the objects and their contacts, as well as have the ability to imagine the system's dynamics through multi-body interactions influenced by the various internal and external forces from the impact. Our task presents a challenging reasoning setup for current machine learning models, including LLMs.

To solve this task, we propose LLMPhy, a black-box optimization setup combining an LLM with a physics engine that leverages the program synthesis abilities of the LLM to communicate with the engine for solving our task. LLMPhy operates in two phases: i) a parameter estimation phase, where LLMPhy is used as a continuous black-box optimization module towards inferring the physical parameters of the objects, including the friction, stiffness, damping, etc. from a given example video sequence, and ii) a scene understanding phase, where the LLM-simulator combination is used as a discrete black-box optimizer to reconstruct the problem layout for synthesizing the setup within the simulator for execution. Our framework builds a feedback loop between the LLM and the physics engine, where the LLM generates programs using its estimates of physical attributes; the programs are executed in the simulator, and the error from the simulations are fed back to the LLM as prompts to refine its estimates until a suitable convergence criteria is met. Note that we do not assume any differentiability properties of the simulator, which makes our setup highly general. This allows the approach to function as a black-box optimization framework, enabling its use with a wide range of simulators without the need for gradient-based methods.

While we may generate unlimited data using our simulation program, given the zero-shot nature of our setup, we synthesized 100 sequences in our *TraySim* dataset to demonstrate the effectiveness of

LLMPhy. Each sample in TraySim has two video sequences: i) the task sequence of which only the first frame is given to a reasoning agent, and ii) a parameter-estimation video sequence which has a lesser number of instances of each of the object types appearing in the task sequence; the latter sequence has an entirely different layout and dynamics of objects after its specific impact settings. To objectively evaluate performance, we cast the task as physical question answering problem, where the LLM is required to select the correct subset of answers from the given candidate answers. Our results on TraySim show that LLMPhy leads to clear improvements in performance ($\sim$ 3% accuracy) against alternatives on the QA task, including using Bayesian optimization, CMA-ES, and solely using an LLM for physical reasoning, while demonstrating better convergence and estimation of the physical parameters.

Before moving forward, we summarize below our main contributions:

- We consider the novel task of reasoning over complex physics of a highly dynamical system by combining LLMs with possibly non-differentiable physics engines.
- We propose a zero-shot reasoning framework LLMPhy, which combines the reasoning and program synthesis abilities of an LLM with the realistic simulation abilities of a physics engine. This approach is used to estimate the physical parameters of the model, the scene layout, and synthesizing the dynamical scene for inferring the solution.
- We introduce a novel synthetic multi-view dataset: TraySim, to study this task. The dataset consists of 100 scenes for zero-shot evaluation.
- Our experiments demonstrate state-of-the-art performances using LLMPhy highlighting its potential for tackling complex physics-based tasks involving both discrete and continuous optimization sub-tasks.

## 2 RELATED WORKS

Large language models (LLMs) demonstrate remarkable reasoning skills across a variety of domains, highlighting their versatility and adaptability. They have shown proficiency in managing complex conversations (Glaese et al., 2022; Thoppilan et al., 2022), engaging in methodical reasoning processes (Wei et al., 2022; Kojima et al., 2022), planning (Huang et al., 2022), tackling mathematical challenges (Lewkowycz et al., 2022; Polu et al., 2022), and even generating code to solve problems (Chen et al., 2021). As we start to incorporate LLMs into physically embodied systems, it's crucial to thoroughly assess their ability for physical reasoning. However, there has been limited investigation into the physical reasoning capabilities of LLMs.

In the field of language-based physical reasoning, previous research has mainly concentrated on grasping physical concepts and the attributes of different objects. (Zellers et al., 2018) introduced grounded commonsense inference, merging natural language inference with commonsense reasoning. Meanwhile, (Bisk et al., 2020) developed the task of physical commonsense reasoning and a corresponding benchmark dataset, discovering that pretrained models often lack an understanding of fundamental physical properties. (Aroca-Ouellette et al., 2021) introduced a probing dataset that evaluates physical reasoning through multiple-choice questions. This dataset tests both causal and masked language models in a zero-shot context. However, many leading pretrained models struggle with reasoning about physical interactions, particularly when answer choices are reordered or questions are rephrased. (Tian et al., 2023) explored creative problem-solving capabilities of modern LLMs in constrained setting. They automatically a generate dataset consisting of real-world problems deliberately designed to trigger innovative usage of objects and necessitate out-of-the-box thinking. (Wang et al., 2023) presented a benchmark designed to assess the physics reasoning skills of large language models (LLMs). It features a range of object-attribute pairs and questions aimed at evaluating the physical reasoning capabilities of various mainstream language models across foundational, explicit, and implicit reasoning tasks. The results indicate that while models like GPT-4 demonstrate strong reasoning abilities in scenario-based tasks, they are less consistent in object-attribute reasoning compared to human performance.

In addition to harnessing LLMs for physical reasoning, recent works have used LLMs for optimization. The main focus has been on targeted optimization for employing LLMs to produce prompts that improves performance of another LLM. (Yang et al., 2024) shows that LLMs are able to find good-quality solutions simply through prompting on small-scale optimization problems. They demon-

strate the ability of LLMs to optimize prompts where the goal is to find a prompt that maximizes the task accuracy. The applicability of various optimization methods depends on whether the directional feedback information is available. In cases when the directional feedback is available, one can choose efficient gradient-based optimization methods (Sun et al., 2019). However, in scenarios without directional feedback, black-box optimization methods (Terayama et al., 2021) are useful such as Bayesian optimization (Mockus, 1974), Multi-Objective BO (Konakovic Lukovic et al., 2020) and CMA-ES (Hansen & Ostermeier, 2001). Only a limited number of studies have explored the potential of LLMs for general optimization problems. (Guo et al., 2023) shows that LLMs gradually produce new solutions for optimizing an objective function, with their pretrained knowledge significantly influencing their optimization abilities. (Nie et al., 2024) study factors that make an optimization process challenging in navigating a complex loss function. They conclude that LLM-based optimizer's performance varies with the type of information the feedback carries, and given proper feedback, LLMs can strategically improve over past outputs. In contrast to these prior works, our goal in this work is to combine an LLM with a physics engine for physics based optimization.

Our work is inspired by the early work in neural de-rendering (Wu et al., 2017) that either (re-) simulates a scene using a physics engine or synthesizes realistic scenes for physical understanding Bear et al. (2021). Similar to our problem setup, CoPhy Baradel et al. (2019) and ComPhy Chen et al. (2022) consider related physical reasoning tasks, however with simplistic physics and using supervised learning. In (Liu et al., 2022), a language model is used to transform a given reasoning question into a program for a simulator, however does not use the LLM-simulator optimization loop as in LLMPhy. In SimLM (Memery et al., 2023), an LLM-simulator combination is presented for predicting the physical parameters of a projectile motion where the feedback from a simulator is used to improve the physics estimation in an LLM, however assumes access to in-context examples from previous successful runs for LLM guidance. In Eureka Ma et al. (2023), an LLM-based program synthesis is presented for designing reward functions in a reinforcement learning (RL) setting, where each iteration of their evolutionary search procedure produces a set of LLM generated candidate reward functions. Apart from the task setup, LLMPhy differs from Eureka in two aspects: (i) Eureka involves additional RL training that may bring in training noise in fitness evaluation, (ii) does not use full trajectory of optimization in its feedback and as a result, the LLM may reconsider previous choices. See F for details.

## 3 PROPOSED METHOD

The purpose of this work is to enable LLMs to perform physics-based reasoning in a zero-shot manner. Although LLMs may possess knowledge of physical principles that are learned from their training data, state-of-the-art models struggle to effectively apply this knowledge when solving specific problems. This limitation, we believe, is due to the inability of the model to interact with the scene to estimate its physical parameters, which are essential and needs to be used in the physics models for reasoning, apart from the stochastic attributes implicit in any such system. While, an LLM may be trained to implicitly model the physics given a visual scene – e.g., generative models such as SoRA[1], Emu-video Girdhar et al. (2023), etc., may be considered as world model simulators – training such models for given scenes may demand exorbitant training data and compute cycles. Instead, in this paper, we seek an alternative approach by leveraging the recent advancements in realistic physics simulation engines and use such simulators as a tool accessible to the LLM for solving its given physical reasoning task. Specifically, we attempt to solve the reasoning task as that of equipping the LLM to model and solve the problem using the simulator, and for which we leverage on the LLM's code generation ability as a bridge. In the following sections, we exposit the technical details involved in achieving this LLM-physics engine synergy.

### 3.1 PROBLEM SETUP

Suppose $\mathbf{X}^v = \langle \mathbf{x}_1^v, \mathbf{x}_2^v, \cdots, \mathbf{x}_T^v \rangle$ denote a video sequence with $T$ frames capturing the dynamics of a system from a camera viewpoint $v$. We will omit the superscript $v$ when referring to all the views jointly. In our setup, we assume the scene consists of a circular disk (let us call it a *tray*) of a given radius, friction, and mass. Further, let $\mathcal{C}$ denote a set of object types, e.g., in Figure 1, there are three types of objects: a *bottle*, a *martini glass*, and a *wine glass*. The tray is assumed to hold

---

[1]https://openai.com/index/sora/

Figure 2: Illustration of the key components of LLMPhy and the control flow between LLM, physics simulator, and the varied input modalities and examples.

a maximum of $K$ object instances, the $k$-th instance is denoted $o_k$; $K$ being a perfect square. To simplify our setup, we assume that the instances on the tray are arranged on a $\sqrt{K} \times \sqrt{K}$ regular grid, with potentially empty locations. We further assume that the masses of the objects in $\mathcal{C}$ are given during inference, while other physical attributes, denoted as $\Phi_c$ for all objects $c \in \mathcal{C}$, are *unknown* and identical for objects of the same type. In line with the standard Mass-Spring-Damping (MSD) dynamical system, we consider the following set of contact physics parameters $\Phi_c \in \mathbb{R}^4$ for each object class: i) coefficient of sliding friction, ii) stiffness, iii) damping, and iv) the rotational inertia (also called armature). To be clear, we do not assume or use any physics model in our optimization pipeline, and our setup is entirely black-box, but the selection of these optimization physics parameters is inspired by the MSD model. We assume the objects do not have any rotational or spinning friction. While the instances $o_k$ of the same type are assumed to share the same physics parameters, they differ in their visual attributes such as color or shape. The tray is impacted by a pusher $p$ that starts at a fixed location and is given an initial velocity of $p_s$ towards the tray. The pusher is assumed to have a fixed mass and known physical attributes, and the direction of impact is assumed to coincide with the center of the circular tray.

## 3.2 PROBLEM FORMULATION

With the notation above, we are now ready to formally state our problem. In our setup, we define an input task instance as: $\mathcal{T} = (\{\mathbf{x}_g^v\}_{v \in |\mathcal{V}|}, p_s, Q, \mathcal{O}, \mathcal{I}, \mathbf{X}_\mathcal{T}, \mathcal{C}_\mathcal{T})$, where $\mathbf{x}_g$ is the first frame of a video sequence $\mathbf{X}$ with $\mathcal{V}$ views, $p_s$ is the initial velocity of the pusher $p$, $Q$ is a question text describing the task, and $\mathcal{O}$ is a set of answer candidates for the question. The goal of our reasoning agent is to select the correct answer set $\mathcal{A} \subset \mathcal{O}$. The notation $\mathcal{C}_\mathcal{T} \subseteq \mathcal{C}$ denotes the subset of object classes that are used in the given task example $\mathcal{T}$. In this paper, we assume the question is the same for all task examples, i.e., *which of the object instances on the tray will remain steady when impacted by the pusher with a velocity of $p_s$?* We also assume to have been given a few in-context examples $\mathcal{I}$ that familiarizes the LLM on the structure of the programs it should generate. We found that such examples embedded in the prompt are essential for the LLM to restrict its generative skills to the problem at hand, while we emphasize that the knowledge of these in-context examples will not by themselves help the LLM to correctly solve a given test example.

As it is physically unrealistic to solve the above setup using only a single image (or multiple views of the same time-step), especially when different task examples have distinct dynamical physics parameters $\Phi$ for $\mathcal{C}_\mathcal{T}$, we also assume to have access to an additional video sequence $\mathbf{X}_\mathcal{T}$ associated with the given task example $\mathcal{T}$ containing the same set of objects as in $\mathbf{x}_g$ but in a different layout and potentially containing a smaller number of object instances. The purpose of having $\mathbf{X}_\mathcal{T}$ is to estimate the physics parameters of the objects in $\mathbf{x}_g$, so that these parameters can then be used to conduct physical reasoning for solving $\mathcal{T}$, similar to the setup in Baradel et al. (2019); Chen et al. (2022). Note that this setup closely mirrors how humans would solve such a reasoning task. Indeed, humans may pick up and interact with some object instances in the scene to understand their physical properties, before applying sophisticated reasoning on a complex setup. Without any loss

of generality, we assume the pusher velocity in $\mathbf{X}_{\mathcal{T}}$ is fixed across all such auxiliary sequences and is different from $p_s$, which varies across examples.

## 3.3 COMBINING LLMS AND PHYSICS ENGINES FOR PHYSICAL REASONING

In this section, our proposed LLMPhy method for our solving physical reasoning task is outlined. Figure 2 illustrates our setup. Since LLMs on their own may be incapable of performing physical reasoning over a given task example, we propose combining the LLM with a physics engine. The physics engine provides the constraints of the world model and evaluates the feasibility of the reasoning hypothesis generated by the LLMs. This setup provides feedback to the LLM that enables it reflect on and improve its reasoning. Effectively solving our proposed task demands inferring two key entities: i) the physical parameters of the setup, and ii) layout of the task scene for simulation using physics to solve the task. We solve for each of these sub-tasks in two distinct phases as detailed below. Figure 3 illustrates our detailed architecture, depicting the two phases and their interactions.

### 3.3.1 LLMPhy PHASE 1: INFERRING PHYSICAL PARAMETERS

As described above, given the task example $\mathcal{T}$, LLMPhy uses the task video $\mathbf{X}_{\mathcal{T}}$ to infer the physical attributes $\Phi$ of the object classes in $\mathcal{C}$. Note that these physical attributes are specific to each task example. Suppose $\tau : \mathcal{X} \to \mathbb{R}^{3 \times T \times |\mathcal{C}_{\mathcal{T}}|}$ be a function that extracts the physical trajectories of each of the objects in the given video $\mathbf{X}_{\mathcal{T}} \in \mathcal{X}$, where $\mathcal{X}$ denotes the set of all videos.[2] Note that we have used a subscript of $\mathcal{T}$ with $\mathcal{C}$ to explicitly show the subset of object types that may be appearing in the given task example.

Suppose $\text{LLM}_1$ denotes the LLM used in phase 1[3], which takes as input the in-context examples $\mathcal{I}_1 \subset \mathcal{I}$ and the object trajectories from $\mathbf{X}_{\mathcal{T}}$, and is tasked to produce a program $\pi(\Phi) \in \Pi$, where $\Pi$ denotes the set of all programs. Further, let $\text{SIM} : \Pi \to \mathbb{R}^{3 \times T \times \mathcal{C}_{\mathcal{T}}}$ be a physics-based simulator that takes as input a program $\pi(\Phi) \in \Pi$ and produce trajectories of objects described by the program using the physics attributes. Then, the objective of phase 1 of $\text{LLMPhy}_1$ can be described as:

$$\arg\min_{\Phi} \|\text{LLMPhy}_1(\pi(\Phi) \mid \tau(\mathbf{X}_{\mathcal{T}}), \mathcal{I}_1) - \tau(\mathbf{X}_{\mathcal{T}})\|^2, \tag{1}$$

where $\text{LLMPhy}_1 = \text{SIM} \circ \text{LLM}_1$ is the composition of the simulator and the LLM through the generated program, with the goal of estimating the correct physical attributes of the system $\Phi$. Note that the notation $\pi(\Phi)$ means the generated program takes as argument the physics parameters $\Phi$ which is what we desire to optimize using the LLM.

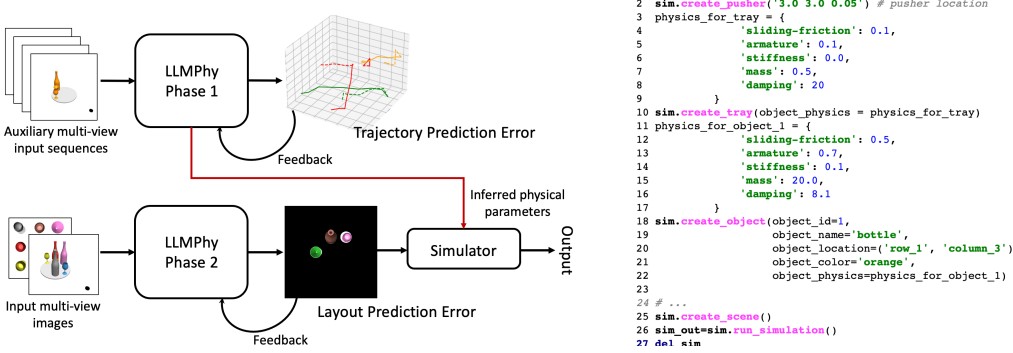

Figure 3: Left: Full architecture of the two phases in LLMPhy. Right: A simplified LLMPhy program. We abstract the complexity in running the simulations through simple API calls so that LLM can focus on the optimization variables. See Appendix I for full program examples.

---

[2]In experiments, we use the simulator to extract object trajectories, thus implementing $\tau$. See Appendix D.1.
[3]We use the same LLM in both phases, but the notation is only for mathematical precision.

### 3.3.2 LLMPhy Phase 2: Simulating Task Example

The second phase of LLMPhy involves applying the inferred physical parameters $\Phi$ for the object classes in $\mathcal{C}$ to solve the task problem described in $\mathbf{x}_g$, i.e., the original multi-view task images (see Figure 3). This involves solving a perception task consisting of two steps: i) understanding the scene layout (i.e., where the various object instances are located on the tray, their classes, and attributes (e.g., color); this is important as we assume that different type of objects have distinct physical attributes, and ii) using the physical attributes and the object layout to produce code that can be executed in the physics engine to simulate the full dynamics of the system to infer the outcome; i.e., our idea is to use the simulator to synthesize a dynamical task video from the given input task images, and use the ending frames of this synthesized video to infer the outcome (see Figure 2).

Suppose $\text{LLM}_2$ denotes the LLM used in Phase 2, which takes as input the multi-view task images $\mathbf{x}_g$, the physical attributes $\Phi^*$, and Phase 2 in-context examples $\mathcal{I}_2 \subset \mathcal{I}$ to produce a program $\pi(\Psi) \in \Pi$ that reproduces the scene layout parameters, i.e., the triplet $\Psi = \{(\text{class}, \text{location}, \text{color})\}_k$ for each instance. The objective for estimating the layout parameters $\Psi$ can be written as:

$$\Psi^* = \arg\min_{\Psi} \|\text{LLMPhy}_2(\pi(\Psi) \mid \mathbf{x}_g, \mathcal{I}_2) - \mathbf{x}_g\|^2, \tag{2}$$

where $\text{LLMPhy}_2 = \text{SIM} \circ \text{LLM}_2$. Once the correct layout parameters $\Psi^*$ are estimated, we can produce a video sequence $\hat{\mathbf{X}} \mid \Psi^*, \Phi^*$ using the simulator, and which can then be used for solving the problem by selecting an answer subset $\mathcal{A}$ from the answer options $\mathcal{O}$. We may use an LLM or extract the pose of the instances within the simulator to solve the question-answering task; in this work, we use the latter for convenience.

### 3.4 Optimizing LLM-Simulator Combination

In Alg. 1, we detail the steps for optimizing LLMPhy. Given that we assume the simulator might be non-differentiable, we frame this as a black-box optimization problem. Here, the optimization variables are sampled based on the inductive bias and the knowledge of physics learned by the LLM from its large corpora of training data. The LLM generates samples over multiple trials, which are then validated using the simulator. The resulting error is used to refine the LLM's hyper-parameter search. A key insight of our approach is that, since the hyper-parameters in our setup have physical interpretations in the real-world, a knowledgeable LLM should be capable of selecting them appropriately by considering the error produced by its previous choices. In order for the LLM to know the history of its previous choices and the corresponding error induced, we augment the LLM prompt with this optimization trace from the simulator at each step.

---

**Algorithm 1** Pseudo-code describing the key steps in optimizing LLMPhy for phases 1 and 2.

**Require:** $\mathbf{X}, \Lambda$       $\triangleright$ $\mathbf{X}$ is the input data, and $\Lambda$ is the desired result, e.g., trajectory, layout, etc.
    prompt $\leftarrow$ **'task prompt'**          $\triangleright$ We assume here a suitable prompt for the LLM.
    **for** $i = 1$ to max_steps **do**
       $\pi \leftarrow \text{LLM}(\mathbf{X}, \mathcal{I}, \text{prompt})$
                   $\triangleright$ Generated program $\pi$ is assumed to have the optimization variables.
       $\hat{\Lambda} \leftarrow \text{SIM}(\pi)$                     $\triangleright$ SIM reproduced result from $\pi$.
       error $\leftarrow \|\Lambda - \hat{\Lambda}\|^2$
       **if** error $\leq \epsilon$ **then**
          **return** $\pi$
       **else**
          prompt $\leftarrow \text{concat}(\text{prompt}, \pi, \text{concat}(\textbf{“Error =”}, \text{error})$
       **end if**
    **end for**

---

## 4 Experiments and Results

In this section, we detail our simulation setup used to build our TraySim dataset, followed by details of other parts of our framework, before presenting our results.

**Simulation Setup:** As described above, we determine the physical characteristics of our simulation using a physics engine. MuJoCo Todorov et al. (2012) was used to setup the simulation and compute the rigid body interactions within the scene. It is important to note that any physics engine capable of computing the forward dynamics of a multi-body system can be integrated within our framework as the simulation is exposed to the LLM through Python API calls for which the physical parameters and layout are arguments. As a result, the entirety of the simulator details are abstracted out from the LLM. Our simulation environment is build upon a template of the *World*, which contains the initial parametrization of our model of Newtonian physics. This includes the gravity vector **g**, time step, and contact formulation, but also graphical and rendering parameters later invoked by the LLM when executing the synthesized program. See Appendix A for details.

**TraySim Dataset:** Using the above setup, we created 100 task sequences using object classes $\mathcal{C} = \{\text{wine glass}, \text{martini glass}, \text{bottle}\}$ with object instances from these classes arranged roughly in a $3 \times 3$ matrix on the tray. The instance classes and the number of instances are randomly chosen with a minimum of 5 and a maximum of 9. Each task sequence is associated with an auxiliary sequence for parameter estimation that contains at least one object instance from every class of object appearing in the task images. We assume each instance is defined by a triplet: (color, type, location), where the color is unique across all the instances on the tray so that it can be identified across the multi-view images. The physical parameters of the objects are assumed to be the same for both the task sequences and the auxiliary sequences, and instances of the same object classes have the same physical parameters. The physics parameters were randomly sampled for each problem in the dataset. Each sequence was rendered using the simulator for 200 time steps, each step has a duration of 0.01s. We used the last video frame from the task sequence to check the stability of each instance using the simulator. We randomly select five object instances and create a multiple choice candidate answer set for the question-answering task, where the ground truth answer is the subset of the candidates that are deemed upright in the last frame. In Figure 4, we illustrate the experimental setup using an example from the TraySim dataset. See Appendix B for more details of the physics parameters, and other settings.

**Large Language Model:** We use the OpenAI o1-mini text-based LLM for our Phase 1 experiments and GPT-4o vision-and-language model (VLM) in Phase 2. Recall that in Phase 1 we pre-extract the object trajectories for optimization.

**Phase 1 Details:** In this phase, we provide as input to the LLM four items: i) a prompt describing the problem setup, the qualitative parameters of the objects (such as mass, height, size of tray, etc.) and the task description, ii) an in-context example consisting of sample trajectories of the object instances from its example auxiliary sequence, iii) a program example that, for the given example auxiliary sequence trajectories, shows their physical parameters and the output structure, and iv) auxiliary task sequence trajectories (from the sequence for which the physical parameters have to be estimated) and a prompt describing what the LLM should do. The in-context example is meant to guide the LLM to understand the setup we have, the program structure we expect the LLM to synthesize, and our specific APIs that need to be called from the synthesized program to reconstruct the scene in our simulator. Please see our Appendices D and I for details.

**Phase 2 Details:** The goal of the LLM in Phase 2 is to predict the object instance triplet from the multi-view task images. Towards this end, the LLM generates code that incorporates these triplets, so that when this code is executed, the simulator will reproduce the scene layout. Similar to Phase 1, we provide to the LLM an in-context example for guiding its code generation, where this in-context example contains multi-view images and the respective program, with the goal that the LLM learns the relation between parts of the code and the respective multi-view images, and use this knowledge to write code to synthesize the layout of the provided task images. When iterating over the optimization steps, we compute an error feedback to the LLM to improve its previously generated code. See Appendix D and I for precise details on the feedback.

**LLMPhy Feedback Settings:** We compute the trajectory reconstruction error in Phase 1 where the synthesized program from the LLM containing the estimated physics parameters is executed in the simulator to produce the motion trajectory of the center of gravity of the instances. We sample the trajectory for every 10 steps and compute the L2 norm between the input and reconstructed trajectories. We use a maximum of 30 LLM-simulator iterations in Phase 1 and use the best reconstruction error to extract the parameters. For Phase 2, we use the Peak Signal-to-Noise ratio (PSNR) in the reconstruction of the first frame by the simulator using the instance triplets predicted by the LLM in

| Expt # | Phase 1 | Phase 2 | mIoU (%) |
|--------|---------|---------|----------|
| 1 | Random | Random | 19.0 |
| 2 | N/A | LLM | 32.1 |
| 3 | Random | LLMPhy | 50.8 |
| 4 | BO | LLMPhy | 59.6 |
| 5 | CMA-ES | LLMPhy | 59.7 |
| 6 | LLMPhy | LLMPhy | **62.0** |
| 7 | GT | LLMPhy | 65.1 |
| 8 | CMA-ES | GT | 75.8 |
| 9 | LLMPhy | GT | 77.5 |

Table 1: Performances on TraySim QA task.

|  | LLMPhy | LLMPhy (1 iter.) |
|--|--------|------------------|
| C+L (%) | 68.7 | 50.0 |
| L+T (%) | 66.3 | 49.3 |
| C+L+T (%) | 56.0 | 36.8 |

Table 2: Experiments presenting the accuracy of generated code compared to the ground truth in Phase 2 of LLMPhy. We report the accuracy of matching the color (C) of the objects, their locations (L) on the $3 \times 3$ grid, and their type (T).

the generated program. We used a maximum of 5 LLMPhy iterations for this phase. As the LLM queries are expensive, we stopped the iterations when the trajectory prediction error is below 0.1 on average for Phase 1 and when the PSNR is more than 45 dB for Phase 2.

**Evaluation Metric and Baselines:** We consider various types of evaluations in our setup. Specifically, we use the intersection-over-union as our key performance metric that computes the overlap between the sets of LLMPhy produced answers in Phase 2 with the ground truth answer set. We also report the performances for correctly localizing the instances on the tray, which is essential for simulating the correct scene. As ours is a new task and there are no previous approaches that use the composition of LLM and physics engine, we compare our method to approaches that are standard benchmarks for continuous black-box optimization, namely using Bayesian optimization Mockus (1974) and Covariance matrix adaptation evolution strategy (CMA-ES) Hansen & Ostermeier (2001); Hansen (2016).

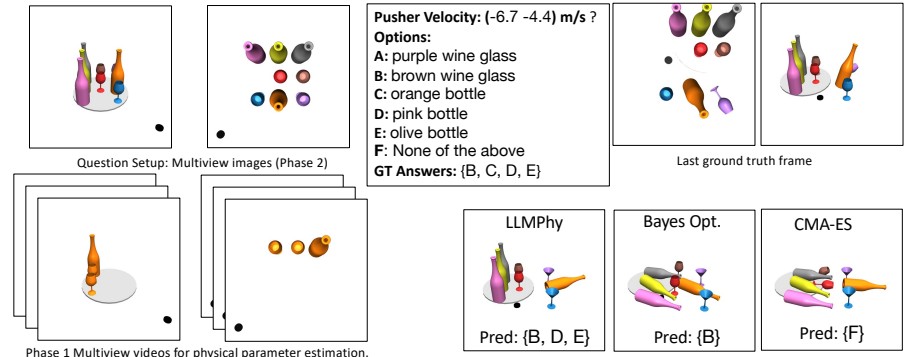

Figure 4: A sample qualitative result using LLMPhy, BO, and CMA-ES illustrating our problem setup. We omit the task question, which is the same for all problems, except the pusher velocity.

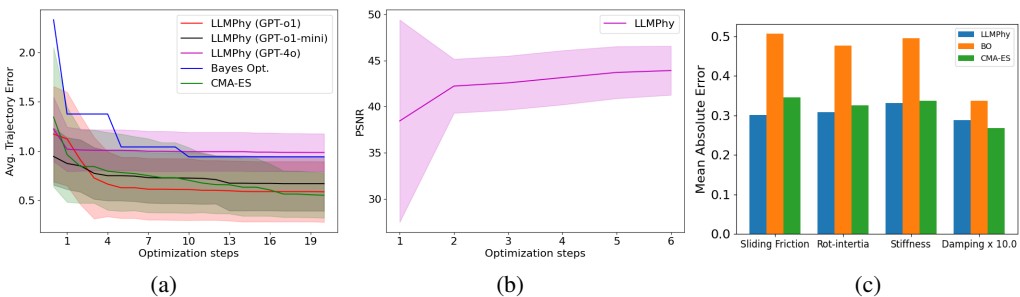

(a)         (b)         (c)

Figure 5: (a) Convergence comparisons using state-of-the-art LLMs in LLMPhy against Bayesian optimization and CMA-ES. We plot the *minimum loss computed thus far* in the optimization against the number of optimization steps. (b) shows the convergence of LLMPhy in Phase 2. (c) Comparison of physical parameter estimation error against alternatives using the ground truth.

**Comparisons to Prior Methods:** In Table 1, we compare the performance of Phase 1 and Phase 2 of LLMPhy to various alternatives and prior black-box optimization methods. Specifically, we see that random parameter sampling (Expt. #1) for the two phases lead to only 20% accuracy. Next, in Expt. #2, we use the Phase 2 multiview images (no sequence) and directly ask the GPT-4o to predict the outcome of the interaction (using the ground truth physics parameters provided), this leads to 32% accuracy, suggesting the LLM may provide an educated guess based on the provided task images. In Expt. #3, we use LLMPhy for Phase 2, however use random sampling for the physics parameters. We see that this leads to some improvement in performance, given we are using the simulator to synthesize the dynamical scene. Although the performance is lower than ideal and as noted from Figure 6 in the Appendix, we see that the outcome is strongly dependent on the physics parameters. In Expt. #4 and #5, we compare to prior black-box optimization methods for estimating the physics parameters while keeping the Phase 2 inference from LLMPhy as in the Expt. #3. To be comparable, we used 30 iterations for all methods.[4] As can be noted from the table, LLMPhy leads to about 2.3% better QA accuracy as is seen in Expt. #6. In Expt #7, we used the ground truth (GT) physics attributes for the respective objects in the simulation, and found 65.1% accuracy, which forms an upper-bound on the accuracy achievable from Phase 1. In Expt. #8 and #9, we compare the performance using GT phase 2 layout. We find from the performances that the physics parameters produced by LLMPhy are better than CMA-ES. In Table 2, we present the accuracy of LLMPhy in localizing the triplets correctly in Phase 2. We find that with nearly 56% accuracy, LLMPhy estimates all the triplets and the performance improves over LLMPhy iterations. See detailed experiments and ablation studies in Appendix E.

**Convergence and Correctness of Physical Parameters:** In Figure 10(a), we plot the mean convergence (over a subset of the dataset) when using GPT-4o, o1-mini, Bayesian Optimization, and CMA-ES. We also include results using the more recent, powerful, expensive, and text-only OpenAI o1-preview model on a subset of 10 examples from TraySim; these experiments used a maximum of 20 optimization iterations. The convergence trajectories show that o1-mini and o1-preview perform significantly better than GPT-4o in Phase 1 optimization. We see that LLMs initial convergence is fast, however with longer iterations CMA-ES appears to outperform in minimizing the trajectory error. However, Table 1 shows better results for LLMPhy. To gain insights into this discrepancy, in Figure 5(c), we plot the mean absolute error between the predicted physics parameters and their ground truth from the comparative methods. Interestingly, we see that LLMPhy estimations are better; perhaps because prior methods optimize variables without any semantics associated to them, while LLMPhy is optimizes "physics" variables, leading to the better performance and faster convergence. In Figure 5(b), we plot the convergence of LLMPhy Phase 2 iterations improving the PSNR between the synthesized (using the program) and the provide task images. As is clear, the correctness of the program improves over iterations. Both BO and CMA-ES are continuous methods and cannot optimize over the discrete space in Phase 2. However, LLMPhy is capable of optimizing in both continuous and discrete optimization spaces. We ought to emphasize this **important benefit**.

## 5 CONCLUSIONS AND LIMITATIONS

In this paper, we introduced the novel task of predicting the outcome of complex physical interactions, solving for which we presented LLMPhy, a novel setup combining an LLM with a physics engine. Our model systematically synergizes the capabilities of each underlying component, towards estimating the physics of the scene and experiments on our proposed TraySim dataset demonstrate LLMPhy's superior performance. Notably, as we make no assumptions on the differentiability of the simulator, our framework could be considered as an LLM-based black-box optimization framework, leveraging LLMs' knowledge for hyperparameter sampling. Our study shows that the recent powerful LLMs have enough world "knowledge" that combining this knowledge with a world model captured using a physics engine allows interactive and iterative problem solving for better reasoning.

While our problem setup is very general, we note that we only experiment with four physical attributes (albeit unique per each object class). While, this may not be limiting from a feasibility study of our general approach, a real-world setup may have other physics attributes as well that needs to be catered to. Further, we consider closed-source LVLMs due to their excellent program synthesis benefits. Our key intention is to show the usefulness of an LLM for solving our task and we hope future open-source LLMs would also demonstrate such beneficial capabilities.

---

[4]For LLMPhy, we are limited by the context window of the LLM and the cost.

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

# Appendices

TABLE OF CONTENTS

## A  SIMULATION SETUP

As discussed in the previous section, we are determining the physical characteristics of our simulation using a physics engine. MuJoCo  Todorov et al. (2012) was used to setup the simulation and compute the rigid body interactions within the scene. It is important to note that any physics engine capable of computing the forward dynamics of a multi-body system can be integrated within our framework. This is because LLMPhy implicitly estimates the outcome of a scene based on the specific physical laws the engine is computing. To be clear, LLMPhy does not assume any physical model of the world and operates entirely as a black-box optimizer. The world model is entirely captured by the physics engine that executes the program LLMPhy produces.

The simulation environment is build upon a template of the *World*, $\mathcal{W}$, which contains the initial parametrization of our model of Newtonian physics. This includes the gravity vector **g**, time step, and contact formulation, but also graphical and rendering parameters later invoked by the LLM when executing the synthesized program. MuJoCo uses internally a soft contact model to compute for instance complementarity constraints; in our implementation we use a non-linear sigmoid function that allows a very small inter-body penetration and increases the simulation stability during abrupt accelerations. We use elliptic friction cones to replicate natural contacts more closely. We further take advantage of the model architecture of MuJoCo by programmatically inserting arbitrary objects $o_k$ from the classes in $\mathcal{C}$ into the scene, as described in Section 3.1. For each parametric object class in $\mathcal{C}$, we generate an arbitrary appearance and physical attributes such as static friction, stiffness, damping, and armature. An arbitrary number of object instances are created from each class (up to a provided limit on their total number) and placed at randomly chosen positions on a regular grid (scene layout). The graphical renderer is used to record the frame sequences **X** corresponding to five orthogonally placed cameras around the *World* origin, including a top-down camera. In addition, we support panoptic segmentation of all objects in the scene and store the corresponding masks for arbitrarily chosen key frames. The simulated data also contains privileged information such as the pusher-tray contact information (*i.e.* force, location, velocity, time stamp), and the stability information for each object, $\mathcal{S}_k = \{1|\arccos(\mathbf{g}, O\mathbf{z}_k) < \alpha, 0|otherwise\}$, where **g** is the gravity vector, $O\mathbf{z}_k$ is the upright direction of object $k$ and $\alpha$ is an arbitrarily chosen allowable tilt. Thus, in our experiments, we use $\alpha = 45°$. Given that we consider only rigid objects with uniformly distributed mass, we assume that this a reasonable and conservative threshold.

Other than the physics parametrization of each object class $\mathcal{C}$ and the scene layout $\cup o_k$, the outcome of the simulation for sequence $\mathbf{X}$ is given by the initial conditions of the pusher object $p$, namely its initial velocity $\dot{\mathbf{p}}_s$ and position $\mathbf{p}_s$. The usual torque representation is used:

$$\boldsymbol{\tau} = \mathbf{I}_C \dot{\boldsymbol{\omega}} + \boldsymbol{\omega} \times \mathbf{I}_C \boldsymbol{\omega}, \tag{3}$$

which relates the angular acceleration $\alpha$ and angular velocity $\dot{\omega}$ to the objects torque $\tau$. The simulator computes in the end the motion of each object based on the contact dynamics model given by:

$$\mathbf{M}(\mathbf{q})\ddot{\mathbf{q}} + \mathbf{C}(\mathbf{q}, \dot{\mathbf{q}}) = \mathbf{S}_a^T \boldsymbol{\tau} + \mathbf{S}_u^T \boldsymbol{\lambda}_u + \mathbf{J}_c^T(\mathbf{q})\boldsymbol{\lambda}_c, \tag{4}$$

where $\mathbf{M}(\mathbf{q}) \in \mathbb{R}^{(n_a+n_u)\times(n_a+n_u)}$ is the mass matrix; $\mathbf{q} \triangleq [\mathbf{q}_a^T, \mathbf{q}_u^T]^T \in \mathbb{R}^{n_a+n_u}$ are generalized coordinates; and $\mathbf{C}(\mathbf{q}, \dot{\mathbf{q}}) \in \mathbb{R}^{n_a+n_u}$ represents the gravitational, centrifugal, and the Coriolis term. The selector matrices $\mathbf{S}_a = [\mathbb{I}_{n_a \times n_a} \ \mathbf{0}_{n_a \times n_u}]$ and $\mathbf{S}_u = [\mathbf{0}_{n_u \times n_a} \ \mathbb{I}_{n_u \times n_u}]$ select the vector of generalized joint forces $\boldsymbol{\tau} \in \mathbb{R}^{n_a}$ for the *actuated* joints $n_a$, or $\boldsymbol{\lambda}_u \in \mathbb{R}^{n_u}$ which are the generalized contact forces of the *unactuated* DOF created by the dynamics model, respectively. $\mathbf{J}_c(\mathbf{q}) \in \mathbb{R}^{6n_c \times (n_a+n_u)}$ is the Jacobian matrix and $\boldsymbol{\lambda}_c \in \mathbb{R}^{6n_c}$ are the generalized contact forces at $n_c$ contact points. In our simulated environment, only the pusher object $p$ has actuated joints which sets its initial velocity and heading, while the rest of the joints are either unactuated or created by contacts. The state of the system is represented by $\mathbf{s} \triangleq [\mathbf{q}^T \ \dot{\mathbf{q}}^T]^T$.

## B    TRAYSIM DATASET

Using the simulation setup described in Sec A, we created 100 task sequences using object classes $\mathcal{C} = \{\text{wine glass, martini glass, bottle}\}$ with object instances from these classes arranged roughly in a $3 \times 3$ matrix on the tray. The instance classes and the number of instances are randomly chosen with a minimum of 5 and a maximum of 9. Each task sequence is associated with an auxiliary sequence for parameter estimation that contains at least one object instance from every class of object appearing in the task images. For example, if a task image (that is, the first image in a task sequence) has 3 bottles, then we will have a bottle in the auxiliary sequence. We assume each instance is defined by a triplet: (color, type, location), where the color is unique across all the instances on the tray so that it can be identified across the multi-view images, especially when some views occlude some of the instances. The physical parameters of the objects are assumed to be the same for both the task sequences and the auxiliary sequences, and instances of the same object classes have the same physical parameters. The physics parameters were randomly sampled for each problem in the dataset. We assume the pusher is placed at the same location in both auxiliary and task data; however this location could be arbitrary and different and will not affect our experiments as such locations will be supplied to the simulator in the respective phases and are not part of inference.

**Ground Truth Physics:** When generating each problem instance in the TraySim dataset, the physical parameters of the object classes are randomly chosen within the following ranges: sliding friction in (0.1, 1], inertia and stiffness in (0, 1), and damping in (0, 10). We assume a fixed and known mass for each object type across problem instances, namely we assume a mass of 20 units for bottle, 10 units for martini glass, and 4 units for the wine glass. The tray used a mass of 0.5 and the pusher with a mass 20. Further, for both the task and the auxiliary sequences we assume the pusher is located at the same initial location in the scene. However, for all the auxiliary sequences, we assume the pusher moves with an initial (x, y) velocity of (-4.8, -4.8) m/s towards the tray, while for the task sequences, this velocity could be arbitrary (but given in the problem question), with each component of velocity in the range of [-7, -3] m/s. We further assume that the pusher impact direction coincides with the center of the circular tray in all problem instances.

**Optimization Space:** We note that each object class has a unique physics, i.e., each object class has its own friction, stiffness, damping, and inertia, which are different from other object classes. However, instances of the same class share the same physics. Thus, our optimization space for physics estimation when using 3 object instances, each one from a unique class, is thus 12. For the Phase 2 optimization, the LLM has to reason over the object classes for each object instance in the layout image, their positions in the $3 \times 3$ grid, and their colors. This is a sufficiently larger optimization space, with 10 instance colors to choose from, 3 object classes, and 9 positions on the grid.

**Additional Objects:** In addition to the setup above that we use for the experiments in the main paper, we also experiment with additional object classes in this supplementary materials to show

the scalability of our approach to more number of parameters to optimize. To this end, we consider two additional object classes, namely: i) *flute_glass* with a mass of 15.0, and *champagne_glass*, with again a mass of 15.0. The physics parameters for these classes are sampled from the same range described above. Even when we use these additional classes, the layout uses the same $3 \times 3$ matrix for phase 2, however their Phase 1 evaluation has now $5 \times 4$ variables to optimize instead of 12. We created 10 sequences with these additional objects, as our goal is to ablate on the scalability of our approach, than running on a full evaluation as against the results reported in the main paper.

**Simulation and QA Task:** Each sequence was rendered using the simulator for 200 time steps, each step has a duration of 0.01s. We used the last video frame from the task sequence to check the stability of each instance. Specifically, if the major axis of an object instance in the last frame of a task sequence makes an angle of more than 45 degrees with the ground plane, then we deem that instance as *stable*. We randomly select five object instances and create a multiple choice candidate answer set for the question-answering task, where the ground truth answer is the subset of the candidates that are deemed upright in the last frame. Our QA question is "Which of the object instances on the tray will remain upright when the tray is impacted by a pusher with a velocity of ($x$, $y$) m/s from the location ($loc_x$, $loc_y$) in a direction coinciding with the center of the tray". Without any loss of generality, we assume ($loc_x, loc_y$) are fixed in all cases, although as it is a part of the question and is simulated (and not inferred) any other location of the tray or the pusher will be an issue when inferring using LLMPhy. From an evaluation perspective, keeping the pusher too close to the tray may result in all object instances toppling down, while placing it far with smaller velocity may result in the pusher halting before colliding with the tray. Our choice of the pusher velocity was empirically selected such that in most cases the outcome of the impact is mixed and cannot be guessed from the setup.

## C  Physics Parameter Sensitivity

A natural question one may ask about the TraySim dataset is *"how sensitive are the physics parameters to influence the outcome?* In Figure 6, we show three Phase 1 sequences consisting of the same objects and their layout, however varying the physics attributes as shown in the histogram plots. The pusher velocity is fixed for all the sequences. As can be seen from the figure, varying the parameters result in entirely different stability for the objects after the impact, substantiating that the correct inference of these parameters is important to reproduce the correct the outcome.

## D  Details of LLMPhy Phases

In this section, we detail the inputs and expected outputs provided in each phase of LLMPhy.

### D.1  Phase 1 Prompt and Details

In this phase, we provide as input to the LLM four items: i) a prompt describing the problem setup, the qualitative parameters of the objects (such as mass, height, size of tray, etc.) and the task description, ii) an in-context example consisting of sample trajectories of the object instances from its example auxiliary sequence, iii) a program example that, for the given example auxiliary sequence trajectories, shows their physical parameters and the output structure, and iv) auxiliary task sequence trajectories (from the sequence for which the physical parameters have to be estimated) and a prompt describing what the LLM should do. The in-context example is meant to guide the LLM to understand the setup, the program structure we expect the LLM to produce, and our specific APIs that need to be called from the synthesized program. Figure 7 shows the prompt preamble we use in Phase 1. Please see our Appendix I for the precise example of the full prompt that we use. Figure 7 (bottom) shows an example trajectories LLM should optimizes against.

When iterating over the LLM predictions, we augment the above prompt with the history of all the estimations of the physical parameters that the LLM produced in the previous iterations (extracted from the then generated code) and the $\ell_2$ norm between the generated and ground truth object trajectories for each object instance in the auxiliary sequence, with an additional prompt to the LLM as follows: *"We ran your code in our simulator using the physical parameters you provided below... The error in the prediction of the trajectories using these physical parameters is given below. Can*

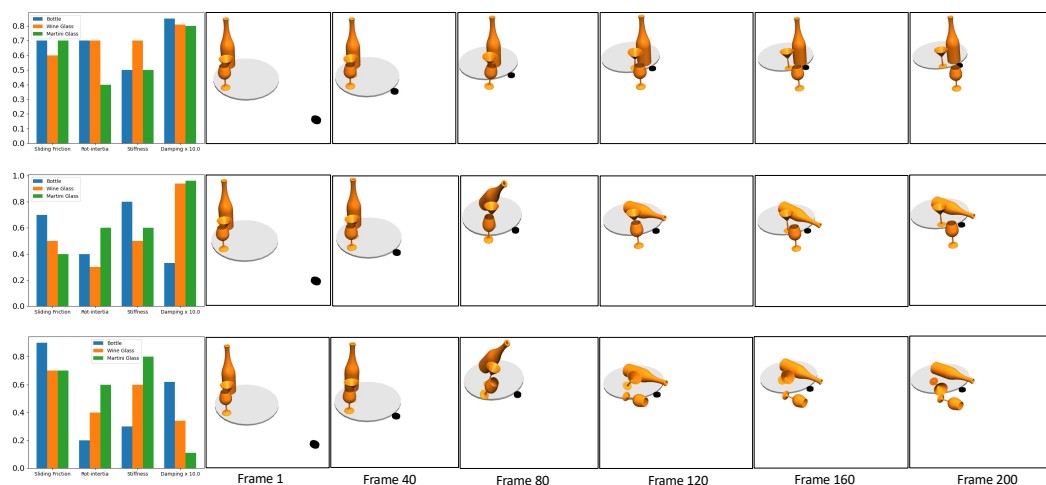

Figure 6: Illustration of the changes in the physical parameters (left histogram, sliding friction, rotation inertia, stiffness, and damping, respectively), and the result of the impact on three objects placed at the same location on the tray (Frame 1) and being impacted by the same force from the pusher. The examples are from the Phase 1 of our dataset. As is clear in the last frame (Frame 200) that changes in the the physical parameters results in entirely different outcomes, substantiating that the estimations of these parameters is important in solving our task.

*you refine your code to make the trajectories look more similar to the ones in given in ...?  Your written code should strictly follow the same code structure as provided in ...*". See Figure 8 for an example. While, we may use computer vision methods for estimating the trajectory of motion of the objects in this Phase, i.e., $\tau$ function in (1), in this work we directly use the trajectories from the simulator for optimization for two reasons: i) we assume the Phase 1 allows complete access to the objects and the setup for parameter estimation, and ii) the focus of this phase is to estimate the physics parameters assuming everything else is known, while the perception task is dealt with in Phase 2. In a real-world setup, we may use AprilTags for producing the object trajectories. This simulation trajectory for Phase 1 will also be provided as part of our TraySim dataset, while also providing the multiview Phase 1 videos for anyone to use vision foundation models for solving the perception problem.

### D.2 Phase 2 Prompt and Details

The goal of the LLM in Phase 2 is to predict the object instance triplet from the multi-view task images. Towards this end, the LLM generates code that incorporates these triplets, so that when this code is executed, the simulator will reproduce the scene layout. Similar to Phase 1, we provide to the LLM an in-context example for guiding its code generation, where this in-context example contains multi-view images and the respective program, with the goal that the LLM learns the relation between parts of the code and the respective multi-view images, and use this knowledge to write code to synthesize the layout of the provided task images. When iterating over the optimization steps, we compute an error feedback to the LLM to improve its previously generated code, where the feedback consists of the following items: i) the program that the LLM synthesized in the previous optimization step, ii) the PSNR between the task image and the simulated image (top-down views), and iii) the color of the object instances in error[5]. Using this feedback, the Phase 2 LLM is prompted to fix the code associated with the triplets in error. Our feedback prompt in Phase 2 thus looks like in the following example: "*The chat history below shows a previous attempt of GPT-4o in generating Python code to reproduce the task images .... For each attempt, we ran the GPT-4o generated code in our simulator and found mistakes. Below we provide the code GPT produced, as well as the PSNR of the generated image against the given top-down image. Can you refine your code to*

---

[5]This is done by inputting a difference image (between the task and synthesized images) to another vision-and-language LLM that is prompted to identify the triplets that are in error

**Prompt Preamble:** The given scene has a tray with three objects (a bottle, a wine_glass, and a martini_glass) on it. The radius of the tray is 1.8 and its center of gravity is 0.05 above the ground with a sliding friction of 0.1 and no spin or roll friction. The radius of bottle is 0.4 and its center of gravity is 1.1 above the ground. The center of gravity of the martini_glass is at a height of 0.5.The center of gravity of the wine_glass is 0.9 above the ground.The tray is impacted by a pusher and the tray with the objects on it moves. Python code in example_code_1.py creates the scene and runs the simulation. The trajectories in object_traj_example_1.txt show the motion of the center of gravity of the objects when running the simulation. Your task is to analyze the given example and then write similar code to produce the trajectories given in 'problem_trajectories.txt'.

You must assume the scene is similar to the one given, however the physics between the tray and the objects are different, that is, the sliding-friction, damping, stiffness, and armature need to be adjusted for all the physical_parameters_for_object_id_* dictionaries in the example_code_1.py so as to reproduce the trajectories in 'problem_trajectories.txt'. You must assume that the physics of the tray with the ground remains the same and so is the external force applied on the tray by the pusher. The trajectories use a timestep of 0.2s. Do not attempt to change the physics parameters beyond their first significant digit. Your written code should strictly follow the same code structure as provided in example_code_1.py. You may further assume that multiple instances of the same object will have the same physical parameters.

You must not change the 'mass' of the objects in your generated code. Do not include the object trajectories in your generated code as that will fail our simulator.

Note that the simulation trajectory in problem_trajectories.txt may use instances of bottle, martini_glass, and wine_glass. The name of the objects is provided in the problem_trajectories.txt file. The mass for the objects are as follows: wine_glass is 4.0, martini_glass is 10.0 and bottle is 20.0.

```
\# nexample_code_1.py

sim = SIMULATOR_MODEL()
sim.create_pusher('3.0 3.0 0.05')
physical_parameters_for_object_id_tray = {
        'sliding-friction': 0.1,
        'armature': 0.1,
        'stiffness': 0.0,
        'mass': 0.5,
        'damping': 20
        }
sim.create_tray(object_physics = physical_parameters_for_object_id_tray)
physical_parameters_for_object_id_1 = {
        'sliding-friction': 0.1,
        'armature': 0.2,
        'stiffness': 0.3,
        'mass': 20.0, # 'mass' is 20.0 for bottle, 10.0 for martini_glass, and 5.0 for wine_glass
        'damping': 5.7
        }
sim.create_object(object_id=1, object_name='bottle', object_location=('row_1', 'column_3'), object_color='orange', object_physics=physical_parameters_for_object_id_1)

physical_parameters_for_object_id_2 = {
        'sliding-friction': 0.5,
        'armature': 0.4,
        'stiffness': 1.0,
        'mass': 10.0, # 'mass' is 20.0 for bottle, 10.0 for martini_glass, and 5.0 for wine_glass
        'damping': 8.8
        }
sim.create_object(object_id=2, object_name='martini_glass', object_location=('row_1', 'column_2'), object_color='orange', object_physics=physical_parameters_for_object_id_2)

...

sim.create_scene()
sim_out=sim.run_simulation()
del sim
```

object_traj_example_1.txt

tray_motion_trajectory (x, y, z) = [(0.0, 0.0, 0.1), (-0.8, -0.8, 0.1), (-1.4, -1.4, 0.1), (-1.8, -1.8, 0.1), (-2.1, -2.1, 0.1), (-2.3, -2.3, 0.1), (-2.4, -2.5, 0.1), (-2.6, -2.6, 0.1), (-2.7, -2.7, 0.1)]

bottle_motion_trajectory (x, y, z) = [(-1.1, -1.1, 1.1), (-1.1, -1.1, 1.1), (-1.1, -1.1, 1.1), (-1.1, -1.1, 1.1), (-1.2, -1.2, 1.1), (-1.3, -1.3, 1.1), (-1.4, -1.5, 1.1), (-1.5, -1.6, 1.1), (-1.6, -1.7, 1.1)]

...

wine_glass_motion_trajectory (x, y, z) = [(-1.0, 1.0, 0.9), (-1.1, 0.9, 1.0), (-1.1, 0.9, 0.8), (-1.2, 0.9, 0.8), (-1.2, 0.9, 0.8), (-1.3, 0.8, 0.8), (-1.3, 0.8, 0.8), (-1.3, 0.8, 0.8), (-1.2, 0.8, 0.8)]

problem_trajectories.txt

tray_motion_trajectory (x, y, z) = [(0.0, 0.0, 0.1), (-0.7, -0.7, 0.1), (-1.1, -1.1, 0.1), (-1.4, -1.4, 0.1), (-1.6, -1.6, 0.1), (-1.8, -1.8, 0.1), (-2.0, -2.0, 0.1), (-2.1, -2.1, 0.1), (-2.2, -2.2, 0.1)]

bottle_motion_trajectory (x, y, z) = [(-1.1, -1.1, 1.1), (-1.1, -1.1, 1.1), (-1.3, -1.3, 1.1), (-1.4, -1.5, 1.1), (-1.5, -1.6, 1.0), (-1.5, -1.6, 0.9), (-1.5, -1.7, 0.6), (-1.5, -1.7, 0.5), (-1.6, -1.8, 0.5)]
...

wine_glass_motion_trajectory (x, y, z) = [(-1.0, 1.0, 0.9), (-1.1, 0.9, 1.0), (-1.2, 0.8, 0.9), (-1.4, 0.8, 0.8), (-1.6, 0.7, 0.8), (-1.6, 0.7, 0.8), (-1.6, 0.7, 0.8), (-1.6, 0.7, 0.8), (-1.5, 0.7, 0.8)]

Figure 7: Top: Prompt preamble used in LLMPhy Phase 1. Middle: The example program provided to the LLM to inform the structure of the to be synthesized program. Bottom: The object instance trajectories provided as input to the LLM. There are two types of trajectories: i) example trajectories (named "object_traj_example_1.txt" for the program example, and ii) "problem_trajectories.txt" which the LLM should produce the program for.

```
We ran your code in our simulator using the physical parameters you provided below in physical_parameters_for_object_id_* dictionaries. The error in the
prediction of the trajectories using these physical parameters is given below. Can you refine your code to make the trajectories look more similar to the ones in
'problem_trajectories.txt'? Your written code should strictly follow the same code structure as provided in example_code_1.py.

GPT Attempt #1 produced :

physical_parameters_for_object_id_1 = {
    'sliding-friction': 0.2,  # Adjusted
    'armature': 0.3,          # Adjusted
    'stiffness': 0.4,         # Adjusted
    'mass': 20.0,             # Mass remains unchanged
    'damping': 6.0            # Adjusted
}

physical_parameters_for_object_id_2 = {
    'sliding-friction': 0.3,  # Adjusted
    'armature': 0.4,          # Adjusted
    'stiffness': 0.7,         # Adjusted
    'mass': 10.0,             # Mass remains unchanged
    'damping': 7.0            # Adjusted
}

physical_parameters_for_object_id_3 = {
    'sliding-friction': 0.3,  # Adjusted
    'armature': 0.5,          # Adjusted
    'stiffness': 0.4,         # Adjusted
    'mass': 4.0,              # Mass remains unchanged
    'damping': 6.0            # Adjusted
}
Trajectory Prediction Error using the above parameters is listed below:
Trajectory Error for bottle = 1.25
Trajectory Error for martini_glass = 2.17
Trajectory Error for wine_glass = 1.22
Total Trajectory Error (including tray)= 6.62
Average Trajectory Error (including tray)= 1.66
```

Figure 8: The prompt shows the LLM feedback, where the parameters from the physical parameter snippet from the synthesized program are extracted to produce the prompt along with the errors the executed code produced (against the trajectories in "problem_trajectories.txt") on each object class. We append all previous responses consecutively when sending the new query to the LLM.

*reproduce the task images correctly? You should not change any part of the code corresponding to correctly inferred objects. ⟨ code ...⟩. Colors of the objects in the code above that are misplaced: colors = {'orange', 'purple', 'cyan'}. PSNR for the generated image against given top-down image = 39.2 Please check the locations of these objects in task_image_top_view_1.png and fix the code accordingly.*". We show a full prompt for the Phase 2 LLM in Sec. I.

## E    PERFORMANCES TO OTHER LLMS

In Table 3, we compare the performance of Phase 1 and Phase 2 of LLMPhy to various alternatives and prior black-box optimization methods. This table includes additional results than those reported in the main paper in Table 1. In Experiments 4–6, we compare to the various black-box optimization methods for estimating the physics parameters while keeping the Phase 2 inference from LLMPhy as in the Experiment 3. To be comparable, we used the same number of iterations for all the methods. As can be noted from the table, LLMPhy leads to better performances compared to other methods in reasoning on the impact outcomes. In Experiments 7–8, we also executed the prior methods for longer number of steps, which improved their performances, however they appear to be still below that of LLMPhy.

In Table 4, we compare the performances to other LLM choices in Phase 1 of LLMPhy. As the experiments that use OpenAI o1 model was conducted on a smaller subset of ten problems from the TraySim dataset, we report only the performance on this subset for all methods. We find that the o1 variant of the models demonstrate better performances against CMA-ES and substantially better than BO.

## F    ABLATION STUDIES

In this section, we analyze various aspects of LLMPhy performance and is reported in Table 5. In addition to Avg. IoU performance as done in the main paper, we also report the 'precise IoU' that counts the number of times the predicted answer (i.e., the set of stable object instances listed in the answer options) match precisely with the ground truth.

| Expt # | Phase 1 | Phase 2 | Avg. IoU (%) |
|---|---|---|---|
| 1 | Random | Random | 19.0 |
| 2 | N/A | LLM | 32.1 |
| 3 | Random | LLMPhy | 50.8 |
| 4 | BO (30 iterations) | LLMPhy | 59.6 |
| 5 | CMA-ES (30 iterations) | LLMPhy | 59.7 |
| 6 | LLMPhy (30 iterations) | LLMPhy | **62.0** |
| 7 | BO (100 iterations) | LLMPhy | 61.0 |
| 8 | CMA-ES (100 iterations) | LLMPhy | 60.7 |
| 9 | Ground Truth (GT) | LLMPhy | 65.1 |
| 10 | CMA-ES | GT | 75.8 |
| 11 | LLMPhy | GT | 77.5 |

Table 3: Performance analysis of LLMPhy Phase 1 and Phase 2 combinations against various alternatives, including related prior methods. We report the intersection-over-union of the predicted answer options and the ground truth answers in the multiple choice solutions.

| Expt # | Phase 1 | Phase 2 | Avg. IoU (%) |
|---|---|---|---|
| 1 | BO | LLMPhy | 49.6 |
| 2 | CMA-ES | LLMPhy | 53.0 |
| 3 | LLMPhy (GPT-4o) | LLMPhy | 53.0 |
| 4 | LLMPhy (o1-mini) | LLMPhy | 55.3 |
| 5 | LLMPhy (o1) | LLMPhy | **57.0** |

Table 4: Performance analysis (on a small subset of 10 examples) of LLMPhy Phase 1 and Phase 2 combinations against various alternatives using various LLMs within LLMPhy.

*1. How will* LLMPhy *scale to more number of object classes?* To answer this question, we extended the TraySim dataset with additional data with five object classes $\mathcal{C}$ = {bottle, martini_glass, wine_glass, flute_glass, champagne_glass}. The last two items having the same mass of 15.0. We created 10 examples with this setup for our ablation study and re-ran all methods on this dataset. Figure 9 show an example of this setup using 5 object classes. The ablation study we report below use this setup. In Expt 1-3 in Table 5, we compare the performance of LLMPhy to BO and CMA-ES. We see that LLM performs the best. We also repeated the experiment in Expt 4-6 using the ground truth (GT) Phase 2 layout, thus specifically evaluating on LLMPhy Phase 1 physics estimation. Again we see the clear benefit in using LLMPhy on both Avg. IoU and Precise IoU, underlining that using more objects and complicating the setup does not affect the performance of our model. We note that all the methods in tis comparison used the same settings, that is the number of optimization iterations was set to 30, and we used o1-mini for LLMPhy.

*2. Robustness of* LLMPhy *Performances?* A natural question is how well do LLMPhy perform in real world settings or when using a different simulation setup. While, it needs significant efforts to create a real-world setup for testing LLMPhy (e.g., that may need programming a robot controller for generating a precise impact for the pusher, etc.) or a significant work to create APIs for a different simulator, we may test the robustness of the framework artificially, for example, by injecting noise to the feedback provided to the LLM/VLM at each iteration. We attempted this route by adding a noise equal to 25% of the smallest prediction error for each of the object instance trajectories in Phase 1. Specifically, we compute $\ell_2$ error between the predicted and the provided object trajectory for each object class in Phase 1 of LLMPhy (let's call it $\{e_k\}_{k=1}^5$), computed the minimum of these errors say $e_m$, and replaced as $\hat{e}_k := e_k + e_m.\zeta/4.0$ for $k = 1, 2, \cdots, 5$ and $\zeta \sim \mathcal{N}(0, 1)$. This will make the LLM essentially uncertain about its physical parameter predictions, while the error (which is sufficiently high given the usual range of the error is between 0.5-4) simulates any underlying errors from a real physical system or simulation errors when using another physics engine. Our results in Expt. 7-8 in Table 5 show that LLMPhy is not very much impacted by the noise. While there is a drop of about 5% in accuracy (72.5% to 67.2%) when using GT, it is still higher than for example, when using CMA-ES on this additional dataset.

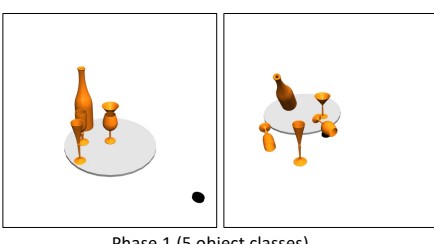 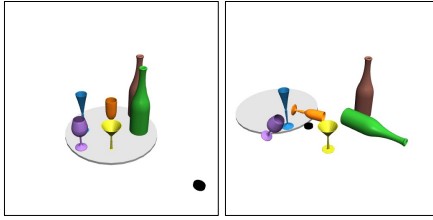

Phase 1 (5 object classes)          Phase 2 (5 object classes)

Figure 9: An example illustrating our extended dataset with 5 object classes.

*3. Advantage of using Optimization Trace?* As we alluded to early on in the paper, one of the differences from prior work such as Ma et al. (2023) is that LLMPhy uses the optimization trace against only the last feedback. In Table 5 Expt 9-10, we compare the performance when not using the full optimization trace. We see a drop of 5% (i.e., 56.4% Avg. IoU to 51.1%) showing that the optimization trace is useful. While using the optimization trace may demand longer context windows, we believe it also helps the LLM to avoid reconsidering previously generated parameter values and thus aids better convergence, especially for black-box optimization approaches, unless there is a provision to include a summary of the optimization trajectory to the LLM in another manner.

| Expt # | Phase 1 | Phase 2 | Avg. IoU (%) | Precise IoU(%) |
|---|---|---|---|---|
| 1 | BO | LLMPhy | 51.2 | 0.0 |
| 2 | CMA-ES | LLMPhy | 39.5 | 0.0 |
| 3 | LLMPhy | LLMPhy | **56.4** | 11.0 |
| 4 | BO | GT | 71.0 | 11.0 |
| 5 | CMA-ES | GT | 63.2 | 22.0 |
| 6 | LLMPhy | GT | **72.5** | **33.0** |
| 7 | LLMPhy + noise | LLMPhy | 52.1 | 22.0 |
| 8 | LLMPhy + noise | GT | 67.2 | 22.0 |
| 9 | LLMPhy (last-only) | LLMPhy | 51.1 | 11.0 |
| 10 | LLMPhy (last-only) | GT | 70.5 | 33.0 |

Table 5: Performance comparison of LLMPhy against alternatives on various scene conditions and when using more number of objects on the simulated tray. In the experiments that show LLMPhy+noise, we perturb the object trajectories with 25% noise so that LLMPhy receives a noisy feedback. In the experiments LLMPhy (last-only), we feedback to LLMPhy only error and the physics parameters from the last iteration, without the full optimization trace.

# G LLMPHY DETAILED CONVERGENCE ANALYSIS

In Figure 10(a), we plot the mean convergence (over a subset of the dataset) when using o1-preview, GPT-4o, o1-mini, Bayesian Optimization, and CMA-ES. We see that the o1 model, that is explicitly trained for solving scientific reasoning, appears to be beneficial in our task. Interestingly, we see that o1's initial convergence is fast, however with longer iterations CMA-ES appears to outperform in minimizing the trajectory error. That being said, the plots in Figure 5(c) and Table 1 points out that having lower trajectory error does not necessarily imply the physical parameters are estimated correctly (as they are implicitly found and are non-linear with regards to the trajectories), and having knowledge of physics in optimization leads to superior results.

Further to this, in Figure 10(d), we plot the histogram of best Phase 1 iterations between the various algorithms. Recall that the optimization methods we use are not based on gradients, instead are sampled discrete points, and the optimization approach is to select the next best sample towards minimizing the error. The plot shows that LLMPhy results in its best sample selections happen early on in its iterations than other methods.

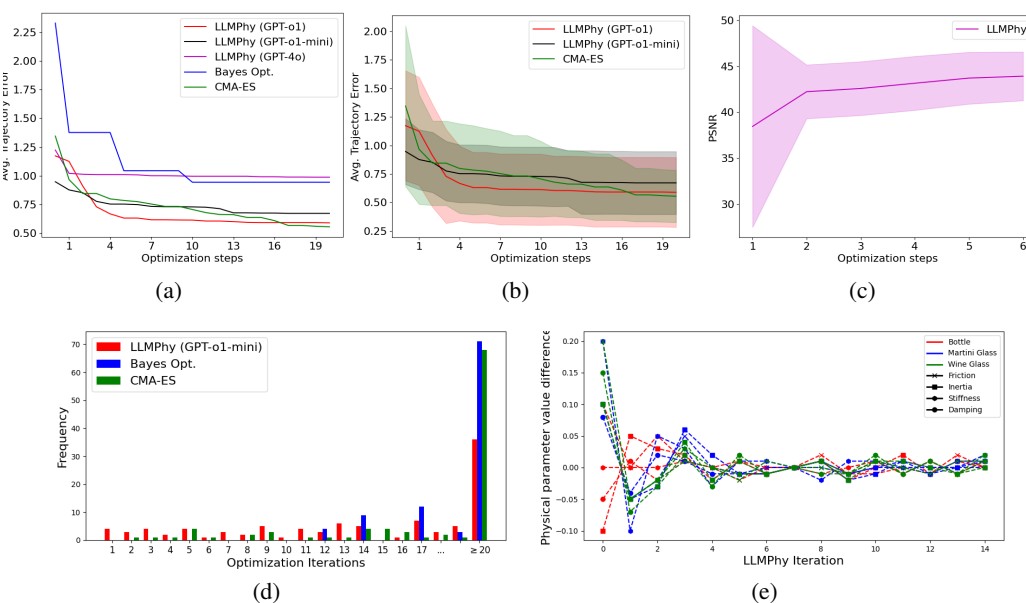

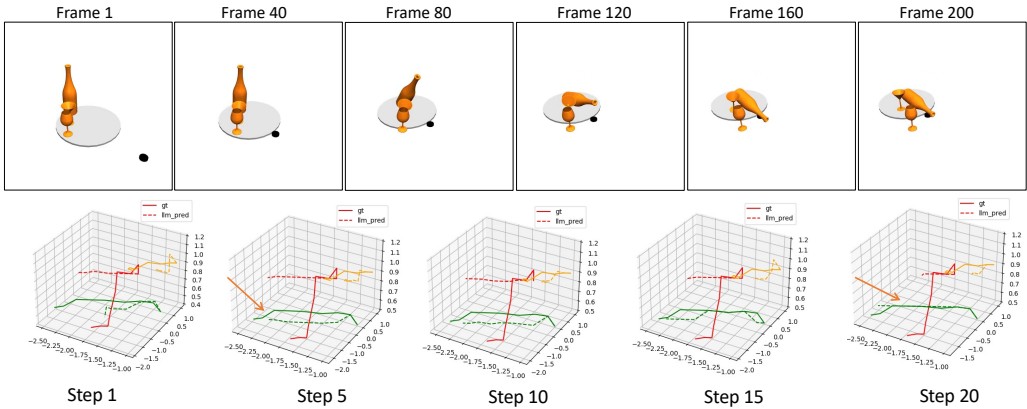

Figure 10: (a) shows comparison of convergence when using various state-of-the-art LLMs in LLMPhy against Bayesian optimization and CMA-ES. We plot the minimum loss computed thus far in the optimization process against the number of optimization steps. (b) plots show the convergence of LLMPhy and the error variance for Phase 1. (c) plots the convergence in Phase 2. We also compare the convergence using OpenAI o1-preview model as the LLM used in LLMPhy. (d) Histogram of the best optimization iteration when using LLMPhy against other methods. (e) shows the differences between subsequent values for the various physical parameters in a typical iteration of LLMPhy from its value in the previous iteration.

Figure 11: We show an example Phase 1 sequence (top). Below, we plot the motion trajectories for each of the objects in the frames and the predicted trajectories by LLMPhy from the optimization steps. The trajectory plots (below) show the ground truth trajectory (gt) and the predicted trajectory (llm_pred), and as the iterations continue, we can see improvements in the alignment of the predicted and the ground truth object trajectories (as pointed out by the arrows).

In Figure 10(e), we plot the optimization parameter trace for one sample sequence, where we plot the differences between the values of the physics parameters produced by the LLM at an iteration against the values from the previous iteration. The plot shows the relative magnitude of changes the LLM makes to the parameters towards adjusting for the object trajectory error. We plot these adjustments for all the three objects and all the four parameters together in one plot so as to see the

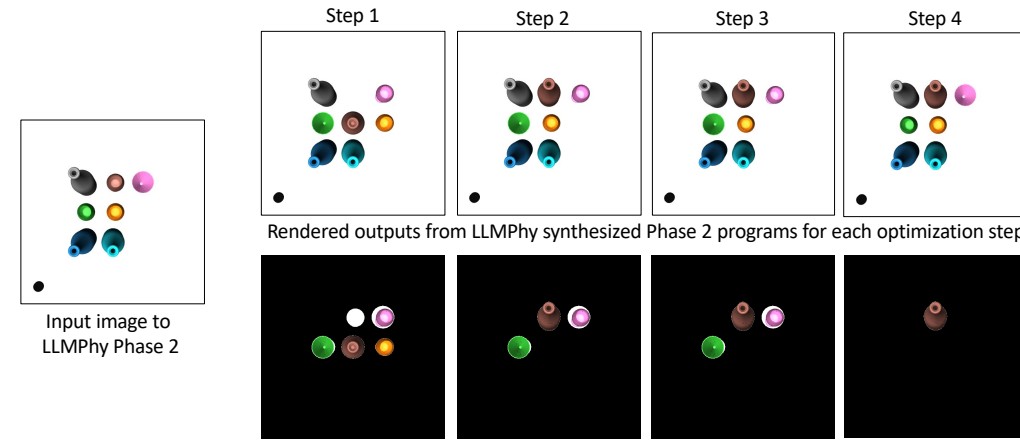

Step 1     Step 2     Step 3     Step 4

Rendered outputs from LLMPhy synthesized Phase 2 programs for each optimization step

Input image to
LLMPhy Phase 2

Difference images showing the error in the layout estimation from each optimization step

Figure 12: We show qualitative results from LLMPhy Phase 2 iterations. The input Phase 2 image is shown on the left. The top row shows the images produced by the simulator using the layout prediction code generated by LLMPhy for each Phase 2 optimization step. Below, we show the difference image between the predicted and the input Phase 2 images, clearly showing the errors. In Phase 2, the feedback to LLMPhy is produced using PSNR computed on the predicted and the ground truth images, as well as asking LLM (using the difference image) which of the objects are in error, and asking the LLM to fix the layout of these objects in the next iteration. As can be seen, the errors in the LLM layout prediction improves over iterations.

overall trend that the LLM makes. We also see that the LLM makes large adjustments in the first few iterations and it reduces in magnitude for subsequently. For this particular example, the LLMPhy converged in 15 iterations.

In Figure 5(a), we plot the convergence of LLMPhy-Phase 1, alongside plotting the variance in the trajectory error from the estimated physical parameters when used in the simulations. We found that a powerful LLM such as OpenAI o1-mini LLM or o1-preview demonstrates compelling convergence, with the lower bound of variance below that of other models. Our experiments suggest that better LLMs may lead to even stronger results.

In Figure 5(b), we plot the convergence of LLMPhy Phase 2 iterations improving the PSNR between the synthesized (using the program) and the provide task images. As is clear, their correctness of the program improves over iterations. We would like to emphasize that BO and CMA-ES are continuous optimization methods and thus cannot optimize over the discrete space of Phase 2 layout. This is an important benefit of using LLMPhy for optimization that can operate on both continuous and discrete state spaces.

## H  QUALITATIVE RESULTS

In Figure 13, we show several qualitative results from our TraySim dataset and comparisons of LLMPhy predictions to those of BO and CMA-ES. In general, we find that when the velocity of the pusher is lower, and the sliding friction is high, objects tend to stay stable if they are heavier (e.g., a bottle), albeit other physics parameters also playing into the outcome. In Figure 11, we show example iterations from Phase 1 that explicitly shows how the adjustment of the physical parameters by LLMPhy is causing the predicted object trajectories to align with the ground truth. In Figure 12, we show qualitative outputs from the optimization steps in Phase 2, demonstrating how the error feedback to the LLM corrects its previous mistakes to improve the layout estimation.

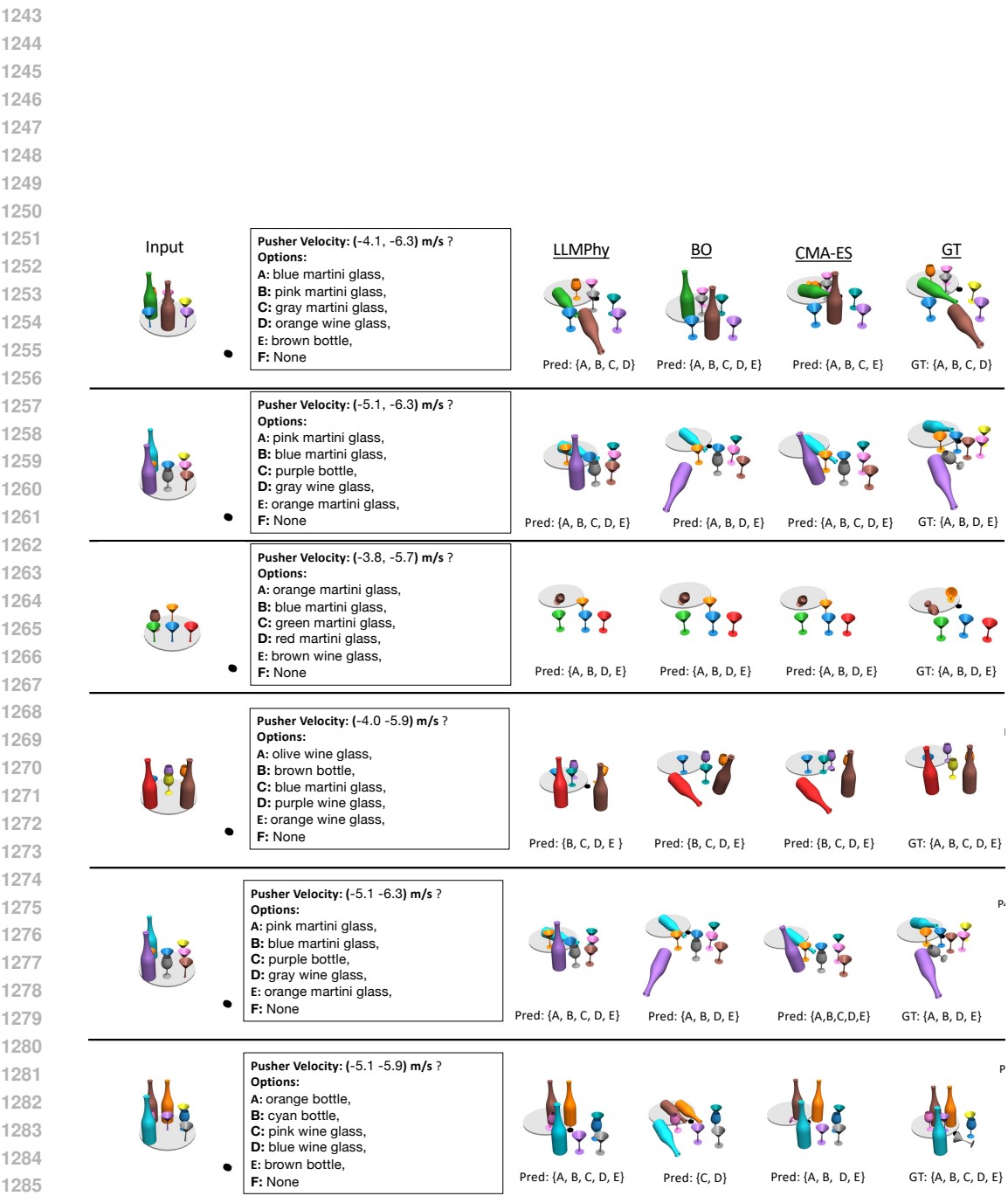

Figure 13: Qualitative comparisons between LLMPhy, Bayesian optimization, and CMA-ES.

# I LLMPHY OPTIMIZATION TRACE, PROGRAM SYNTHESIS, AND LLM INTERACTIONS

Below, we present the exact prompts we used for the LLM in our experiments for Phases 1 and 2, as well as depicting the programs LLM generate.

**Phase 1 Prompt:**

The given scene has a tray with three objects (a bottle, a wine_glass, and a martini_glass) on it. The radius of the tray is 1.8 and its center of gravity is 0.05 above the ground with a sliding friction of 0.1 and no spin or roll friction. The radius of bottle is 0.4 and its center of gravity is 1.1 above the ground. The center of gravity of the martini_glass is at a height of 0.5.The center of gravity of the wine_glass is 0.9 above the ground. The tray is impacted by a pusher and the tray with the objects on it moves. Python code in example_code_1.py creates the scene and runs the simulation. The trajectories in object_traj_example_1.txt show the motion of the center of gravity of the objects when running the simulation. Your task is to analyze the given example and then write similar code to produce the trajectories given in 'problem_trajectories.txt'.

You must assume the scene is similar to the one given, however the physics between the tray and the objects are different, that is, the sliding-friction, damping, stiffness, and armature need to be adjusted for all the physical_parameters_for_object_id_* dictionaries in the example_code_1.py so as to reproduce the trajectories in 'problem_trajectories.txt'. You must assume that the physics of the tray with the ground remains the same and so is the external force applied on the tray by the pusher. The trajectories use a time step of 0.2s. Do not attempt to change the physics parameters beyond their first significant digit. Your written code should strictly follow the same code structure as provided in example_code_1.py. You may further assume that multiple instances of the same object will have the same physical parameters.

You must not change the 'mass' of the objects in your generated code. Do not include the object trajectories in your generated code as that will fail our simulator.

Note that the simulation trajectory in problem_trajectories.txt may use instances of bottle, martini_glass, and wine_glass. The name of the objects is provided in the problem_trajectories.txt file. The mass for the objects are as follows: wine_glass is 4.0, martini_glass is 10.0 and bottle is 20.0.''

```
\# nexample\_code\_1.py
sim = SIMULATOR_MODEL()
sim.create_pusher('3.0 3.0 0.05')
physical_parameters_for_object_id_tray = {
            'sliding-friction': 0.1,
            'armature': 0.1,
            'stiffness': 0.0,
            'mass': 0.5,
            'damping': 20
        }
sim.create_tray(object_physics = physical_parameters_for_object_id_tray)
physical_parameters_for_object_id_1 = {
            'sliding-friction': 0.1,
            'armature': 0.2,
```

```
                    'stiffness': 0.3,
                    'mass': 20.0,
                    'damping': 5.7
            }
sim.create_object(object_id=1, object_name='bottle',
object_location=('row_1', 'column_3'), object_color='orange',
object_physics=physical_parameters_for_object_id_1)
...

sim.create_scene()
sim_out=sim.run_simulation()
del sim

# object\_traj\_example\_1.txt
...

bottle_motion_trajectory (x, y, z) = [(-1.1, -1.1, 1.1), (-1.1, -1.1,
1.1), (-1.1, -1.1, 1.1), (-1.1, -1.1, 1.1), (-1.2, -1.2, 1.1), (-1.3,
-1.3, 1.1), (-1.4, -1.5, 1.1), (-1.5, -1.6, 1.1), (-1.6, -1.7, 1.1)]

martini_glass_motion_trajectory (x, y, z) = [(-1.0, 0.0, 0.5), (-1.1,
-0.0, 0.6), (-1.2, -0.1, 0.6), (-1.4, -0.4, 0.5), (-1.6, -0.6, 0.5),
(-1.8, -0.8, 0.5), (-2.0, -0.9, 0.5), (-2.1, -1.0, 0.5), (-2.2, -1.1,
0.5)]

...
```

**Phase 2 Prompt:**

```
Attached are two images: 'example_1_top_down_view_1.png' (top-down view)
and 'example_1_side_view_2.png' (side view) of the same scene. The top-
down view shows a scene arranged roughly on a 3x3 grid. The scene was
rendered using the code in 'example_code_1.py'. Objects in the scene
belong to one of the following classes: {martini_glass, wine_glass,
bottle} and can be one of the following colors: {purple, red, green,
blue, olive, cyan, brown, pink, orange, gray}. Each color appears only
once in the scene. Can you interpret the provided code using the images?
Use the top-down image to determine the arrangement and color of the
objects, and correlate this with the side view to identify the object
classes. Each object instance has a unique color, helping you identify
the same object across different views.

example_1_top_down_view_1.png
Image: top-down-image url
example_1_side_view_2.png
Image: side-view image url

example_code_1.py

sim = SIMULATOR_MODEL()
sim.create_pusher('3.0 3.0 0.05')
physical_parameters_for_object_id_tray = {
        'sliding-friction': 0.1,
        'armature': 0.1,
        'stiffness': 0.0,
        'mass': 0.5,
        'damping': 20
    }
sim.create_tray(object_physics = physical_parameters_for_object_id_tray)
physical_parameters_for_object_id_1 = {
        'sliding-friction': 0.1,
        'armature': 0.2,
```

```
        'stiffness': 0.3,
        'mass': 20.0, # 'mass' is 20.0 for bottle, 10.0 for
            martini_glass, and 5.0 for wine_glass
        'damping': 5.7
    }
sim.create_object(object_id=1, object_name='bottle', object_location=('
    row_2', 'column_3'), object_color='brown', object_physics=
    physical_parameters_for_object_id_1)

physical_parameters_for_object_id_2 = {
        'sliding-friction': 0.6,
        'armature': 0.8,
        'stiffness': 0.6,
        'mass': 4.0, # 'mass' is 20.0 for bottle, 10.0 for
            martini_glass, and 5.0 for wine_glass
        'damping': 8.3
    }
sim.create_object(object_id=2, object_name='wine_glass', object_location
    =('row_3', 'column_2'), object_color='pink', object_physics=
    physical_parameters_for_object_id_2)

physical_parameters_for_object_id_3 = {
        'sliding-friction': 0.1,
        'armature': 0.2,
        'stiffness': 0.3,
        'mass': 20.0, # 'mass' is 20.0 for bottle, 10.0 for
            martini_glass, and 5.0 for wine_glass
        'damping': 5.7
    }
sim.create_object(object_id=3, object_name='bottle', object_location=('
    row_1', 'column_1'), object_color='purple', object_physics=
    physical_parameters_for_object_id_3)

physical_parameters_for_object_id_4 = {
        'sliding-friction': 0.1,
        'armature': 0.2,
        'stiffness': 0.3,
        'mass': 20.0, # 'mass' is 20.0 for bottle, 10.0 for
            martini_glass, and 5.0 for wine_glass
        'damping': 5.7
    }
sim.create_object(object_id=4, object_name='bottle', object_location=('
    row_1', 'column_2'), object_color='olive', object_physics=
    physical_parameters_for_object_id_4)

physical_parameters_for_object_id_5 = {
        'sliding-friction': 0.1,
        'armature': 0.2,
        'stiffness': 0.3,
        'mass': 20.0, # 'mass' is 20.0 for bottle, 10.0 for
            martini_glass, and 5.0 for wine_glass
        'damping': 5.7
    }
sim.create_object(object_id=5, object_name='bottle', object_location=('
    row_3', 'column_1'), object_color='orange', object_physics=
    physical_parameters_for_object_id_5)

physical_parameters_for_object_id_6 = {
        'sliding-friction': 0.5,
        'armature': 0.4,
        'stiffness': 1.0,
        'mass': 10.0, # 'mass' is 20.0 for bottle, 10.0 for
            martini_glass, and 5.0 for wine_glass
        'damping': 8.8
    }
```

```
1458    sim.create_object(object_id=6, object_name='martini_glass',
1459        object_location=('row_2', 'column_2'), object_color='cyan',
1460        object_physics=physical_parameters_for_object_id_6)
1461
1462    physical_parameters_for_object_id_7 = {
1463            'sliding-friction': 0.5,
1464            'armature': 0.4,
1465            'stiffness': 1.0,
1466            'mass': 10.0, # 'mass' is 20.0 for bottle, 10.0 for
1467                martini_glass, and 5.0 for wine_glass
1468            'damping': 8.8
1469        }
1470    sim.create_object(object_id=7, object_name='martini_glass',
1471        object_location=('row_2', 'column_1'), object_color='gray',
1472        object_physics=physical_parameters_for_object_id_7)
1473
1474    physical_parameters_for_object_id_8 = {
1475            'sliding-friction': 0.5,
1476            'armature': 0.4,
1477            'stiffness': 1.0,
1478            'mass': 10.0, # 'mass' is 20.0 for bottle, 10.0 for
1479                martini_glass, and 5.0 for wine_glass
1480            'damping': 8.8
1481        }
1482    sim.create_object(object_id=8, object_name='martini_glass',
1483        object_location=('row_3', 'column_3'), object_color='green',
1484        object_physics=physical_parameters_for_object_id_8)
1485
1486    physical_parameters_for_object_id_9 = {
1487            'sliding-friction': 0.1,
1488            'armature': 0.2,
1489            'stiffness': 0.3,
1490            'mass': 20.0, # 'mass' is 20.0 for bottle, 10.0 for
1491                martini_glass, and 5.0 for wine_glass
1492            'damping': 5.7
1493        }
1494    sim.create_object(object_id=9, object_name='bottle', object_location=('
1495        row_1', 'column_3'), object_color='blue', object_physics=
1496        physical_parameters_for_object_id_9)

    sim.create_scene()
    sim_out=sim.run_simulation()
    del sim

    Using the above information, can you write code similar to '
        example_code_1.py' to reproduce the two images given below for a
        different scene? The images are named: 'task_image_top_down_view_1.
        png' for the top-down view of the scene and 'task_image_side_view_2.
        png' for the side-view of the same scene. Note that not all positions
         on the grid need to have an object.
    task_image_top_view_1.png
    Image: top-down image url
    task_image_side_view_2.png
    Image: side-view image url
    You should further use the following set of physical attributes for the
    respective objects in the scene when generating the code. Note that all
    the instances of the same object use the same physical attributes.
    object_name: bottle, mass: 20.0, 'sliding-friction': 0.3, 'armature':
        0.5, 'stiffness': 0.4, 'damping': 6.3

    object_name: martini_glass, mass: 10.0, 'sliding-friction': 0.5, '
        armature': 0.3, 'stiffness': 1.2, 'damping': 8.3
```

```
object_name: wine_glass, mass: 4.0, 'sliding-friction': 0.6, 'armature':
    0.7, 'stiffness': 0.8, 'damping': 7.8
```

The chat history below shows various attempts of GPT-4o in generating
Python code to reproduce the task images (task_image_top_view_1.png,
task_image_side_view_2.png). For each attempt, we ran the GPT-4o
generated code in our simulator and found mistakes. Below we provide the
code GPT produced in each attempt, titled GPT Attempt #K, where K is the
attempt number, as well as the PSNR of the generated image against the
given top-down image. Can you refine your code to reproduce the task
images correctly? You should not change any part of the code
corresponding to correctly inferred objects.

GPT Attempt #0:

GPT generated code:

```
sim = SIMULATOR_MODEL()
sim.create_pusher('3.0 3.0 0.05')

physical_parameters_for_object_id_tray = {
    'sliding-friction': 0.1,
    'armature': 0.1,
    'stiffness': 0.0,
    'mass': 0.5,
    'damping': 20
}
sim.create_tray(object_physics=physical_parameters_for_object_id_tray)

# Defining the physical parameters for each class of objects
physical_parameters_for_bottle = {
    'sliding-friction': 0.3,
    'armature': 0.5,
    'stiffness': 0.4,
    'mass': 20.0,
    'damping': 6.3
}

physical_parameters_for_wine_glass = {
    'sliding-friction': 0.6,
    'armature': 0.7,
    'stiffness': 0.8,
    'mass': 4.0,
    'damping': 7.8
}

physical_parameters_for_martini_glass = {
    'sliding-friction': 0.5,
    'armature': 0.3,
    'stiffness': 1.2,
    'mass': 10.0,
    'damping': 8.3
}

# Creating objects based on analysis of the task image
sim.create_object(object_id=1, object_name='bottle', object_location=('
    row_2', 'column_1'), object_color='gray', object_physics=
    physical_parameters_for_bottle)
sim.create_object(object_id=2, object_name='wine_glass', object_location
    =('row_1', 'column_1'), object_color='yellow', object_physics=
    physical_parameters_for_wine_glass)
sim.create_object(object_id=3, object_name='wine_glass', object_location
    =('row_3', 'column_1'), object_color='blue', object_physics=
    physical_parameters_for_wine_glass)
```

```
sim.create_object(object_id=4, object_name='martini_glass',
    object_location=('row_2', 'column_2'), object_color='orange',
    object_physics=physical_parameters_for_martini_glass)
sim.create_object(object_id=5, object_name='bottle', object_location=('
    row_2', 'column_3'), object_color='red', object_physics=
    physical_parameters_for_bottle)
sim.create_object(object_id=6, object_name='bottle', object_location=('
    row_3', 'column_2'), object_color='pink', object_physics=
    physical_parameters_for_bottle)

# Adding the missing martini glass with cyan color at row 1, column 2
sim.create_object(object_id=7, object_name='martini_glass',
    object_location=('row_1', 'column_3'), object_color='cyan',
    object_physics=physical_parameters_for_martini_glass)

# Run the simulation
sim.create_scene()
sim_out = sim.run_simulation()
del sim

Colors of the objects in the code above that are misplaced: {'purple', '
    cyan', 'pink'}

PSNR for the generated image against given top-down image = 40.9
Please check the locations of these objects in task_image_top_view_1.png
    and fix the code accordingly.

End of Prompt
```

## J   EXAMPLE OF SYNTHESIZED PROGRAMS

Below, we show an example of Python programs synthesized by LLMPhy, which are then executed to invoke the Mujoco Simulator to run and render the dynamical sequence, to extract the outcome of the simulation.

**Phase 1 program:**

```
sim = SIMULATOR_MODEL()
sim.create_pusher('3.0 3.0 0.05')
physical_parameters_for_object_id_tray = {
            'sliding-friction': 0.1,
            'armature': 0.1,
            'stiffness': 0.0,
            'mass': 0.5,
            'damping': 20
        }
sim.create_tray(object_physics = physical_parameters_for_object_id_tray)

physical_parameters_for_object_id_1 = {
    'sliding-friction': 0.18,  # finely adjusted as previous good attempt
    'armature': 0.1,           # keeping consistent based on the effective value
    'stiffness': 0.16,         # slight readjustment from previous effective value
    'mass': 20.0,              # mass remains 20.0 for bottle
    'damping': 4.5             # slight decrease for finer adjustment
}

physical_parameters_for_object_id_2 = {
    'sliding-friction': 0.35,  # minor adjustment from better attempt
    'armature': 0.26,          # further decreasing for better control
    'stiffness': 0.75,         # slight decrease for better compliance
```

```
1620        'mass': 10.0,                # mass remains 10.0 for martini_glass
1621        'damping': 6.25              # slight adjustment for better trajectory control
1622    }
1623
1624    physical_parameters_for_object_id_3 = {
1625        'sliding-friction': 0.42,  # slight decrease for better fit
1626        'armature': 0.58,          # minor decrease for better control
1627        'stiffness': 0.39,         # minor decrease for better adjustment
1628        'mass': 4.0,               # mass remains 4.0 for wine_glass
1629        'damping': 6.55            # slight decrease for finer adjustment
1630    }
1631    sim.create_object(object_id=1, object_name='bottle',
1632        object_location=('row_1', 'column_3'),
1633        object_color='orange',
1634        object_physics=physical_parameters_for_object_id_1)
1635
1636    sim.create_object(object_id=2, object_name='martini_glass',
1637        object_location=('row_1', 'column_2'),
1638        object_color='orange',
1639        object_physics=physical_parameters_for_object_id_2)
1640
1641    sim.create_object(object_id=3, object_name='wine_glass',
1642        object_location=('row_1', 'column_1'),
1643            object_color='orange',
1644            object_physics=physical_parameters_for_object_id_3)
1645
1646    sim.create_scene()
1647    sim_out = sim.run_simulation()
1648    del sim
```

**Phase 2 program:**

```
sim = SIMULATOR_MODEL()
sim.create_pusher('3.0 3.0 0.05')

# Set physical attributes for the tray
physical_parameters_for_object_id_tray = {
    'sliding-friction': 0.1,
    'armature': 0.1,
    'stiffness': 0.0,
    'mass': 0.5,
    'damping': 20
}
sim.create_tray(object_physics=physical_parameters_for_object_id_tray)

# Physical parameters for each class of objects
physical_parameters_bottle = {
    'sliding-friction': 0.9,
    'armature': 0.2,
    'stiffness': 0.3,
    'mass': 20.0,
    'damping': 6.2
}

physical_parameters_wine_glass = {
    'sliding-friction': 0.7,
    'armature': 0.4,
    'stiffness': 0.6,
    'mass': 4.0,
```

```
1674        'damping': 3.4
1675    }
1676
1677    physical_parameters_martini_glass = {
1678        'sliding-friction': 0.7,
1679        'armature': 0.6,
1680        'stiffness': 0.8,
1681        'mass': 10.0,
1682        'damping': 1.1
1683    }
1684
1685    # Create objects with corrected positions and colors
1686    sim.create_object(object_id=1, object_name='wine_glass',
1687        object_location=('row_1', 'column_1'),
1688        object_color='gray',
1689        object_physics=physical_parameters_wine_glass)
1690
1691    sim.create_object(object_id=2, object_name='wine_glass',
1692        object_location=('row_1', 'column_2'),
1693        object_color='red',
1694        object_physics=physical_parameters_wine_glass)
1695
1696    sim.create_object(object_id=3, object_name='wine_glass',
1697        object_location=('row_1', 'column_3'),
1698        object_color='pink',
1699        object_physics=physical_parameters_wine_glass)
1700
1701    sim.create_object(object_id=4, object_name='martini_glass',
1702        object_location=('row_2', 'column_1'),
1703        object_color='cyan',
1704        object_physics=physical_parameters_martini_glass)
1705
1706    sim.create_object(object_id=5, object_name='bottle',
1707        object_location=('row_2', 'column_2'),
1708        object_color='yellow',
1709        object_physics=physical_parameters_bottle)
1710
1711    sim.create_object(object_id=6, object_name='bottle',
1712        object_location=('row_2', 'column_3'),
1713        object_color='purple',
1714        object_physics=physical_parameters_bottle)
1715
1716    sim.create_object(object_id=7, object_name='martini_glass',
1717        object_location=('row_3', 'column_1'), object_color='blue',
1718        object_physics=physical_parameters_martini_glass)
1719
1720    sim.create_object(object_id=8, object_name='bottle',
1721        object_location=('row_3', 'column_2'),
1722        object_color='orange',
1723        object_physics=physical_parameters_bottle)
1724
1725    sim.create_object(object_id=9, object_name='martini_glass',
1726        object_location=('row_3', 'column_3'),
1727        object_color='brown',
        object_physics=physical_parameters_martini_glass)

    sim.create_scene()
    sim_out = sim.run_simulation()
    del sim
```

# K LLMPHY OPTIMIZATION AND INTERACTION TRACE (PHASE1)

Below, we provide the full log of the optimization iterations produced in phase 1 by LLMPhy. Specifically, the text shows how the LLM uses the provided feedback to adjust the physical attributes, with explanation of the justification for the adjustments. Note that each response from the LLM is separated by strings "======".

```
In order to produce the desired trajectories given in '
    problem_trajectories.txt', we need to adjust the physics parameters
    for the objects ('bottle', 'wine_glass', and 'martini_glass') in the
    scene. Based on the instructions, we need to adjust the 'sliding-
    friction', 'damping', 'stiffness', and 'armature' parameters. We
    should follow the given code structure closely and only change the
    parameters to their first significant digit.

Below is the adjusted Python code for 'example_code_1.py' to produce the
    desired trajectories:

'''python
# example_code_problem.py

sim = SIMULATOR_MODEL()
sim.create_pusher('3.0 3.0 0.05')

# The physics of the tray with the ground remains the same
physical_parameters_for_object_id_tray = {
        'sliding-friction': 0.1,
        'armature': 0.1,
        'stiffness': 0.0,
        'mass': 0.5,
        'damping': 20
    }
sim.create_tray(object_physics = physical_parameters_for_object_id_tray)

# Adjusted physics parameters for the bottle
physical_parameters_for_object_id_1 = {
        'sliding-friction': 0.2,
        'armature': 0.3,
        'stiffness': 0.4,
        'mass': 20.0, # mass remains unchanged
        'damping': 6.5
    }
sim.create_object(object_id=1, object_name='bottle', object_location=('
    row_1', 'column_3'), object_color='orange', object_physics=
    physical_parameters_for_object_id_1)

# Adjusted physics parameters for the martini_glass
physical_parameters_for_object_id_2 = {
        'sliding-friction': 0.6,
        'armature': 0.5,
        'stiffness': 1.1,
        'mass': 10.0, # mass remains unchanged
        'damping': 9.0
    }
sim.create_object(object_id=2, object_name='martini_glass',
    object_location=('row_1', 'column_2'), object_color='orange',
    object_physics=physical_parameters_for_object_id_2)

# Adjusted physics parameters for the wine_glass
physical_parameters_for_object_id_3 = {
        'sliding-friction': 0.8,
        'armature': 0.9,
```

```python
            'stiffness': 1.0,
            'mass': 4.0, # mass remains unchanged
            'damping': 8.4
        }
sim.create_object(object_id=3, object_name='wine_glass', object_location
    =('row_1', 'column_1'), object_color='orange', object_physics=
    physical_parameters_for_object_id_3)

sim.create_scene()
sim_out = sim.run_simulation()
del sim
```

In this adjusted code, I have modified the `sliding-friction`, `damping`,
    `stiffness`, and `armature` parameters for each object to produce
    the desired motion trajectories as suggested by the `
    problem_trajectories.txt` file. These modifications respect the
    constraint that parameters can only be adjusted up to their first
    significant digit.

You can now run this simulation code to generate the trajectories which
    should match closely to the ones provided in `problem_trajectories.
    txt`.

========

Let's see if we can refine the physical parameters for the objects to
    reduce the trajectory errors. To do this, we need to adjust the
    parameters more carefully while adhering to the given constraints.

Let's analyze the trajectories and try to make minor adjustments to the
    parameters:

1. **Bottle:** The predicted trajectory error is 1.9, implying the need
    for finer adjustments. The last attempt modified the parameters to:
    - 'sliding-friction': 0.2
    - 'armature': 0.3
    - 'stiffness': 0.4
    - 'damping': 6.5

2. **Martini_glass:** The predicted trajectory error is relatively high
    at 3.16. We will need to make significant adjustments.
    - 'sliding-friction': 0.6
    - 'armature': 0.5
    - 'stiffness': 1.1
    - 'damping': 9.0

   Let's reduce the damping a bit and adjust other parameters slightly.

3. **Wine_glass:** The predicted trajectory error is low at 0.24.
    Therefore, fine adjustments may be enough.
    - 'sliding-friction': 0.8
    - 'armature': 0.9
    - 'stiffness': 1.0
    - 'damping': 8.4

### New physical parameters:

```python
sim = SIMULATOR_MODEL()
sim.create_pusher('3.0 3.0 0.05')
physical_parameters_for_object_id_tray = {
    'sliding-friction': 0.1,
    'armature': 0.1,
    'stiffness': 0.0,
```

```
     'mass': 0.5,
     'damping': 20
}

sim.create_tray(object_physics = physical_parameters_for_object_id_tray)

physical_parameters_for_object_id_1 = {
    'sliding-friction': 0.25, # slight increase in friction
    'armature': 0.4,      # slight increase in armature
    'stiffness': 0.5,     # slight increase in stiffness
    'mass': 20.0,         # mass remains unchanged
    'damping': 7.0        # slight increase in damping
}

sim.create_object(object_id=1, object_name='bottle', object_location=('
    row_1', 'column_3'), object_color='orange', object_physics=
    physical_parameters_for_object_id_1)

physical_parameters_for_object_id_2 = {
    'sliding-friction': 0.55, # slight decrease in friction
    'armature': 0.6,      # slight increase in armature
    'stiffness': 1.2,     # slight increase in stiffness
    'mass': 10.0,         # mass remains unchanged
    'damping': 8.5        # decrease in damping
}

sim.create_object(object_id=2, object_name='martini_glass',
    object_location=('row_1', 'column_2'), object_color='orange',
    object_physics=physical_parameters_for_object_id_2)

physical_parameters_for_object_id_3 = {
    'sliding-friction': 0.75, # slight decrease in friction
    'armature': 1.0,      # increase in armature
    'stiffness': 0.9,     # slight decrease in stiffness
    'mass': 4.0,          # mass remains unchanged
    'damping': 8.2        # slight decrease in damping
}

sim.create_object(object_id=3, object_name='wine_glass', object_location
    =('row_1', 'column_1'), object_color='orange', object_physics=
    physical_parameters_for_object_id_3)

sim.create_scene()
sim_out=sim.run_simulation()
del sim
```

Please use the above code to run the simulation again and see if the
    trajectory errors decrease. Adjustments have been made based on the
    analysis of the previous attempt.

========

Based on the trend and error information from the previous attempts, we'
    ll refine the parameters further to attempt a better match for the
    trajectories in 'problem_trajectories.txt'. The key insights from
    previous attempts suggest that we need finer adjustments,
    particularly to friction, damping, and stiffness.

Here's an updated attempt:

```python
# Code structure and constants same as provided

sim = SIMULATOR_MODEL()
```

```
sim.create_pusher('3.0 3.0 0.05')
physical_parameters_for_object_id_tray = {
    'sliding-friction': 0.1,
    'armature': 0.1,
    'stiffness': 0.0,
    'mass': 0.5,
    'damping': 20
}
sim.create_tray(object_physics=physical_parameters_for_object_id_tray)

# Refine bottle parameters
physical_parameters_for_object_id_1 = {
    'sliding-friction': 0.23, # slight increase in friction
    'armature': 0.45,    # increase in armature
    'stiffness': 0.55,   # slight increase in stiffness
    'mass': 20.0,        # mass remains unchanged
    'damping': 7.5       # increase in damping
}
sim.create_object(object_id=1, object_name='bottle', object_location=('
    row_1', 'column_3'), object_color='orange', object_physics=
    physical_parameters_for_object_id_1)

# Refine martini_glass parameters
physical_parameters_for_object_id_2 = {
    'sliding-friction': 0.54, # slight decrease in friction
    'armature': 0.65,    # increase in armature
    'stiffness': 1.25,   # increase in stiffness
    'mass': 10.0,        # mass remains unchanged
    'damping': 8.3       # slight decrease in damping
}
sim.create_object(object_id=2, object_name='martini_glass',
    object_location=('row_1', 'column_2'), object_color='orange',
    object_physics=physical_parameters_for_object_id_2)

# Refine wine_glass parameters
physical_parameters_for_object_id_3 = {
    'sliding-friction': 0.72, # slight decrease in friction
    'armature': 1.05,    # increase in armature
    'stiffness': 0.95,   # slight decrease in stiffness
    'mass': 4.0,         # mass remains unchanged
    'damping': 8.1       # decrease in damping
}
sim.create_object(object_id=3, object_name='wine_glass', object_location
    =('row_1', 'column_1'), object_color='orange', object_physics=
    physical_parameters_for_object_id_3)

# Run the simulation
sim.create_scene()
sim_out = sim.run_simulation()
del sim
```
```

This refinement aims to balance the dynamics and interactions of the
    objects with the tray, considering the error metrics from previous
    attempts. Monitoring the changes in each property helps to
    iteratively converge towards a more accurate trajectory.

========

Based on the provided attempts and their errors, let's attempt to refine
    the physical parameters for the objects in such a way to specifically
    address the notable errors. Here's another attempt:

1. The bottle ('object_id=1') has a relatively high trajectory error, so
    we may need to adjust its parameters further to get a close match.

2. The martini_glass (`object_id=2`) has the highest trajectory error,
   requiring focus on reducing its discrepancy.
3. The wine_glass (`object_id=3`) has the least error, suggesting that
   its parameters are close to what's needed but might still need slight
   adjustments to further align trajectories.

Considering these points, we'll make small but precise adjustments,
   particularly targeting increased damping for the bottle, slightly
   decreasing friction for the martini_glass, and incrementally changing
   the armature and stiffness for both martini_glass and wine_glass.

Here is the code:

```python
# Generated Python code to accurately reproduce the trajectories

sim = SIMULATOR_MODEL()
sim.create_pusher('3.0 3.0 0.05')

physical_parameters_for_object_id_tray = {
    'sliding-friction': 0.1,
    'armature': 0.1,
    'stiffness': 0.0,
    'mass': 0.5,
    'damping': 20
}
sim.create_tray(object_physics = physical_parameters_for_object_id_tray)

physical_parameters_for_object_id_1 = {
    'sliding-friction': 0.22, # slight decrease in friction to adjust
        movement
    'armature': 0.43,    # slight decrease in armature
    'stiffness': 0.53,   # slight decrease in stiffness
    'mass': 20.0,        # mass remains unchanged
    'damping': 7.4       # slight decrease in damping
}
sim.create_object(object_id=1, object_name='bottle', object_location=('
    row_1', 'column_3'), object_color='orange', object_physics=
    physical_parameters_for_object_id_1)

physical_parameters_for_object_id_2 = {
    'sliding-friction': 0.52, # slight decrease in friction to reduce
        error
    'armature': 0.62,    # slight decrease in armature
    'stiffness': 1.23,   # slight decrease in stiffness
    'mass': 10.0,        # mass remains unchanged
    'damping': 8.1       # slight decrease in damping to reduce discrepancy
}
sim.create_object(object_id=2, object_name='martini_glass',
    object_location=('row_1', 'column_2'), object_color='orange',
    object_physics=physical_parameters_for_object_id_2)

physical_parameters_for_object_id_3 = {
    'sliding-friction': 0.71, # slight decrease in friction for close
        alignment
    'armature': 1.04,    # slight decrease in armature
    'stiffness': 0.94,   # slight decrease in stiffness
    'mass': 4.0,         # mass remains unchanged
    'damping': 8.0       # slight decrease in damping for close alignment
}
sim.create_object(object_id=3, object_name='wine_glass', object_location
    =('row_1', 'column_1'), object_color='orange', object_physics=
    physical_parameters_for_object_id_3)

sim.create_scene()
```

```
1998   sim_out=sim.run_simulation()
1999   del sim
2000   ```
2001
2002   Make sure to run this code through your simulation environment to verify
2003       the trajectory errors. The adjustments are subtle yet precise to
2004       better align with the required trajectories in 'problem_trajectories.
           txt'.
2005
2006   ========
2007
2008   To refine the parameters and reduce the trajectory prediction error, I
2009       will make gradual and strategic adjustments while making sure not to
2010       alter beyond the first significant digit. Below are the physical
2011       parameters for each object, followed by the refined code that will be
            used to generate the trajectories.
2012
2013   ```python
2014   # refined_code.py
2015   sim = SIMULATOR_MODEL()
2016   sim.create_pusher('3.0 3.0 0.05')
2017
2018   physical_parameters_for_object_id_tray = {
2019       'sliding-friction': 0.1,
2020       'armature': 0.1,
2021       'stiffness': 0.0,
2022       'mass': 0.5,
2023       'damping': 20
       }
2024   sim.create_tray(object_physics=physical_parameters_for_object_id_tray)
2025   physical_parameters_for_object_id_1 = {
2026       'sliding-friction': 0.21, # slight decrease in friction for finer
2027           adjustment
2028       'armature': 0.42,    # slight decrease in armature
2029       'stiffness': 0.54,   # slight adjustment in stiffness
2030       'mass': 20.0,        # mass remains unchanged
2031       'damping': 7.3       # slight decrease in damping
       }
2032   sim.create_object(object_id=1, object_name='bottle', object_location=('
2033       row_1', 'column_3'), object_color='orange', object_physics=
           physical_parameters_for_object_id_1)
2034
2035   physical_parameters_for_object_id_2 = {
2036       'sliding-friction': 0.51, # slight decrease in friction to further
            reduce error
2037       'armature': 0.61,    # slight decrease in armature
2038       'stiffness': 1.22,   # slight adjustment in stiffness
2039       'mass': 10.0,        # mass remains unchanged
2040       'damping': 8.0       # slight decrease in damping
2041   }
2042   sim.create_object(object_id=2, object_name='martini_glass',
2043       object_location=('row_1', 'column_2'), object_color='orange',
           object_physics=physical_parameters_for_object_id_2)
2044
2045   physical_parameters_for_object_id_3 = {
2046       'sliding-friction': 0.73, # slight increase in friction for better
2047           alignment
2048       'armature': 1.03,    # slight decrease in armature
2049       'stiffness': 0.96,   # slight increase in stiffness
2050       'mass': 4.0,         # mass remains unchanged
       'damping': 7.9       # slight decrease in damping
2051   }
```

```
2052   sim.create_object(object_id=3, object_name='wine_glass', object_location
2053       =('row_1', 'column_1'), object_color='orange', object_physics=
2054       physical_parameters_for_object_id_3)
2055
2056   # Create and run the scene
2057   sim.create_scene()
2058   sim_out = sim.run_simulation()
2059   del sim
       ```
2060
2061   The above adjustments are strategic, aiming for finer tuning of the
2062       sliding-friction, armature, stiffness, and damping parameters to get
2063       results closer to the desired trajectories. By changing only up to
2064       the first significant digit, these increments offer more precise
2065       control over the movements of the objects in the simulation.
2066
2067   You can run this refined code in your simulation environment to compare
2068       the actual trajectory errors and see if the refinements have reduced
2069       the prediction errors further.

       ========

2070   To refine the parameters further, let's focus on making smaller and yet
2071       significant adjustments based on the previous attempts and their
2072       respective errors. We will perform slight tweaking to the parameters
2073       to minimize the trajectory errors further.
2074
2075   Given that the previous attempts were gradually reducing errors, let's
2076       make slight fine-tuning adjustments to improve the accuracy even
2077       further:
2078
2079   - Slightly tweaking the sliding-friction for each object to balance the
           motion.
2080   - Small adjustments to armature, stiffness, and damping to achieve
2081       optimal results.
2082   - Keeping all changes within the first significant digit, as instructed.
2083
2084   Here is the adjusted version of the code that aims to further refine the
           object parameters:
2085
2086   ```python
2087   sim = SIMULATOR_MODEL()
2088   sim.create_pusher('3.0 3.0 0.05')
2089   physical_parameters_for_object_id_tray = {
2090       'sliding-friction': 0.1,
2091       'armature': 0.1,
2092       'stiffness': 0.0,
2093       'mass': 0.5,
2094       'damping': 20
       }
2095   sim.create_tray(object_physics=physical_parameters_for_object_id_tray)
2096
2097   physical_parameters_for_object_id_1 = {
2098       'sliding-friction': 0.22, # slight increase in friction
2099       'armature': 0.41,    # slight decrease in armature
2100       'stiffness': 0.52,   # slight adjustment in stiffness
2101       'mass': 20.0,        # mass remains unchanged
2102       'damping': 7.1       # slight decrease in damping
       }
2103   sim.create_object(object_id=1, object_name='bottle', object_location=('
2104       row_1', 'column_3'), object_color='orange', object_physics=
2105       physical_parameters_for_object_id_1)

       physical_parameters_for_object_id_2 = {
```

```
'sliding-friction': 0.53, # slight decrease in friction
'armature': 0.64,    # slight increase in armature
'stiffness': 1.24,   # slight adjustment in stiffness
'mass': 10.0,        # mass remains unchanged
'damping': 8.1       # slight decrease in damping
}
sim.create_object(object_id=2, object_name='martini_glass',
    object_location=('row_1', 'column_2'), object_color='orange',
    object_physics=physical_parameters_for_object_id_2)

physical_parameters_for_object_id_3 = {
'sliding-friction': 0.72, # slight decrease in friction
'armature': 1.02,    # slight decrease in armature
'stiffness': 0.93,   # slight decrease in stiffness
'mass': 4.0,         # mass remains unchanged
'damping': 7.8       # slight decrease in damping
}
sim.create_object(object_id=3, object_name='wine_glass', object_location
    =('row_1', 'column_1'), object_color='orange', object_physics=
    physical_parameters_for_object_id_3)

sim.create_scene()
sim_out = sim.run_simulation()

del sim
```

This adjustment attempts to balance the parameters more precisely by
    making small changes while focusing on the physical behavior that
    impacts the simulation. The goal remains to accurately reproduce the
    trajectories provided in 'problem_trajectories.txt'.

========

To further refine the simulation parameters and achieve lower trajectory
    prediction errors, we will use small adjustments based on the
    previous attempts. Here is the updated code based on a careful
    analysis of the prior errors and corresponding changes:

```python
sim = SIMULATOR_MODEL()
sim.create_pusher('3.0 3.0 0.05')

physical_parameters_for_object_id_tray = {
'sliding-friction': 0.1,
'armature': 0.1,
'stiffness': 0.0,
'mass': 0.5,
'damping': 20
}
sim.create_tray(object_physics = physical_parameters_for_object_id_tray)

physical_parameters_for_object_id_1 = {
'sliding-friction': 0.22, # increased slightly
'armature': 0.40,    # decreased slightly
'stiffness': 0.53,   # adjusted slightly
'mass': 20.0,        # mass remains unchanged
'damping': 7.2       # decreased slightly
}
sim.create_object(object_id=1, object_name='bottle', object_location=('
    row_1', 'column_3'), object_color='orange', object_physics=
    physical_parameters_for_object_id_1)

physical_parameters_for_object_id_2 = {
'sliding-friction': 0.54, # decreased slightly
```

```
      'armature': 0.63,    # adjusted slightly
      'stiffness': 1.23,   # adjusted slightly
      'mass': 10.0,        # mass remains unchanged
      'damping': 8.2       # adjusted slightly
}
sim.create_object(object_id=2, object_name='martini_glass',
      object_location=('row_1', 'column_2'), object_color='orange',
      object_physics=physical_parameters_for_object_id_2)

physical_parameters_for_object_id_3 = {
      'sliding-friction': 0.73, # increased slightly
      'armature': 1.04,    # adjusted slightly
      'stiffness': 0.94,   # adjusted slightly
      'mass': 4.0,         # mass remains unchanged
      'damping': 8.0       # adjusted slightly
}
sim.create_object(object_id=3, object_name='wine_glass', object_location
      =('row_1', 'column_1'), object_color='orange', object_physics=
      physical_parameters_for_object_id_3)

sim.create_scene()
sim_out=sim.run_simulation()
del sim
```

Filename: `example_code_refined.py`

Explanation of Changes:
1. **Bottle (object_id_1)**:
   - Increased sliding friction from 0.21 to 0.22 to further adjust
      movement.
   - Decreased armature from 0.42 to 0.40 for a finer balance.
   - Adjusted stiffness slightly from 0.54 to 0.53.
   - Decreased damping from 7.3 to 7.2 for finer tuning.

2. **Martini Glass (object_id_2)**:
   - Decreased sliding friction to 0.54 for slight adjustment.
   - Increased armature slightly from 0.61 to 0.63.
   - Adjusted stiffness slightly to 1.23.
   - Decreased damping to 8.2 for finer tuning.

3. **Wine Glass (object_id_3)**:
   - Increased sliding friction slightly to 0.73 for better alignment.
   - Adjusted armature to 1.04 for balance.
   - Adjusted stiffness to 0.94.
   - Kept damping at 8.0 for slight adjustment.

This should result in a better alignment of the predicted trajectories
   with the ones given in `problem_trajectories.txt`.

========

To refine the physical parameters further and reduce the trajectory
   prediction errors, let's analyze the errors from previous attempts
   and try to make incremental adjustments.

Given the previous errors and the adjustments made, it is noticed that
   attempts have incrementally changed both the sliding-friction and the
   armature values slightly, while trying to balance out the
   trajectories of the bottle, martini_glass, and wine_glass.

Let's make further fine-tuned adjustments to these values in an effort to
   bring the prediction errors closer to the intended trajectories.

```python
```

```
2214    sim = SIMULATOR_MODEL()
2215    sim.create_pusher('3.0 3.0 0.05')
2216    physical_parameters_for_object_id_tray = {
2217            'sliding-friction': 0.1,
2218            'armature': 0.1,
2219            'stiffness': 0.0,
2220            'mass': 0.5,
2221            'damping': 20
        }
2222
2223    sim.create_tray(object_physics = physical_parameters_for_object_id_tray)
2224
2225    physical_parameters_for_object_id_1 = {
2226            'sliding-friction': 0.23, # slight increase
2227            'armature': 0.39,   # slight decrease
2228            'stiffness': 0.53,  # slight adjustment
2229            'mass': 20.0,       # mass remains unchanged
            'damping': 7.2      # slight adjustment
2230        }
2231    sim.create_object(object_id=1, object_name='bottle', object_location=('
2232        row_1', 'column_3'), object_color='orange', object_physics=
        physical_parameters_for_object_id_1)
2233
2234    physical_parameters_for_object_id_2 = {
2235            'sliding-friction': 0.53, # decrease slightly
2236            'armature': 0.66,   # increase slightly
2237            'stiffness': 1.23,  # adjusted slightly
2238            'mass': 10.0,       # mass remains unchanged
            'damping': 8.2      # slight adjustment
2239        }
2240    sim.create_object(object_id=2, object_name='martini_glass',
2241        object_location=('row_1', 'column_2'), object_color='orange',
        object_physics=physical_parameters_for_object_id_2)
2242
2243    physical_parameters_for_object_id_3 = {
2244            'sliding-friction': 0.72, # slight decrease
2245            'armature': 1.05,   # slight increase
2246            'stiffness': 0.96,  # slight adjustment
2247            'mass': 4.0,        # mass remains unchanged
            'damping': 7.9      # slight adjustment
2248        }
2249    sim.create_object(object_id=3, object_name='wine_glass', object_location
2250        =('row_1', 'column_1'), object_color='orange', object_physics=
        physical_parameters_for_object_id_3)
2251
2252    sim.create_scene()
2253    sim_out=sim.run_simulation()
2254    del sim
        ```
2255
2256    This updated code aims to fine-tune the physical parameters (sliding-
2257        friction, armature, stiffness, and damping) based on the errors
2258        observed in previous attempts. The adjustments aim for minimal yet
2259        effective changes to reduce the trajectory prediction errors. You
2260        should use this code and verify the resulting trajectory errors to
        see if they are minimized.
2261
2262    ========
2263
2264    Below is a refined code structure to simulate the scene, taking into
2265        consideration the errors from previous attempts and aiming to
2266        minimize the overall trajectory prediction error. The goal is to fine
2267        -tune the physical parameters for each object to closely match the
        provided `problem_trajectories.txt`.
```

```python
# refined_code.py

sim = SIMULATOR_MODEL()
sim.create_pusher('3.0 3.0 0.05')

# Tray parameters remain unchanged as per the instructions
physical_parameters_for_tray = {
    'sliding-friction': 0.1,
    'armature': 0.1,
    'stiffness': 0.0,
    'mass': 0.5,
    'damping': 20
}
sim.create_tray(object_physics=physical_parameters_for_tray)

# New calculated parameters for bottle
physical_parameters_for_object_id_1 = {
    'sliding-friction': 0.22, # slight decrease in friction for finer
        adjustment
    'armature': 0.42,     # slight adjustment in armature
    'stiffness': 0.54,    # slight increase in stiffness
    'mass': 20.0,         # mass remains unchanged
    'damping': 7.3        # slight decrease in damping
}
sim.create_object(object_id=1, object_name='bottle', object_location=('
    row_1', 'column_3'), object_color='orange', object_physics=
    physical_parameters_for_object_id_1)

# New calculated parameters for martini_glass
physical_parameters_for_object_id_2 = {
    'sliding-friction': 0.51, # slight decrease in friction for reduction
        in error
    'armature': 0.61,     # adjusted slightly
    'stiffness': 1.21,    # adjusted slightly
    'mass': 10.0,         # mass remains unchanged
    'damping': 8.0        # adjusted slightly
}
sim.create_object(object_id=2, object_name='martini_glass',
    object_location=('row_1', 'column_2'), object_color='orange',
    object_physics=physical_parameters_for_object_id_2)

# New calculated parameters for wine_glass
physical_parameters_for_object_id_3 = {
    'sliding-friction': 0.73, # adjusted for better alignment
    'armature': 1.03,     # slight adjustment
    'stiffness': 0.94,    # slight increase in stiffness
    'mass': 4.0,          # mass remains unchanged
    'damping': 7.9        # slight adjustment
}
sim.create_object(object_id=3, object_name='wine_glass', object_location
    =('row_1', 'column_1'), object_color='orange', object_physics=
    physical_parameters_for_object_id_3)

sim.create_scene()
sim_out = sim.run_simulation()
del sim
```

### Key Adjustments Made:
- **Bottle (ID 1)**:
  - Slight decrease in sliding friction to 0.22.
  - Adjusted armature to 0.42.
  - Increased stiffness to 0.54.
  - Decreased damping to 7.3.

```
- **Martini Glass (ID 2)**:
  - Decreased sliding friction to 0.51.
  - Adjusted armature to 0.61.
  - Stiffness set to 1.21.
  - Damping decreased to 8.0.

- **Wine Glass (ID 3)**:
  - Sliding friction adjusted up to 0.73.
  - Armature adjusted to 1.03.
  - Stiffness increased to 0.94.
  - Damping adjusted to 7.9.

These adjustments target reducing the overall prediction error based on
    the last feedback received. Running this simulation with these
    parameters should yield predictions closer to the proposed `
    problem_trajectories.txt`.

========

To further refine the physical parameters and reduce the trajectory
    prediction error, let's carefully adjust each parameter based on the
    insights from previous attempts. It's crucial to keep the adjustments
     minimal and focus on specific discrepancies between the predicted
    and actual trajectories to fine-tune the parameters more effectively.

Here is the Python code for `example_code_1.py` with updated physical
    parameters:

```python
sim = SIMULATOR_MODEL()
sim.create_pusher('3.0 3.0 0.05')

# Physics parameters for the tray (unchanged)
physical_parameters_for_object_id_tray = {
    'sliding-friction': 0.1,
    'armature': 0.1,
    'stiffness': 0.0,
    'mass': 0.5,
    'damping': 20
}
sim.create_tray(object_physics = physical_parameters_for_object_id_tray)

# Adjusted physical parameters for the bottle
physical_parameters_for_object_id_1 = {
    'sliding-friction': 0.21, # slight decrease
    'armature': 0.43,    # slight adjustment
    'stiffness': 0.53,   # slight adjustment
    'mass': 20.0,        # mass remains unchanged
    'damping': 7.2       # slight adjustment
}
sim.create_object(object_id=1, object_name='bottle', object_location=('
    row_1', 'column_3'), object_color='orange', object_physics=
    physical_parameters_for_object_id_1)

# Adjusted physical parameters for the martini_glass
physical_parameters_for_object_id_2 = {
    'sliding-friction': 0.52, # slight adjustment
    'armature': 0.64,    # slight adjustment
    'stiffness': 1.21,   # slight adjustment
    'mass': 10.0,        # mass remains unchanged
    'damping': 8.1       # slight adjustment
}
```

```
sim.create_object(object_id=2, object_name='martini_glass',
    object_location=('row_1', 'column_2'), object_color='orange',
    object_physics=physical_parameters_for_object_id_2)

# Adjusted physical parameters for the wine_glass
physical_parameters_for_object_id_3 = {
    'sliding-friction': 0.73, # slight adjustment
    'armature': 1.03,     # slight adjustment
    'stiffness': 0.93,    # slight adjustment
    'mass': 4.0,          # mass remains unchanged
    'damping': 7.8        # slight adjustment
}
sim.create_object(object_id=3, object_name='wine_glass', object_location
    =('row_1', 'column_1'), object_color='orange', object_physics=
    physical_parameters_for_object_id_3)

sim.create_scene()
sim_out = sim.run_simulation()
del sim
```

Explanation of refinement:
1. **Sliding Friction**: Slightly decreased for the bottle and
   martini_glass based on their sensitivity.
2. **Armature**: Tweaked by a small degree for each object considering
   their resistance to movement.
3. **Stiffness**: Adjusted to fine-tune the object's resistance to
   deformation.
4. **Damping**: Decreased slightly to improve the motion accuracy of all
   objects.

This fine-tuned approach aims to reduce the prediction error while
    ensuring minimal changes to the parameters, focusing on enhancing the
     alignment of the trajectories with the given problem_trajectories.
    txt. Adjust only to their first significant digit to conform to the
    constraints.

========

Given the previous attempts and their errors, it seems that the
    adjustments have incrementally improved but not drastically reduced
    the errors. To reduce errors further, we should continue fine-tuning
    the parameters while considering the inherent dynamics of each object
    . Here is another attempt with refined parameters:

```python
# example_code_1.py

sim = SIMULATOR_MODEL()
sim.create_pusher('3.0 3.0 0.05')
physical_parameters_for_object_id_tray = {
        'sliding-friction': 0.1,
        'armature': 0.1,
        'stiffness': 0.0,
        'mass': 0.5,
        'damping': 20
    }
sim.create_tray(object_physics = physical_parameters_for_object_id_tray)
physical_parameters_for_object_id_1 = {
        'sliding-friction': 0.21, # slight decrease for finer control
        'armature': 0.41,    # adjusted armature
        'stiffness': 0.55,   # slight increase in stiffness
        'mass': 20.0,        # mass remains unchanged
        'damping': 7.0       # slight decrease in damping
    }
```

```
2430  sim.create_object(object_id=1, object_name='bottle', object_location=('
2431      row_1', 'column_3'), object_color='orange', object_physics=
2432      physical_parameters_for_object_id_1)
2433
2434  physical_parameters_for_object_id_2 = {
2435          'sliding-friction': 0.52, # adjusted friction
2436          'armature': 0.63,    # slight adjustment in armature
2437          'stiffness': 1.22,   # slight adjustment in stiffness
2438          'mass': 10.0,        # mass remains unchanged
2439          'damping': 8.0       # slight increase in damping
         }
2440  sim.create_object(object_id=2, object_name='martini_glass',
2441      object_location=('row_1', 'column_2'), object_color='orange',
2442      object_physics=physical_parameters_for_object_id_2)
2443
2444  physical_parameters_for_object_id_3 = {
2445          'sliding-friction': 0.72, # slight adjustment
2446          'armature': 1.02,    # slight decrease
2447          'stiffness': 0.94,   # slight adjustment in stiffness
2448          'mass': 4.0,         # mass remains unchanged
2449          'damping': 7.9       # slight adjustment
         }
2450  sim.create_object(object_id=3, object_name='wine_glass', object_location
2451      =('row_1', 'column_1'), object_color='orange', object_physics=
2452      physical_parameters_for_object_id_3)
2453
2454  sim.create_scene()
      sim_out = sim.run_simulation()
      del sim
      ```
2455
2456  ### Key Adjustments:
2457  1. **Bottle ('object_id=1'):**
2458    - Reduced sliding friction further to improve response.
2459    - Increased stiffness for better stability.
2460    - Slightly reduced damping to allow more movement.
2461
2462  2. **Martini Glass ('object_id=2'):**
2463    - Adjusted sliding friction and armature for more controlled dynamics.
2464    - Slightly increased damping to prevent overshooting.
2465    - Marginal adjustments in stiffness.
2466
2467  3. **Wine Glass ('object_id=3'):**
2468    - Slightly adjusted sliding friction and armature for finer tuning.
      - Minor adjustments in stiffness for better alignment.
      - Damping adjusted for controlled oscillations.
2469
2470  Run this updated code in your simulator and check if the error is further
2471      reduced. It's crucial to analyze the trajectories post-simulation to
2472      ensure each parameter adjustment is aiding in convergence toward the
      desired trajectories.
2473
2474  ========
2475
2476  To optimize the code further and reduce the trajectory error, I'll base
2477      my changes on the closest attempts with slight adjustments to each
      parameter to improve the alignment with 'problem_trajectories.txt'.
2478
2479  Let's try small incremental adjustments of the successful parameters from
2480      GPT Attempt #11:
2481
2482  ```python
      # example_code_optimized.py
2483
      sim = SIMULATOR_MODEL()
```

```
2484    sim.create_pusher('3.0 3.0 0.05')
2485
2486    # Tray parameters remain unchanged
2487    physical_parameters_for_object_id_tray = {
2488        'sliding-friction': 0.1,
2489        'armature': 0.1,
2490        'stiffness': 0.0,
2491        'mass': 0.5,
2492        'damping': 20
        }
2493    sim.create_tray(object_physics = physical_parameters_for_object_id_tray)
2494
2495    # Optimized parameters based on closest previous attempt with slight
            adjustments
2496    physical_parameters_for_object_id_1 = { # bottle
2497        'sliding-friction': 0.22, # slight increase for better alignment
2498        'armature': 0.42,    # slight adjustment
2499        'stiffness': 0.54,   # slight adjustment
2500        'mass': 20.0,        # mass remains unchanged
2501        'damping': 7.1       # slight adjustment
        }
2502
2503    physical_parameters_for_object_id_2 = { # martini_glass
2504        'sliding-friction': 0.53, # slight adjustment
2505        'armature': 0.64,    # slight adjustment
2506        'stiffness': 1.23,   # slight adjustment
2507        'mass': 10.0,        # mass remains unchanged
2508        'damping': 8.2       # slight adjustment
        }
2509
2510    physical_parameters_for_object_id_3 = { # wine_glass
2511        'sliding-friction': 0.73, # slight adjustment
2512        'armature': 1.03,    # slight adjustment
2513        'stiffness': 0.95,   # slight adjustment
2514        'mass': 4.0,         # mass remains unchanged
        'damping': 7.8       # slight adjustment
        }
2515
2516    sim.create_object(object_id=1, object_name='bottle', object_location=('
2517        row_1', 'column_3'), object_color='orange', object_physics=
2518        physical_parameters_for_object_id_1)
2519    sim.create_object(object_id=2, object_name='martini_glass',
2520        object_location=('row_1', 'column_2'), object_color='orange',
2521        object_physics=physical_parameters_for_object_id_2)
2522    sim.create_object(object_id=3, object_name='wine_glass', object_location
2523        =('row_1', 'column_1'), object_color='orange', object_physics=
2524        physical_parameters_for_object_id_3)

    sim.create_scene()
2525    sim_out = sim.run_simulation()
2526    del sim
2527
2528    ```
2529
2530    In this adjustment, I've increased or decreased parameters slightly
2531        within the first significant digit and maintained the structure of
2532        the initial code. This approach should bring the predicted
2533        trajectories closer in alignment with the ones in '
            problem_trajectories.txt'.
2534
2535    ========
2536
2537    Given the existing attempts and their respective trajectory prediction
        errors, let's incrementally adjust the physical parameters to better
        align with the desired trajectories in 'problem_trajectories.txt'. We
```

'll focus on making small adjustments to the individual parameters to minimize the error.

### Revised Code

```python
sim = SIMULATOR_MODEL()
sim.create_pusher('3.0 3.0 0.05')

physical_parameters_for_object_id_tray = {
    'sliding-friction': 0.1,
    'armature': 0.1,
    'stiffness': 0.0,
    'mass': 0.5,
    'damping': 20
}
sim.create_tray(object_physics=physical_parameters_for_object_id_tray)

physical_parameters_for_object_id_1 = {
    'sliding-friction': 0.21, # decreased friction
    'armature': 0.41,    # slightly decreased armature
    'stiffness': 0.54,   # slight increase in stiffness
    'mass': 20.0,        # mass remains unchanged
    'damping': 7.2       # slightly decreased damping
}
sim.create_object(object_id=1, object_name='bottle', object_location=(
    'row_1', 'column_3'), object_color='orange', object_physics=
    physical_parameters_for_object_id_1)

physical_parameters_for_object_id_2 = {
    'sliding-friction': 0.52, # slightly decreased friction
    'armature': 0.62,    # slightly decreased armature
    'stiffness': 1.24,   # slight increase in stiffness
    'mass': 10.0,        # mass remains unchanged
    'damping': 8.0       # slight increase in damping
}
sim.create_object(object_id=2, object_name='martini_glass',
    object_location=('row_1', 'column_2'), object_color='orange',
    object_physics=physical_parameters_for_object_id_2)

physical_parameters_for_object_id_3 = {
    'sliding-friction': 0.73, # slight increase in friction
    'armature': 1.02,    # slightly decreased armature
    'stiffness': 0.95,   # slight increase in stiffness
    'mass': 4.0,         # mass remains unchanged
    'damping': 7.9       # slight increase in damping
}
sim.create_object(object_id=3, object_name='wine_glass', object_location
    =('row_1', 'column_1'), object_color='orange', object_physics=
    physical_parameters_for_object_id_3)

sim.create_scene()
sim_out=sim.run_simulation()
del sim
```

The changes applied here are small and precise:

1. For the 'bottle':
   - Minor decrease in 'sliding-friction' to possibly tune finer control over motion.
   - Slight decrease in 'armature' for more conservative moment inertia.
   - Small increase in 'stiffness' to ensure some adjustment in impact response.
   - Slight decrease in 'damping' to adjust energy dissipation.

```
2. For the 'martini_glass':
   - Slight decrease in 'sliding-friction' to reduce resistance.
   - Slight decrease in 'armature' to have fewer moments of inertia.
   - Small increase in 'stiffness' to adjust the rigidity.
   - Slight increase in 'damping' to tune the damping effect.

3. For the 'wine_glass':
   - Slight increase in 'sliding-friction' to increase motion resistance
      towards alignment.
   - Slight decrease in 'armature' for finer inertia control.
   - Small increase in 'stiffness' for more precise impact response.
   - Slight increase in 'damping' for tuned energy dissipation.

By maintaining the careful balance between these suggested adjustments,
   we are likely to achieve a closer match to the desired trajectories
   in 'problem_trajectories.txt'. The actual prediction should be
   evaluated by running this updated code in the provided simulator and
   examining the resulting trajectory errors.

========

Based on the trajectory prediction errors from previous attempts, it
   appears that the parameters for the 'bottle' are quite close to
   accurate, whereas the 'martini_glass' exhibits higher errors which
   need to be reduced. The 'wine_glass' parameters seem to be nearly
   accurate.

To further reduce the trajectory prediction error, we will optimize the
   parameters while maintaining significant digits:

1. Slightly adjust the 'sliding-friction' and 'damping' for all objects.
2. Fine-tune 'armature' and 'stiffness' precisely for each object to
   better align with the desired trajectories.

Here is a suggested set of parameters for another attempt:

```python
sim = SIMULATOR_MODEL()
sim.create_pusher('3.0 3.0 0.05')
physical_parameters_for_object_id_tray = {
        'sliding-friction': 0.1,
        'armature': 0.1,
        'stiffness': 0.0,
        'mass': 0.5,
        'damping': 20
    }
sim.create_tray(object_physics = physical_parameters_for_object_id_tray)

physical_parameters_for_object_id_1 = {
    'sliding-friction': 0.21, # slight adjustment in friction
    'armature': 0.42,   # slight decrease in armature
    'stiffness': 0.53,  # slight decrease in stiffness
    'mass': 20.0,       # mass remains unchanged
    'damping': 7.3      # slight decrease in damping
}

physical_parameters_for_object_id_2 = {
    'sliding-friction': 0.54, # slight adjustment to decrease friction
    'armature': 0.63,   # slight adjustment
    'stiffness': 1.24,  # slight adjustment
    'mass': 10.0,       # mass remains unchanged
    'damping': 8.1      # slight adjustment
}
```

```
2646    physical_parameters_for_object_id_3 = {
2647        'sliding-friction': 0.74, # slight increase to better align friction
2648        'armature': 1.01,    # slight adjustment
2649        'stiffness': 0.94,   # slight decrease in stiffness
2650        'mass': 4.0,         # mass remains unchanged
2651        'damping': 7.8       # slight decrease in damping
2652    }
2653
2654    sim.create_object(object_id=1, object_name='bottle', object_location=('
2655        row_1', 'column_3'), object_color='orange', object_physics=
2656        physical_parameters_for_object_id_1)
2657    sim.create_object(object_id=2, object_name='martini_glass',
2658        object_location=('row_1', 'column_2'), object_color='orange',
2659        object_physics=physical_parameters_for_object_id_2)
2660    sim.create_object(object_id=3, object_name='wine_glass', object_location
2661        =('row_1', 'column_1'), object_color='orange', object_physics=
2662        physical_parameters_for_object_id_3)
2663
2664    sim.create_scene()
2665    sim_out=sim.run_simulation()
2666    del sim
2667    ```
2668
2669    This adjustment continues to follow the structure of the example Python
2670        code, and the changes made to the parameters are conservative to
2671        prevent large deviations while aiming to reduce the overall
2672        trajectory error. This should produce a better alignment of the
2673        trajectories to the given data in 'problem_trajectories.txt'.
2674
2675    ========
2676
2677    To better align the trajectories with those in 'problem_trajectories.txt
2678        ', I will refine the physical parameters for the objects. Based on
2679        previous attempts and trajectory errors, I will focus on minor
2680        adjustments, especially honing in on the slight variations in
2681        friction, armature, stiffness, and damping.
2682
2683    Here's the revised code following the given structure:
2684
2685    ```python
2686    sim = SIMULATOR_MODEL()
2687    sim.create_pusher('3.0 3.0 0.05')
2688    physical_parameters_for_object_id_tray = {
2689            'sliding-friction': 0.1,
2690            'armature': 0.1,
2691            'stiffness': 0.0,
2692            'mass': 0.5,
2693            'damping': 20
2694        }
2695    sim.create_tray(object_physics = physical_parameters_for_object_id_tray)
2696
2697    physical_parameters_for_object_id_1 = { # bottle
2698        'sliding-friction': 0.22, # slight adjustment
2699        'armature': 0.42,    # fine-tune armature
        'stiffness': 0.54,   # fine-tune stiffness
        'mass': 20.0,        # mass remains unchanged
        'damping': 7.1       # minor adjustment
    }
    sim.create_object(object_id=1, object_name='bottle', object_location=('
        row_1', 'column_3'), object_color='orange', object_physics=
        physical_parameters_for_object_id_1)

    physical_parameters_for_object_id_2 = { # martini_glass
        'sliding-friction': 0.52, # slight adjustment
        'armature': 0.63,    # fine-tune armature
```

```
      'stiffness': 1.23,   # fine-tune stiffness
      'mass': 10.0,        # mass remains unchanged
      'damping': 8.1       # minor adjustment
   }
   sim.create_object(object_id=2, object_name='martini_glass',
       object_location=('row_1', 'column_2'), object_color='orange',
       object_physics=physical_parameters_for_object_id_2)

   physical_parameters_for_object_id_3 = { # wine_glass
      'sliding-friction': 0.73, # slight adjustment
      'armature': 1.04,    # fine-tune armature
      'stiffness': 0.95,   # fine-tune stiffness
      'mass': 4.0,         # mass remains unchanged
      'damping': 7.8       # minor adjustment
   }
   sim.create_object(object_id=3, object_name='wine_glass', object_location
       =('row_1', 'column_1'), object_color='orange', object_physics=
       physical_parameters_for_object_id_3)

   sim.create_scene()
   sim_out = sim.run_simulation()
   del sim
   ```

   Explanation of changes:
   - For the bottle, martini_glass, and wine_glass, slight adjustments are
       made to the sliding-friction, armature, and damping parameters in
       very small increments to better mimic the trajectories.
   - The fine-tuning is based on a better alignment seen in the errors of
       previous attempts.

   Please run this refined code in your simulator to evaluate the accuracy
       of the trajectories produced.

   ========

   Based on the previous attempts and their resulting errors, it appears
       that fine-tuning the physical parameters can potentially bring the
       trajectory prediction error closer to the desired trajectories. Given
        that previous attempts were close but required further refinement,
       let's slightly adjust the parameters once more to seek better
       alignment.

   Here is the refined attempt:

   ```python
   sim = SIMULATOR_MODEL()
   sim.create_pusher('3.0 3.0 0.05')

   physical_parameters_for_object_id_tray = {
      'sliding-friction': 0.1,
      'armature': 0.1,
      'stiffness': 0.0,
      'mass': 0.5,
      'damping': 20
   }
   sim.create_tray(object_physics=physical_parameters_for_object_id_tray)

   physical_parameters_for_object_id_1 = { # bottle
      'sliding-friction': 0.21, # slight adjustment
      'armature': 0.40,    # fine-tune armature
      'stiffness': 0.54,   # fine-tune stiffness
      'mass': 20.0,        # mass remains unchanged
      'damping': 7.2       # minor adjustment
   }
```

```python
physical_parameters_for_object_id_2 = { # martini_glass
    'sliding-friction': 0.52, # slight adjustment
    'armature': 0.64,    # fine-tune armature
    'stiffness': 1.23,   # fine-tune stiffness
    'mass': 10.0,        # mass remains unchanged
    'damping': 8.1       # minor adjustment
}

physical_parameters_for_object_id_3 = { # wine_glass
    'sliding-friction': 0.73, # slight adjustment
    'armature': 1.04,    # fine-tune armature
    'stiffness': 0.94,   # fine-tune stiffness
    'mass': 4.0,         # mass remains unchanged
    'damping': 7.9       # minor adjustment
}

sim.create_object(object_id=1, object_name='bottle', object_location=('
    row_1', 'column_3'), object_color='orange', object_physics=
    physical_parameters_for_object_id_1)
sim.create_object(object_id=2, object_name='martini_glass',
    object_location=('row_1', 'column_2'), object_color='orange',
    object_physics=physical_parameters_for_object_id_2)
sim.create_object(object_id=3, object_name='wine_glass', object_location
    =('row_1', 'column_1'), object_color='orange', object_physics=
    physical_parameters_for_object_id_3)

sim.create_scene()
sim_out = sim.run_simulation()
del sim
```

This code has been adjusted based on the trend observed in previous
    attempts. Small refinements are made to the parameters to attempt a
    better match to the target trajectories:

1. Slight adjustments in sliding friction, armature, stiffness, and
    damping for each object.
2. Ensured to keep all changes within the realm of minor adjustments,
    considering past errors.

These changes are aimed to further fine-tune the physics parameters
    closer to the desired trajectories. This should help in reducing the
    trajectory prediction error further.

========

It seems like all the example attempts produced code that follows the
    original problem's constraints and shows some progress in reducing
    the trajectory error. However, there is still room for improvement in
     reducing the trajectory prediction error further. Let's adjust the
    parameters again slightly, being mindful of not changing them beyond
    the first significant digit, and pick a set that should help us
    achieve better accuracy:

Here's an updated set of parameters for the objects:

```python
sim = SIMULATOR_MODEL()
sim.create_pusher('3.0 3.0 0.05')
physical_parameters_for_object_id_tray = {
    'sliding-friction': 0.1,
    'armature': 0.1,
    'stiffness': 0.0,
    'mass': 0.5,
```

```
2808      'damping': 20
2809    }
2810    sim.create_tray(object_physics=physical_parameters_for_object_id_tray)
2811
2812    physical_parameters_for_object_id_1 = { # bottle
2813       'sliding-friction': 0.21, # slight adjustment in friction
2814       'armature': 0.40,    # slight decrease in armature
2815       'stiffness': 0.55,   # slight increase in stiffness
2816       'mass': 20.0,        # mass remains unchanged
2817       'damping': 7.2       # slight decrease in damping
        }
2818
2819    physical_parameters_for_object_id_2 = { # martini_glass
2820       'sliding-friction': 0.53, # slight adjustment to decrease friction
2821       'armature': 0.63,    # slight adjustment in armature
2822       'stiffness': 1.22,   # slight increase in stiffness
2823       'mass': 10.0,        # mass remains unchanged
        'damping': 8.2       # slight decrease in damping
2824    }
2825
2826    physical_parameters_for_object_id_3 = { # wine_glass
2827       'sliding-friction': 0.72, # slight adjustment to increase traction
2828       'armature': 1.04,    # slight decrease
2829       'stiffness': 0.96,   # slight adjustment in stiffness
2830       'mass': 4.0,         # mass remains unchanged
        'damping': 8.0       # slight decrease in damping
2831    }
2832
2833    sim.create_object(object_id=1, object_name='bottle', object_location=('
           row_1', 'column_3'), object_color='orange', object_physics=
2834       physical_parameters_for_object_id_1)
2835    sim.create_object(object_id=2, object_name='martini_glass',
           object_location=('row_1', 'column_2'), object_color='orange',
2836       object_physics=physical_parameters_for_object_id_2)
2837    sim.create_object(object_id=3, object_name='wine_glass', object_location
           =('row_1', 'column_1'), object_color='orange', object_physics=
2838       physical_parameters_for_object_id_3)
2839
2840    sim.create_scene()
2841    sim_out = sim.run_simulation()
2842    del sim
        ```
2843
2844    In this iteration:
2845    - We fine-tuned the 'sliding-friction', 'armature', 'stiffness', and '
2846       damping' values for each object, but ensured they remained within
2847       their first significant digit constraints.
        - We used previous attempts' values as a base to make minute adjustments
2848       that could better align the trajectories with the given problem.
2849
2850    Once this new code set is tested, it should ideally show an improvement
2851       in error reduction. Please run this updated code using your simulator
2852       environment to verify the trajectory errors. If further adjustment
2853       is needed, making iterative small changes around these values can
        help further fine-tune and minimize errors.
2854
2855    ========
2856
2857    To refine your code based on the trajectory prediction error results and
2858       better align with the trajectories given in 'problem_trajectories.txt
2859       ', we need to follow a reasonable approach to adjust the physical
        parameters for each object slightly further. Below is a refined
2860       attempt based on the provided feedback:
2861
        ### Refined Physical Parameters Code:
```

```python
# example_code_1.py

sim = SIMULATOR_MODEL()
sim.create_pusher('3.0 3.0 0.05')
physical_parameters_for_object_id_tray = {
        'sliding-friction': 0.1,
        'armature': 0.1,
        'stiffness': 0.0,
        'mass': 0.5,
        'damping': 20
    }
sim.create_tray(object_physics = physical_parameters_for_object_id_tray)
physical_parameters_for_object_id_1 = { # bottle
    'sliding-friction': 0.22, # slight increase for better alignment
    'armature': 0.41,    # fine-tune armature
    'stiffness': 0.55,   # fine-tune stiffness a bit more
    'mass': 20.0,        # mass remains unchanged
    'damping': 7.1       # slight adjustment
}
sim.create_object(object_id=1, object_name='bottle', object_location=('
    row_1', 'column_3'), object_color='orange', object_physics=
    physical_parameters_for_object_id_1)

physical_parameters_for_object_id_2 = { # martini_glass
    'sliding-friction': 0.52, # slight decrease for alignment
    'armature': 0.62,    # slight fine-tune
    'stiffness': 1.21,   # slight fine-tune
    'mass': 10.0,        # mass remains unchanged
    'damping': 8.1       # slight fine-tune
}
sim.create_object(object_id=2, object_name='martini_glass',
    object_location=('row_1', 'column_2'), object_color='orange',
    object_physics=physical_parameters_for_object_id_2)

physical_parameters_for_object_id_3 = { # wine_glass
    'sliding-friction': 0.73, # increase slightly for better traction
    'armature': 1.03,    # adjusted slightly
    'stiffness': 0.95,   # fine-tune to keep closer to expected
    'mass': 4.0,         # mass remains unchanged
    'damping': 8.0       # fine-tune
}
sim.create_object(object_id=3, object_name='wine_glass', object_location
    =('row_1', 'column_1'), object_color='orange', object_physics=
    physical_parameters_for_object_id_3)

demonstration_tray_motion_trajectory = [
    (0.0, 0.0, 0.1), (-0.7, -0.7, 0.1), (-1.2, -1.2, 0.1), (-1.6, -1.6,
        0.1),
    (-1.9, -2.0, 0.1), (-2.1, -2.3, 0.1), (-2.3, -2.5, 0.1), (-2.4, -2.6,
        0.1),
    (-2.6, -2.8, 0.1)
]

sim.create_scene()
sim_out = sim.run_simulation()
del sim
```

This refinement involves slight increments and decrements on the physical
    parameters of the objects based on the significant aberrations
    observed in the previous outputs. The adjustments focus on enhancing
    the sliding friction, armature, and damping more accurately towards
    the expected trajectories per the 'problem_trajectories.txt'.

Ensure that you test this code using the same simulator constraints you
    have been using. If further fine-tuning is necessary, keep iterating
    with slight adjustments, focusing on realigning the most anomalous
    trajectory points first.

========

Given the simulation errors and the iterations you provided, it looks
    like adjustments to `sliding-friction`, `armature`, `stiffness`, and
    `damping` were evaluated. The challenge is to find a balance among
    these parameters to minimize the total trajectory error. Below, I've
    made refined adjustments based on the observed errors in previous
    attempts:

```python
sim = SIMULATOR_MODEL()
sim.create_pusher('3.0 3.0 0.05')

physical_parameters_for_object_id_tray = {
        'sliding-friction': 0.1,
        'armature': 0.1,
        'stiffness': 0.0,
        'mass': 0.5,
        'damping': 20
    }
sim.create_tray(object_physics = physical_parameters_for_object_id_tray)

physical_parameters_for_object_id_1 = { # bottle
    'sliding-friction': 0.22, # slight alignment adjustment
    'armature': 0.42,    # fine-tune armature
    'stiffness': 0.54,   # fine-tune stiffness
    'mass': 20.0,        # mass remains unchanged
    'damping': 7.2       # slight fine-tune
}

physical_parameters_for_object_id_2 = { # martini_glass
    'sliding-friction': 0.52, # slight increase
    'armature': 0.63,    # slight adjustment
    'stiffness': 1.22,   # slight fine-tune
    'mass': 10.0,        # mass remains unchanged
    'damping': 8.1       # fine-tune damping
}

physical_parameters_for_object_id_3 = { # wine_glass
    'sliding-friction': 0.73, # increase slightly
    'armature': 1.03,    # slight adjustment
    'stiffness': 0.95,   # fine-tune stiffness
    'mass': 4.0,         # mass remains unchanged
    'damping': 8.0       # fine-tune damping
}

sim.create_object(object_id=1, object_name='bottle', object_location=('
    row_1', 'column_3'), object_color='orange', object_physics=
    physical_parameters_for_object_id_1)
sim.create_object(object_id=2, object_name='martini_glass',
    object_location=('row_1', 'column_2'), object_color='orange',
    object_physics=physical_parameters_for_object_id_2)
sim.create_object(object_id=3, object_name='wine_glass', object_location
    =('row_1', 'column_1'), object_color='orange', object_physics=
    physical_parameters_for_object_id_3)

sim.create_scene()
sim_out = sim.run_simulation()
del sim
```

These refined adjustments should help get closer to the trajectory
    provided in `problem_trajectories.txt`:

- Slightly increased `sliding-friction` and `damping` for all objects to
    match their interaction with the tray more closely.
- Fine-tuned `armature` and `stiffness` to slightly alter responses
    without overdoing adjustments.

Please test this configuration in your simulation environment to check
    for reduced trajectory prediction errors. Fine-tuning these
    configurations iteratively based on errors should help achieve the
    desired result.

========

Thank you for providing the detailed analysis and results. Based on the
    previous attempts, it appears that the adjustments to the sliding-
    friction, armature, stiffness, and damping parameters need further
    fine-tuning to achieve a closer match with the trajectories in '
    problem_trajectories.txt'. I will make slight adjustments to these
    parameters to try to minimize the trajectory prediction error further
    .

**Adjusted Physical Parameters:**

```python
# refined_code.py

sim = SIMULATOR_MODEL()
sim.create_pusher('3.0 3.0 0.05')

physical_parameters_for_object_id_tray = {
    'sliding-friction': 0.1,
    'armature': 0.1,
    'stiffness': 0.0,
    'mass': 0.5,
    'damping': 20
}
sim.create_tray(object_physics=physical_parameters_for_object_id_tray)

physical_parameters_for_object_id_1 = { # bottle
    'sliding-friction': 0.22, # slight adjustment for better alignment
    'armature': 0.41,    # fine-tuning armature
    'stiffness': 0.54,   # fine-tuning stiffness
    'mass': 20.0,        # mass remains unchanged
    'damping': 7.3       # fine-tuning damping
}

physical_parameters_for_object_id_2 = { # martini_glass
    'sliding-friction': 0.52, # fine-tuning sliding-friction
    'armature': 0.63,    # fine-tuning armature
    'stiffness': 1.23,   # fine-tuning stiffness
    'mass': 10.0,        # mass remains unchanged
    'damping': 8.0       # fine-tuning damping
}

physical_parameters_for_object_id_3 = { # wine_glass
    'sliding-friction': 0.72, # fine-tuning sliding-friction
    'armature': 1.04,    # slight adjustment
    'stiffness': 0.94,   # fine-tuning stiffness
    'mass': 4.0,         # mass remains unchanged
    'damping': 8.0       # slight fine-tuning damping
}
```

```
3024    sim.create_object(object_id=1, object_name='bottle', object_location=('
3025        row_1', 'column_3'), object_color='orange', object_physics=
3026        physical_parameters_for_object_id_1)
3027    sim.create_object(object_id=2, object_name='martini_glass',
3028        object_location=('row_1', 'column_2'), object_color='orange',
3029        object_physics=physical_parameters_for_object_id_2)
3030    sim.create_object(object_id=3, object_name='wine_glass', object_location
3031        =('row_1', 'column_1'), object_color='orange', object_physics=
            physical_parameters_for_object_id_3)
3032
3033    sim.create_scene()
3034    sim_out = sim.run_simulation()
        del sim
3035    ```
3036
3037    This refined code makes slight adjustments to the physical parameters,
3038        considering the previous results. The aim is to better align the
3039        simulated trajectories with the problem trajectories given in '
3040        problem_trajectories.txt'.
3041
        Let's run these refined parameters through the simulator and compare the
3042        generated trajectories with the reference ones.
3043
3044    Continue this iterative process, making small adjustments within the
3045        significant digits, to minimize the error further.
3046
        ========
3047
3048    Let's iteratively fine-tune the physical parameters for each object using
3049         the minimal changes approach. As each previous attempt had varied
            errors, I'll aim to adjust the parameters finely to achieve a closer
3050        alignment with the target trajectories.
3051
3052    Here is a refined attempt following the same code structure:
3053
3054    ```python
3055    # Generating refined code based on the provided feedback and previous
            attempts
3056
3057    # Initializing the simulator
3058    sim = SIMULATOR_MODEL()
3059    sim.create_pusher('3.0 3.0 0.05')
3060
3061    # Setting up the tray with the same parameters
        physical_parameters_for_object_id_tray = {
3062            'sliding-friction': 0.1,
3063            'armature': 0.1,
                'stiffness': 0.0,
3064            'mass': 0.5,
                'damping': 20
3065        }
3066    sim.create_tray(object_physics = physical_parameters_for_object_id_tray)
3067
3068    # Setting up physical parameters for each object
3069    physical_parameters_for_object_id_1 = { # bottle, trying finer
            adjustments
3070    'sliding-friction': 0.22,
3071    'armature': 0.41,
3072    'stiffness': 0.54,
3073    'mass': 20.0,
3074    'damping': 7.2
3075    }
3076
3077    physical_parameters_for_object_id_2 = { # martini_glass, slight
            adjustments
```

```
3078        'sliding-friction': 0.53,
3079        'armature': 0.63,
3080        'stiffness': 1.23,
3081        'mass': 10.0,
3082        'damping': 8.0
3083    }
3084
3085    physical_parameters_for_object_id_3 = { # wine_glass, finer tuning
3086        'sliding-friction': 0.73,
3087        'armature': 1.04,
3088        'stiffness': 0.95,
3089        'mass': 4.0,
3090        'damping': 7.9
3091    }
3092
3093    # Creating objects on the tray
3094    sim.create_object(object_id=1, object_name='bottle', object_location=('
            row_1', 'column_3'), object_color='orange', object_physics=
            physical_parameters_for_object_id_1)
3095    sim.create_object(object_id=2, object_name='martini_glass',
            object_location=('row_1', 'column_2'), object_color='orange',
            object_physics=physical_parameters_for_object_id_2)
3096    sim.create_object(object_id=3, object_name='wine_glass', object_location
            =('row_1', 'column_1'), object_color='orange', object_physics=
            physical_parameters_for_object_id_3)
3097
3098    # Creating the scene and running the simulation
3099    sim.create_scene()
3100    sim_out = sim.run_simulation()
3101    del sim
3102    ```
```

The rationale behind these values is as follows:
- For `bottle`, the parameters were slightly fine-tuned to achieve
   balanced damping and appropriate friction for better alignment.
- For `martini_glass`, adjustments were made to sliding friction and
   damping to fine-tune the alignment.
- For `wine_glass`, tuning involved slight adjustments to friction,
   stiffness, and armature to refine the trajectory.

Let's run the simulation with these parameters to evaluate the trajectory
   prediction errors.

## L  LLMPHY OPTIMIZATION AND INTERACTION TRACE (PHASE2)

Similar to above, we present an example optimization steps from a phase 2 LLMPhy optimization.

```
    Sure, here's the code for creating the provided scene based on '
        task_image_top_view_1.png' and 'task_image_side_view_2.png'. The
        object classes and colors are arranged as follows, based on the
        interpretation of the images.

### Arrangement of Objects on the 3x3 Grid:
- (1,1): Yellow (Bottle)
- (1,2): Cyan (Bottle)
- (1,3): Orange (Bottle)
- (2,1): Brown (Martini Glass)
- (2,2): Blue (Martini Glass)
- (2,3): Pink (Martini Glass)
- (3,1): Purple (Wine Glass)
- (3,2): None
- (3,3): Green (Wine Glass)
```

Here's the code to create the scene using the specified physical
    attributes:

```python
sim = SIMULATOR_MODEL()
sim.create_pusher('3.0 3.0 0.05')
physical_parameters_for_object_id_tray = {
    'sliding-friction': 0.1,
    'armature': 0.1,
    'stiffness': 0.0,
    'mass': 0.5,
    'damping': 20
}
sim.create_tray(object_physics = physical_parameters_for_object_id_tray)

# Physical parameters for bottle
physical_parameters_for_bottle = {
    'sliding-friction': 0.21,
    'armature': 0.41,
    'stiffness': 0.55,
    'mass': 20.0,
    'damping': 7.0
}

# Physical parameters for martini glass
physical_parameters_for_martini_glass = {
    'sliding-friction': 0.52,
    'armature': 0.63,
    'stiffness': 1.22,
    'mass': 10.0,
    'damping': 8.0
}

# Physical parameters for wine glass
physical_parameters_for_wine_glass = {
    'sliding-friction': 0.72,
    'armature': 1.02,
    'stiffness': 0.94,
    'mass': 4.0,
    'damping': 7.9
}

sim.create_object(object_id=1, object_name='bottle', object_location=('
    row_1', 'column_1'), object_color='yellow', object_physics=
    physical_parameters_for_bottle)
sim.create_object(object_id=2, object_name='bottle', object_location=('
    row_1', 'column_2'), object_color='cyan', object_physics=
    physical_parameters_for_bottle)
sim.create_object(object_id=3, object_name='bottle', object_location=('
    row_1', 'column_3'), object_color='orange', object_physics=
    physical_parameters_for_bottle)
sim.create_object(object_id=4, object_name='martini_glass',
    object_location=('row_2', 'column_1'), object_color='brown',
    object_physics=physical_parameters_for_martini_glass)
sim.create_object(object_id=5, object_name='martini_glass',
    object_location=('row_2', 'column_2'), object_color='blue',
    object_physics=physical_parameters_for_martini_glass)
sim.create_object(object_id=6, object_name='martini_glass',
    object_location=('row_2', 'column_3'), object_color='pink',
    object_physics=physical_parameters_for_martini_glass)
sim.create_object(object_id=7, object_name='wine_glass', object_location
    =('row_3', 'column_1'), object_color='purple', object_physics=
    physical_parameters_for_wine_glass)
```

```
3186    sim.create_object(object_id=8, object_name='wine_glass', object_location
3187        =('row_3', 'column_3'), object_color='green', object_physics=
3188        physical_parameters_for_wine_glass)
3189
3190    sim.create_scene()
3191    sim_out=sim.run_simulation()
        del sim
3192    ```
3193
3194    This code will recreate the scene as depicted in the provided images '
3195        task_image_top_view_1.png' and 'task_image_side_view_2.png' using the
3196         specific physical parameters given.
3197
        ========
3198
3199    Certainly! Based on the feedback from the previous attempt and a careful
3200        review of the top-down view 'task_image_top_view_1.png', here is a
3201        refined version of the code to correctly match the object positions
        and colors.
3202
3203    ```python
3204    sim = SIMULATOR_MODEL()
3205    sim.create_pusher('3.0 3.0 0.05')
3206    physical_parameters_for_object_id_tray = {
3207        'sliding-friction': 0.1,
3208        'armature': 0.1,
        'stiffness': 0.0,
3209        'mass': 0.5,
3210        'damping': 20
3211    }
        sim.create_tray(object_physics = physical_parameters_for_object_id_tray)
3212
3213    # Physical parameters for bottle
3214    physical_parameters_for_bottle = {
3215        'sliding-friction': 0.21,
3216        'armature': 0.41,
        'stiffness': 0.55,
3217        'mass': 20.0,
3218        'damping': 7.0
3219    }
3220
3221    # Physical parameters for martini glass
3222    physical_parameters_for_martini_glass = {
        'sliding-friction': 0.52,
3223        'armature': 0.63,
3224        'stiffness': 1.22,
3225        'mass': 10.0,
        'damping': 8.0
3226    }
3227
3228    # Physical parameters for wine glass
3229    physical_parameters_for_wine_glass = {
3230        'sliding-friction': 0.72,
3231        'armature': 1.02,
        'stiffness': 0.94,
3232        'mass': 4.0,
3233        'damping': 7.9
3234    }
3235
3236    # Object placements based on correct interpretation
        sim.create_object(object_id=1, object_name='bottle', object_location=('
3237        row_1', 'column_1'), object_color='yellow', object_physics=
3238        physical_parameters_for_bottle)
3239
```

```
3240    sim.create_object(object_id=2, object_name='bottle', object_location=('
3241        row_1', 'column_2'), object_color='cyan', object_physics=
3242        physical_parameters_for_bottle)
3243    sim.create_object(object_id=3, object_name='bottle', object_location=('
3244        row_1', 'column_3'), object_color='orange', object_physics=
3245        physical_parameters_for_bottle)
3246    sim.create_object(object_id=4, object_name='martini_glass',
3247        object_location=('row_2', 'column_1'), object_color='brown',
3248        object_physics=physical_parameters_for_martini_glass)
3249    sim.create_object(object_id=5, object_name='wine_glass', object_location
3250        =('row_2', 'column_2'), object_color='blue', object_physics=
3251        physical_parameters_for_martini_glass)
3252    sim.create_object(object_id=6, object_name='martini_glass',
3253        object_location=('row_2', 'column_3'), object_color='pink',
3254        object_physics=physical_parameters_for_martini_glass)
3255    sim.create_object(object_id=7, object_name='wine_glass', object_location
3256        =('row_3', 'column_1'), object_color='purple', object_physics=
3257        physical_parameters_for_wine_glass)
3258    sim.create_object(object_id=8, object_name='wine_glass', object_location
3259        =('row_3', 'column_2'), object_color='brown', object_physics=
3260        physical_parameters_for_wine_glass)
        sim.create_object(object_id=9, object_name='wine_glass', object_location
            =('row_3', 'column_3'), object_color='green', object_physics=
            physical_parameters_for_wine_glass)

3261    sim.create_scene()
3262    sim_out=sim.run_simulation()
3263    del sim
3264    ```
```

Steps taken:
- Identified that the issue with the object colors in previous Attempt
  #0.
 - 'cyan' should be 'blue'.
 - 'orange' should be 'purple'.
 - 'pink' should be 'brown'.
 - 'blue' should be 'cyan'.
- Corrected these placements and colors to match the top-down view '
  task_image_top_view_1.png'.

Please run this corrected code in your simulator to check if it matches
  the desired output.

========

Certainly! To reproduce the task images correctly, we need to ensure that
  each object is placed in the correct grid location and given the
  correct color. Let's carefully inspect the top-down image and the
  side-view image again:

From 'task_image_top_view_1.png':
1. Top-left (row 1, column 1): Yellow object
2. Top-middle (row 1, column 2): Cyan object
3. Top-right (row 1, column 3): Orange object
4. Middle-left (row 2, column 1): Brown object
5. Middle-middle (row 2, column 2): Blue object
6. Middle-right (row 2, column 3): Pink object
7. Bottom-left (row 3, column 1): Purple object
8. Bottom-middle (row 3, column 2): No object
9. Bottom-right (row 3, column 3): Green object

From 'task_image_side_view_2.png', identify the types of objects:
1. Bottles have a slender, tall shape.
2. Martini glasses have a distinctive cone shape.
3. Wine glasses have a bowl-like shape and are shorter than bottles.

```python
Using these identifications and physical attributes provided, we will
    correct the mistake identified in the attempt. Below is the refined
    version of the Python code to correctly reproduce the top-down and
    side-view images of the scene.

'''python
sim = SIMULATOR_MODEL()
sim.create_pusher('3.0 3.0 0.05')

physical_parameters_for_object_id_tray = {
    'sliding-friction': 0.1,
    'armature': 0.1,
    'stiffness': 0.0,
    'mass': 0.5,
    'damping': 20
}
sim.create_tray(object_physics = physical_parameters_for_object_id_tray)

# Physical parameters for bottle
physical_parameters_for_bottle = {
    'sliding-friction': 0.21,
    'armature': 0.41,
    'stiffness': 0.55,
    'mass': 20.0,
    'damping': 7.0
}

# Physical parameters for martini glass
physical_parameters_for_martini_glass = {
    'sliding-friction': 0.52,
    'armature': 0.63,
    'stiffness': 1.22,
    'mass': 10.0,
    'damping': 8.0
}

# Physical parameters for wine glass
physical_parameters_for_wine_glass = {
    'sliding-friction': 0.72,
    'armature': 1.02,
    'stiffness': 0.94,
    'mass': 4.0,
    'damping': 7.9
}

# Correct object placements and colors based on the images
sim.create_object(object_id=1, object_name='bottle', object_location=('
    row_1', 'column_1'), object_color='yellow', object_physics=
    physical_parameters_for_bottle)
sim.create_object(object_id=2, object_name='bottle', object_location=('
    row_1', 'column_2'), object_color='cyan', object_physics=
    physical_parameters_for_bottle)
sim.create_object(object_id=3, object_name='bottle', object_location=('
    row_1', 'column_3'), object_color='orange', object_physics=
    physical_parameters_for_bottle)
sim.create_object(object_id=4, object_name='martini_glass',
    object_location=('row_2', 'column_1'), object_color='brown',
    object_physics=physical_parameters_for_martini_glass)
sim.create_object(object_id=5, object_name='bottle', object_location=('
    row_2', 'column_2'), object_color='blue', object_physics=
    physical_parameters_for_bottle)
sim.create_object(object_id=6, object_name='martini_glass',
    object_location=('row_2', 'column_3'), object_color='pink',
    object_physics=physical_parameters_for_martini_glass)
```

```
sim.create_object(object_id=7, object_name='martini_glass',
    object_location=('row_3', 'column_1'), object_color='purple',
    object_physics=physical_parameters_for_martini_glass)
sim.create_object(object_id=8, object_name='wine_glass', object_location
    =('row_3', 'column_3'), object_color='green', object_physics=
    physical_parameters_for_wine_glass)

sim.create_scene()
sim_out=sim.run_simulation()
del sim
```

This code ensures that each object is assigned the correct position and
    color as per the task images. The corrected physical parameters will
    also contribute to a more accurate simulation output.

========

To correctly reproduce the scene shown in the "task_image_top_view_1.png"
    and "task_image_side_view_2.png" images, we need to ensure that the
    placement, object types, and colors are all accurately represented.
    From GPT Attempt #0, we've identified that there are issues with the
    colors of the objects placed in the scene.

Let's refine the code to align the objects' colors and positions
    accurately according to the provided images.

Here's the corrected code snippet:

```python
sim = SIMULATOR_MODEL()
sim.create_pusher('3.0 3.0 0.05')

physical_parameters_for_object_id_tray = {
    'sliding-friction': 0.1,
    'armature': 0.1,
    'stiffness': 0.0,
    'mass': 0.5,
    'damping': 20
}
sim.create_tray(object_physics = physical_parameters_for_object_id_tray)

# Physical parameters for bottle
physical_parameters_for_bottle = {
    'sliding-friction': 0.21,
    'armature': 0.41,
    'stiffness': 0.55,
    'mass': 20.0,
    'damping': 7.0
}

# Physical parameters for martini glass
physical_parameters_for_martini_glass = {
    'sliding-friction': 0.52,
    'armature': 0.63,
    'stiffness': 1.22,
    'mass': 10.0,
    'damping': 8.0
}

# Physical parameters for wine glass
physical_parameters_for_wine_glass = {
    'sliding-friction': 0.72,
    'armature': 1.02,
    'stiffness': 0.94,
```

```
       'mass': 4.0,
       'damping': 7.9
}

# Correct object placements and colors based on the images
sim.create_object(object_id=1, object_name='bottle', object_location=('
    row_1', 'column_1'), object_color='yellow', object_physics=
    physical_parameters_for_bottle)
sim.create_object(object_id=2, object_name='bottle', object_location=('
    row_1', 'column_2'), object_color='cyan', object_physics=
    physical_parameters_for_bottle)
sim.create_object(object_id=3, object_name='bottle', object_location=('
    row_1', 'column_3'), object_color='orange', object_physics=
    physical_parameters_for_bottle)
sim.create_object(object_id=4, object_name='martini_glass',
    object_location=('row_2', 'column_1'), object_color='brown',
    object_physics=physical_parameters_for_martini_glass)
sim.create_object(object_id=5, object_name='bottle', object_location=('
    row_2', 'column_2'), object_color='blue', object_physics=
    physical_parameters_for_bottle)
sim.create_object(object_id=6, object_name='wine_glass', object_location
    =('row_2', 'column_3'), object_color='pink', object_physics=
    physical_parameters_for_wine_glass) # Corrected to wine glass
sim.create_object(object_id=7, object_name='martini_glass',
    object_location=('row_3', 'column_1'), object_color='purple',
    object_physics=physical_parameters_for_martini_glass) # Corrected
    color position
sim.create_object(object_id=8, object_name='wine_glass', object_location
    =('row_3', 'column_3'), object_color='green', object_physics=
    physical_parameters_for_wine_glass)

sim.create_scene()
sim_out=sim.run_simulation()
del sim
```

Changes made:
1. For `object_id=6`, the type was changed to `wine_glass` instead of `martini_glass`.
2. The color `purple` was moved to position `(row_3, column_1)` for the `martini_glass` as per the task image.

Ensure the scene setup provided above now accurately reflects the task images. With these corrections, the positions and object types should match the top-down and side view images respectively.

========

To refine the Python code for generating the scene to exactly match the given images ('task_image_top_view_1.png' and 'task_image_side_view_2.png'), we need to correct the object colors and their placements on the grid. Below is the mistake analysis from GPT Attempt #0 and the necessary corrections:

1. **Object at `(row_1, column_2)`**: The given image shows this is yellow, but the existing code has it as cyan.
2. **Object at `(row_1, column_3)`**: The given image shows this is cyan, but the existing code has it as orange.
3. **Object at `(row_3, column_1)`**: The given image shows this is orange, but the existing code has it as purple.

We can use this information to adjust the code appropriately:

1. Place the cyan bottle at `(row_1, column_3)`.
2. Place the orange bottle at `(row_3, column_1)`.

```python
3. Place the purple martini glass at `(row_2, column_2)`.

Let's write the corrected code accordingly:

```python
sim = SIMULATOR_MODEL()
sim.create_pusher('3.0 3.0 0.05')

physical_parameters_for_object_id_tray = {
    'sliding-friction': 0.1,
    'armature': 0.1,
    'stiffness': 0.0,
    'mass': 0.5,
    'damping': 20
}
sim.create_tray(object_physics = physical_parameters_for_object_id_tray)

# Physical parameters for bottle
physical_parameters_for_bottle = {
    'sliding-friction': 0.21,
    'armature': 0.41,
    'stiffness': 0.55,
    'mass': 20.0,
    'damping': 7.0
}

# Physical parameters for martini glass
physical_parameters_for_martini_glass = {
    'sliding-friction': 0.52,
    'armature': 0.63,
    'stiffness': 1.22,
    'mass': 10.0,
    'damping': 8.0
}

# Physical parameters for wine glass
physical_parameters_for_wine_glass = {
    'sliding-friction': 0.72,
    'armature': 1.02,
    'stiffness': 0.94,
    'mass': 4.0,
    'damping': 7.9
}

# Correct object placements and colors based on the images
sim.create_object(object_id=1, object_name='bottle', object_location=(
    'row_1', 'column_1'), object_color='yellow', object_physics=
    physical_parameters_for_bottle)
sim.create_object(object_id=2, object_name='bottle', object_location=(
    'row_1', 'column_3'), object_color='cyan', object_physics=
    physical_parameters_for_bottle) # Corrected to cyan and position
sim.create_object(object_id=3, object_name='bottle', object_location=(
    'row_3', 'column_1'), object_color='orange', object_physics=
    physical_parameters_for_bottle) # Corrected to orange and position
sim.create_object(object_id=4, object_name='martini_glass',
    object_location=('row_2', 'column_1'), object_color='brown',
    object_physics=physical_parameters_for_martini_glass)
sim.create_object(object_id=5, object_name='bottle', object_location=(
    'row_2', 'column_2'), object_color='blue', object_physics=
    physical_parameters_for_bottle)
sim.create_object(object_id=6, object_name='wine_glass', object_location
    =('row_2', 'column_3'), object_color='pink', object_physics=
    physical_parameters_for_wine_glass)
sim.create_object(object_id=7, object_name='martini_glass',
    object_location=('row_1', 'column_2'), object_color='purple',
```

```
3510        object_physics=physical_parameters_for_martini_glass) # Corrected to
3511        purple and position
3512    sim.create_object(object_id=8, object_name='wine_glass', object_location
3513        =('row_3', 'column_3'), object_color='green', object_physics=
3514        physical_parameters_for_wine_glass)
3515
3516    sim.create_scene()
3517    sim_out=sim.run_simulation()
3518    del sim
        ```

3519
3520    This code should now correctly reproduce the task images:
3521
3522    - **Object Placements**:
3523      - '(row_1, column_1)' -> Yellow bottle
3524      - '(row_1, column_2)' -> Purple martini glass
3525      - '(row_1, column_3)' -> Cyan bottle
3526      - '(row_2, column_1)' -> Brown martini glass
3527      - '(row_2, column_2)' -> Blue bottle
3528      - '(row_2, column_3)' -> Pink wine glass
        - '(row_3, column_1)' -> Orange bottle
        - '(row_3, column_3)' -> Green wine glass
3529
3530    With these corrected attributes and proper placements, running the
        simulation should yield the desired images.
3531
3532    ========
3533
3534    Thank you for providing the various details and previous attempts. Let's
        analyze the array of object placements and the corresponding colors
        more carefully, given both views and the physical properties.
3535
3536
3537    The given task_images (task_image_top_view_1.png and
3538        task_image_side_view_2.png) indicate a specific arrangement of
3539        colored objects. Let's identify and correct the colors and positions
3540        of the objects in the generated scene to match the task:
3541
3542    1. Yellow object: A bottle, located in row 1, column 1.
3543    2. Cyan object: A bottle, located in row 1, column 3.
3544    3. Orange object: A bottle, located in row 1, column 2.
        4. Purple object: A martini glass, located in row 3, column 2.
3545    5. Blue object: A bottle, located in row 2, column 2.
3546    6. Pink object: A wine glass, located in row 3, column 1.
3547    7. Green object: A wine glass, located in row 3, column 3.
        8. Brown object: A martini glass, located in row 2, column 1.
3548
3549    Using the corrected layout information, here is the refined Python code
3550        to reproduce the task images correctly:
3551
3552    ```python
3553    sim = SIMULATOR_MODEL()
        sim.create_pusher('3.0 3.0 0.05')
3554
3555    # Physical parameters for the tray
        physical_parameters_for_object_id_tray = {
3556        'sliding-friction': 0.1,
3557        'armature': 0.1,
3558        'stiffness': 0.0,
3559        'mass': 0.5,
3560        'damping': 20
        }
3561    sim.create_tray(object_physics = physical_parameters_for_object_id_tray)
3562
3563    # Physical parameters for bottle
        physical_parameters_for_bottle = {
```

```
'sliding-friction': 0.21,
'armature': 0.41,
'stiffness': 0.55,
'mass': 20.0,
'damping': 7.0
}

# Physical parameters for martini glass
physical_parameters_for_martini_glass = {
    'sliding-friction': 0.52,
    'armature': 0.63,
    'stiffness': 1.22,
    'mass': 10.0,
    'damping': 8.0
}

# Physical parameters for wine glass
physical_parameters_for_wine_glass = {
    'sliding-friction': 0.72,
    'armature': 1.02,
    'stiffness': 0.94,
    'mass': 4.0,
    'damping': 7.9
}

# Correct object placements and colors based on the task images
sim.create_object(object_id=1, object_name='bottle', object_location=('
    row_1', 'column_1'), object_color='yellow', object_physics=
    physical_parameters_for_bottle)
sim.create_object(object_id=2, object_name='bottle', object_location=('
    row_1', 'column_3'), object_color='cyan', object_physics=
    physical_parameters_for_bottle)
sim.create_object(object_id=3, object_name='bottle', object_location=('
    row_1', 'column_2'), object_color='orange', object_physics=
    physical_parameters_for_bottle)
sim.create_object(object_id=4, object_name='martini_glass',
    object_location=('row_2', 'column_1'), object_color='brown',
    object_physics=physical_parameters_for_martini_glass)
sim.create_object(object_id=5, object_name='bottle', object_location=('
    row_2', 'column_2'), object_color='blue', object_physics=
    physical_parameters_for_bottle)
sim.create_object(object_id=6, object_name='wine_glass', object_location
    =('row_3', 'column_1'), object_color='pink', object_physics=
    physical_parameters_for_wine_glass)
sim.create_object(object_id=7, object_name='wine_glass', object_location
    =('row_3', 'column_3'), object_color='green', object_physics=
    physical_parameters_for_wine_glass)
sim.create_object(object_id=8, object_name='martini_glass',
    object_location=('row_3', 'column_2'), object_color='purple',
    object_physics=physical_parameters_for_martini_glass)

sim.create_scene()
sim_out=sim.run_simulation()
del sim
```
```

This code corrects the placement and color assignments for the given
    objects, ensuring they match what is observed in the task image. Run
    this code in your simulator to verify the outputs.

