# OpenReview forum: "LLMPhy: Complex Physical Reasoning Using Large Language Models and World Models"
_ICLR.cc/2025/Conference — Submitted to ICLR 2025_

### Official Review · Reviewer_THXz · 2024-10-31

**Soundness:** 2
**Presentation:** 2
**Contribution:** 1
**Rating:** 5
**Confidence:** 4

**Summary:**

This work introduces LLMPhy, a zero-shot black-box optimization framework that integrates LLMs with physics engines to predict the dynamics of objects on a tray under external impact. The framework comprises two main phases:
1. The LLM infers the physical parameters of objects, including the coefficient of sliding friction, stiffness, damping, and rotational inertia, based on in-context learning examples and the objects' physical trajectories extracted from video data. The LLM then generates a program to simulate these trajectories, aiming to minimize the error between simulated and actual observed trajectories.
2. The LLM determines the scene layout by inferring object characteristics such as class, location, and color, based on the previously inferred physical parameters, the initial frame of the video sequence, and additional in-context examples.

Finally, LLMPhy refines its predictions using feedback from the simulator to minimize error. At each step, it keeps a record of previous attempts and errors, which helps the model improve its accuracy over multiple trials.


This work also introduces TraySim, a new dataset of simulated video sequences featuring a tray with varying numbers of objects. Evaluation of TraySim reveals that the proposed method outperforms baseline approaches, including random selection of physical parameters and direct use of an LLM.

**Strengths:**

This work presents a new approach that combines the strengths of large language models with advanced physics simulators. Additionally, it establishes a benchmark dataset, TraySim, which could be valuable to support future research in physical dynamics prediction.

**Weaknesses:**

1. The optimization step in Algorithm 1 appears to depend largely on the LLM’s ability to learn from previous errors through repeated attempts, making the approach heavily reliant on trial-and-error. Other optimization algorithms, such as CMA-ES, might also achieve comparable or potentially even better results. To strengthen the paper’s claims, consider including comparisons with additional optimization algorithms.
2. The proposed method is similar to the approach in the paper "Eureka: Human-Level Reward Design via Coding Large Language Models." In that work, an LLM is used to design reward functions that help train a reinforcement learning agent to teach a five-fingered robotic hand how to spin a pen.
3. The choice of intersection over union (IoU) as the evaluation metric is unclear. What does the answer set consist of? Why not simply use an exact match with the answer set instead?
4. In the second phase of LLMPhy, it’s unclear why predicting the color of the object is relevant for understanding its dynamics. Additionally, why is an LLM necessary to infer the object’s starting location? Wouldn’t it be simpler to use the tau function to extract the object’s location directly from the first frame of the image?
5. In Figure 2, an LLM processes the question along with the synthesized dynamics video to produce an answer, but it is unclear where the paper explains this process in detail. How exactly is this achieved?
6. The generalizability of the proposed Algorithm 1 is unclear, as it appears that the algorithm needs to be reinvoked for each new scenario to infer the physical parameters of the objects. Is there a way to enable the LLM to remember what it has learned and apply it to future scenarios?

**Questions:**

1. How is $\tau$ implemented?
2. Why choose to generate programs rather than directly generating relevant values for each attribute, especially when no real program logic, such as if conditions, for loops, or external libraries, is used?

---

> ### Author Response · Authors · 2024-11-23
> **Response to Reviewer THXz**
>
> We thank the reviewer for carefully reading our paper and for the valuable  suggestions. We have thoroughly revised the paper and made all the changes that were suggested. We sincerely hope the reviewer can have a look at our revised paper and let us know of any additional changes. Please see below for the precise changes that we have made in the paper to account for your criticisms.
>
> * *Other optimization algorithms, such as CMA-ES, might also achieve comparable or potentially even better results.*
>
> Thanks for this important point. We have now incorporated comparisons to CMA-ES in all our experiments along with comparisons to Bayesian Optimization. Please see our revised results in Table 1, 3,4,5, as well as plots in Figure 5 and 10.
>
> * *The proposed method is similar to the approach in the paper "Eureka: Human-Level Reward Design via Coding Large Language Models."*
>
> Thanks for pointing us to this important paper. We have now cited and contrasted this paper in our Related Works. Specifically, Eureka proposes an LLM-based program synthesis for designing reward functions in a reinforcement learning (RL) setting, where each iteration of their evolutionary search procedure produces a set of LLM generated candidate reward functions. Apart from the task we consider in LLMPhy, which is to bring out the continuous and discrete black-box optimization capabilities of the LLM-Simulator combination, LLMPhy differs from Eureka in two other aspects: (i) Eureka involves additional RL training that may bring extraneous noise from the training dynamics in fitness evaluation and  (ii) does not use full trajectory of optimization in its feedback and as a result, the LLMs may reconsider previous choices. To validate the last point where Eureka only uses the best solution so far for training an RL agent, we attempted a similar setup with LLMPhy where instead of providing the full optimization trace as we do, we only provided to the LLM the last best solution. Our results in Table 5 (Experiments 9,10) show that this leads to about 5% drop in performance against using the full optimization trajectory. We believe using the longer history allows the LLMs to avoid retrying parameters and thus allows better search of the optimization landscape. LLMPhy is also guided by the physical / semantic interpretations of the optimization variables that we consider, i.e., unlike Eureka that optimizes over variables, LLMPhy optimizes over variables that have physical meanings as characterized by their variable names in the program, e.g., our optimization variables are named sliding-friction, stiffness, etc. We see from Appendix K (eg., L1804, L1884, etc.).
>
> * *The choice of intersection over union (IoU) as the evaluation metric is unclear. What does the answer set consist of? Why not simply use an exact match with the answer set instead?*
>
> We apologize for the omission of this detail. Please see examples in Figures 4 and 13 that show the evaluation setup clearly along with qualitative examples. We now report the exact match accuracy in Table 5 of the revised paper where we see LLMPhy offers better performances. Unfortunately, for the results in Table 1, we did not see any significant difference in the precise IoU between LLMPhy and other methods (LLMPhy showed an average precise IoU of 11% while others show <~10%); and the numbers being low, we thought it is not worth reporting, while the Avg. IoU appeared a better evaluation.
>
> * *In the second phase, it’s unclear why predicting the color of the object is relevant, why is an LLM necessary to infer the object’s starting location? use the tau function to extract the object’s location?*
>
> The goal of Phase 2 is to estimate the scene layout (perception task) using discrete black-box optimization so that the dynamics of these objects can be produced from the provided task images. We have clarified this aspect in Figure 3 in the revised paper. As the answer needs to be selected from the options, as depicted in Figure 4, the model needs correct inference of the object color and type, any error in the inference of these Phase 2 attributes will make an incorrect selection of the options. Specifically, note that the dynamics of the objects after the impact depend on their location on the tray and the location of the impact. Thus, if an object is incorrectly placed in a different location, then it will have different dynamics and incorrect inference of its final state. We have included **a few sample videos in the supplementary materials to clarify our setup better**.
>
> *Use the tau function to extract the object’s location?* Note that the \tau function is only available on the auxiliary sequences and assumes access to the videos. However, our task is to produce videos from images in Phase 2 using the simulator with the physics estimated in Phase 1 so as to answer the QA question. We hope Figure 3 will clarify this important point.

---

> ### Author Response · Authors · 2024-11-23
> **Response to Reviewer THXz (2)**
>
> * *In the second phase, it’s unclear why predicting the color of the object is relevant, why is an LLM necessary to infer the object’s starting location? use the tau function to extract the object’s location?*
>
> The goal of Phase 2 is to estimate the scene layout which poses a discrete optimization task and presenting an opportunity to explore the discrete black-box optimization capabilities of LLMPhy. Note that the $\tau$ function is assumed only available on the auxiliary sequences. In the experiments reported in Table 1,3,4,5, we also report situations when the ground truth locations of the objects and their types are assumed given to LLMPhy, which we believe is exactly the setup the Reviewer is seeking when a form of the \tau function is available for Phase 2. We see better results using LLMPhy in this case as well.
>
> * *In Figure 2, an LLM processes the question along with the synthesized dynamics video to produce an answer, but it is unclear where the paper explains this process in detail. How exactly is this achieved?*
>
> Thanks for asking this question. In our current architecture, the LLM does not process the generated video, instead our code takes the last video frame, processes it, extracts the pose of the objects, and uses this pose to decide their stability to answer the QA question. We ought to say that we may in fact use an LLM to do the same task, where the LLM may be asked which objects on the tray have fallen down. We thought this will make the QA setup general, however this is currently not being done in the paper. This is now clarified in L342.
>
> *The generalizability of the proposed Algorithm 1 is unclear, as it appears that the algorithm needs to be reinvoked for each new scenario to infer the physical parameters of the objects. Is there a way to enable the LLM to remember what it has learned and apply it to future scenarios?*
>
> As we expect a zero-shot setting, we do not assume the knowledge of a previous solution is useful in solving a later problem. However, including such a possibility is straightforward and could be done either through in-context examples or doing a pre-processing of the problem setup to match against a database of solutions, and initializing the LLMPhy Phase 1 programs with the best history estimates. Our goal in this paper is to avoid any such assumptions and treat the black-box optimization setup be evaluated in the most general manner.
>
> * *How is τ implemented?*
>
> Thanks for asking this. We missed to explain it in the paper and have now clarified in Appendix/Section D1 (L890-898). We use the simulator in Phase 1 to extract the trajectories of the object instances after the impact.
>
> * *Why choose to generate programs rather than directly generating relevant values for each attribute, especially when no real program logic, such as if conditions, for loops, or external libraries, is used?*
>
> Thanks for this great question! We found that using a (Python) program as a mode of communication between the LLM and the simulator (and leveraging the coding abilities of LLMs) allows for great flexibility in the implementation. Specifically,
>
>  i) it disconnects the implementation details of the simulation from the LLM, i.e., when executing the program, the simulator will be invoked directly and the code will assign the physical attributes to the respective object instances automatically, while this will need a manual parsing of the parameter values otherwise.
>
> ii) programs allow better interpretability for the LLM and in its responses, e.g., we associate the optimization variables with physical meanings for the LLM such as naming them as ‘friction’, ‘mass’, etc., the LLM may use such details in generating its responses (See Section J/Appendix/L1600 for specific examples where the LLM explanations for adjusting the physical parameters are provided),
>
> iii) the specific structure of our used programs allows for invariance to their order of execution, e.g., sim.create_object() functions (See Figure 3), and
>
> iv) each example in our dataset in Phase 2 contains an arbitrary number of object instances, and having a program to recreate our setup is significantly easier than parsing the LLM responses, matching their parameters, and execution.
>
> Looking at this from a programming perspective, we are leveraging variable reuse through program synthesis, where all the shared physics parameters associated with the same class of objects share the same physics variables in Phase 2. See for example, a synthesized program in Phase 2 in L1648 onwards in the Appendix.

---

> > ### Comment · Reviewer_THXz · 2024-11-27
> > **Response to Author**
> >
> > The quality of the paper has improved significantly. However, I still have a few questions regarding its presentation and contributions:
> >
> > 1. I find Figure 2 confusing. Does the physics simulator generate the synthesized dynamic video, which is then sent to the LLM/SIM? Also, what exactly is LLM/SIM? My understanding is that the LLM is not used to answer questions. Could you clarify this?
> >
> > 2. Regarding my earlier comment about using $\tau$ as a replacement for Phase 2: if someone already has access to $\tau$, what is the purpose of Phase 2? Can the proposed method work without requiring $\tau$?
> >
> > 3. I am not convinced why color matters in this setting. Couldn’t the objects be identified using alternative labels, such as assigning each one a unique ID?
> >
> > 4. The explanation of "How is τ implemented" is confusing. In line 288, it states that τ takes in a video and outputs a trajectory, but the appendix seems to suggest otherwise. Can you clarify?
> >
> > Overall, I have broader concerns about the practical use cases of this work. The approach appears to rely on many assumptions, and I would appreciate clarification on its use cases. Could you provide a concrete example to help me better understand how it can be used in real life? I will adjust my score accordingly if my concerns are addressed convincingly.

---

> ### Author Response · Authors · 2024-11-28
> **Response to Additional Questions of Reviewer THXz (1)**
>
> Dear Reviewer THXz,
>
> We thank you for carefully reading our revision and for asking additional questions. We apologize that some of our previous responses were not clear enough. Please find below our responses to your questions.
>
> *1. Does the simulator generate the dynamic video, which is then sent to the LLM/SIM? Also, what is LLM/SIM?*
>
> Yes, our task is to reason on the images and thus in Phase 2, the model uses the physics estimated in Phase 1 within the simulator to produce a dynamic video from the images, the last video frame from this synthesized video is used to answer the question regarding the stability of the objects.
>
> By LLM / SIM, we meant to say that the inference of the final answer can either be done using a large vision-language model (VLM) or using the simulator. That is, we may use a VLM with input as the last video frame and the task question, and retrieve the VLM's response to answer or we may directly extract the pose of each of the objects from the simulator to answer. While the former is a more general approach, we use the latter as it is straightforward to implement. Thus, in our setup, a VLM is not used to produce the final answer, however, we meant to say that if the video is synthesized by a different means other than a simulator, then a VLM may be used. We will clarify this in the revised paper.
>
> *2. If someone has access to $\tau$ what is the purpose of Phase 2? Can the method work without $\tau$?*
>
> Thanks for asking to clarify this point better. Note that $\tau$ abstracts the implementation of how the measurements of the *observables* in the scene (e.g., object trajectories) are done towards estimating the *unobservables* (e.g., physics attributes). As $\tau$ is used to extract object trajectories from videos and since there are no videos provided in Phase 2, $\tau$ cannot be used in Phase 2. However, an implementation of $\tau$ is essential to the estimation of the physical attributes of the system in Phase 1, where attributes such as friction, stiffness, etc., are dynamical parameters that cannot be estimated from the images alone, (e.g., Table 1, Phase 1: Random, Phase 2: LLMPhy, show ~10% lower accuracy). We would like to emphasize that while $\tau$ is essential for our task, its specific implementation using a simulator is not essential, and it could be implemented in other ways such as object motion tracks, AprilTags, or other sensors.
>
> *3. Why color matters? Couldn’t the objects be identified using labels?*
>
> Thanks again for letting us clarify this part better.  The Phase 2 of LLMPhy solves the perception task of inferring the scene layout (i.e., objects' positions and classes) from the given multiview images to render the synthesized videos correctly.  Now, *why is color so important in this process?* In order to understand this, we need to understand what the VLM produces in a  Phase 2 program, which are the triplets (class, position, color) for each object in the task image. As we found it impossible for the VLM to produce exact (x, y, z) coordinates of the objects on the tray, we had to discretize their positions on a 3x3 grid so that the VLM may produce the location of an object as (row_x, col_y) in the synthesized program. To do this, the VLM uses a top-down view image. We found this to be important as any other view will have object occlusions and thus VLM cannot infer the grid positions correctly (see e.g., Figure 4 top-left). However, the top-down view makes it difficult for the VLM to infer the object class (e.g., martini and wine glasses may look similar), and side-views of the scene are better (Figure 4 top-left).
>
> Now, *how will the VLM know if it is looking at the same object in the two views when the view information is not provided?* We found that having a unique color for the objects could guide the VLM to do the association between the views to infer the position and class of the same object, i.e., the top-down view helps infer the position and the side-views for the class, and the views are connected by the object color. In fact, we may use any other possibility for connecting the views, e.g., a unique label as the reviewer suggests or visually unique shapes, textures, characters, etc., however color offers a simple way to fix this issue. In the experiments presented in Table 1 Experiment 8 and 9, we in fact assume that the Phase 2 program for the layout of the objects is known (GT), LLMPhy demonstrating better results then as well.
>
> *4. Explanation is confusing. Can you clarify?*
>
> Thank you for pointing this out. In L288, we meant to write $\tau$ in its mathematically general form using auxiliary videos, however in our specific case, we use the simulator as an implementation of $\tau$, as explained in the supplementary L890. We have uploaded the revised the paper fixing this inconsistency (L289).
>
> Your questions are extremely valuable to improve our work and we sincerely thank you for asking them. Please let us know if you have other questions.

---

> ### Author Response · Authors · 2024-11-28
> **Response to Additional Questions of Reviewer THXz (2)**
>
> *I have broader concerns about the practical use cases of this work. The approach appears to rely on many assumptions, and I would appreciate clarification on its use cases. Could you provide a concrete example to help me better understand how it can be used in real life?*
>
> We believe our work represents a important step towards utilizing an LLM-Simulator combination to solve black-box optimization problems, especially for objectives that do not permit end-to-end gradients. Our study demonstrates that leveraging the program synthesis capabilities of state-of-the-art Large Language Models (LLMs) enables our method to effectively handle both discrete and continuous black-box optimization tasks. To the best of our knowledge, this is a novel finding, as typically distinct methods—such as CMA-ES or Bayesian Optimization for continuous tasks and other specialized techniques for discrete tasks—are employed.
>
> Furthermore, LLMPhy allows for the association of semantic meanings with optimization variables, thereby tapping into the world knowledge inherent in LLMs to enhance optimization performance. Our approach also provides an interpretable and accessible means to directly estimate unobservable physical attributes of a system using observable entities. We believe that our findings have significant implications for various fields, including computer vision, robotics, and optimization.
>
> **Use Case: A Service Robot**
>
> A practical application of LLMPhy could involve a service robot in a restaurant tasked with carrying a tray containing multiple glasses to a serving table without any glasses falling down. In this scenario, the robot is expected to infer the object layout from vision while also have knowledge of the physics of the objects on the tray so that it can plan its maneuvers. Additionally, it may have access to auxiliary videos or interactive setups where it can observe the outcomes of its interactions with the objects so that it learns the physics involved with the objects. In fact, the robot may infer the physics from the initial steps of its tray (with objects) manipulation as well and observing the motion of the glasses.
>
> In our setup, we employ a simulator as a stand-in for a physical robot to demonstrate the feasibility of our algorithm. This choice allows us to focus on validating our approach without the complexities of real-world hardware interactions. We operate under a zero-shot setting, where the robot is not trained on new types of objects. Instead, it learns the physics of the objects with the minimal number of interactions required. These conditions motivated the assumptions underlying our approach.
>
> Our results indicate that by integrating an LLM with a simulator (or potentially a physical robot), we can solve this problem in a zero-shot manner. This integration not only simplifies the learning process but also enhances the robot's ability to generalize its understanding of object stability without extensive retraining.

---

> > ### Comment · Reviewer_THXz · 2024-11-29
> > **Response to the Author**
> >
> > For my second question, why do you assume that there is no video in phase 2? Does phase 2 involve a completely different setup compared to phase 1, even though the same objects are used? Is your approach leveraging phase 1 to infer object attributes from observed video sequences and then generalizing these to other settings? Algorithm 1 mentions optimizing for both phase 1 and phase 2, but this is unclear: which part of algorithm 1 corresponds to phase 1, and which to phase 2?
> >
> > Additionally, I am not sure about the validity of the claim regarding the novelty of using LLMs to handle both discrete and continuous black-box optimization tasks. For reference, please see Large Language Models as Optimizers and Language to Rewards for Robotic Skill Synthesis.
> >
> > In the example involving the service robot, why is Phase 2 necessary? With powerful perception models today, we already have highly capable systems that can detect the location, class, and color of objects. Why rely on a VLM to perform the task? For Phase 1, reinforcement learning could achieve similar results, though its training time might be a concern. However, I see a similar issue with current LLMs, as they also tend to be very slow in such applications.

---

> ### Author Response · Authors · 2024-11-29
> **Response to Additional Questions**
>
> Dear Reviewer THXz,
>
> Thank you for your questions to further clarify our approach. We will be incorporating all your feedback when revising our paper. Please find below our responses to your questions. Kindly let us know if you have more questions.
>
> * *1. For my second question, why do you assume that there is no video in phase 2?*
>
> The goal of our task is to predict the outcome of the impact from only the given multi-view images of the initial scene. If there is a video provided for Phase 2, then we may use the video to directly find the answer to the QA question and in that case, we may be solving a different task.
>
> * *2. Does phase 2 involve a completely different setup compared to phase 1, though the same objects are used? Is your approach leveraging phase 1 to infer object attributes from observed video sequences and then generalizing these to other settings?*
>
> Yes, that is correct. Phase 2 uses the same object classes that have the same physical parameters estimated in Phase 1, however **uses a significantly different setup**, including the number of objects' instances (which is 5-9 instead of $\leq$ 3), their color and positions, the (x, y) velocities of the pusher (random in both x and y), etc.
>
> * *3. Algorithm 1 mentions optimizing for both phase 1 and phase 2, but this is unclear: which part of algorithm 1 corresponds to phase 1, and which to phase 2?*
>
> We use the same optimization approach described in Algorithm 1 for both Phases 1 and 2. In Phase 1, the variable $\Lambda$ in Algorithm 1 will stand for the physical attributes which are to be optimized, while in Phase 2, $\Lambda$ comprises the spatial and other qualitative attributes to be inferred.
>
> * *I am not sure about the validity of the claim on the novelty of using LLMs to handle both discrete and continuous black-box optimization tasks. See Large Language Models as Optimizers and Language to Rewards for Robotic Skill Synthesis.*
>
> Thank you for this question and allowing us to highlight the differences to these relevant references.
> We have already cited the work presented in “Large Language Models as Optimizers”, which explores the use of LLMs as optimizers for directly and iteratively generating parameter values to solve objectives such as continuous linear regression and discrete traveling salesman problems. However, a key novelty of our work lies in demonstrating that both continuous and discrete black-box optimization problems can be effectively addressed by **leveraging the program synthesis capabilities of LLMs** to minimize objectives that may not even possess an analytical form, thus *generalizing* the setup in the prior work. In our approach, we integrate a simulator within this framework, enabling the handling of complex, non-analytical objectives.
>
> Regarding the paper “Language to Rewards for Robotic Skill Synthesis”, we appreciate you bringing it to our attention. We will include it in our revised paper. However, this study utilizes an LLM to generate reward functions for a simulated robot, **without incorporating the LLM-simulator feedback loop** that is central to LLMPhy.
>
> Our proposed setup, which **integrates LLMs with a simulator to provide feedback using program synthesis**, is essential for solving the tasks addressed in our paper and represents a novel contribution to the field of black-box optimization.
>
> * *In the example involving the service robot, why is Phase 2 necessary? Why rely on a VLM to perform the task?*
>
> While current computer vision foundation models are indeed powerful and capable of detecting the location, class, and color of objects, implementing Phase 2 using these models would require integrating multiple specialized components. For instance, one would need separate models for object detection, spatial relation detection, and potentially additional training for non-standard object classes. This modular approach not only increases the complexity of the system but also demands significant effort to coordinate and maintain these disparate models effectively. In contrast, our approach leverages a powerful VLM, such as GPT-4, to perform these tasks cohesively without the need for assembling various specialized models. The VLM inherently understands object semantics and spatial relationships, thereby simplifying the implementation process.
>
> Additionally, our iterative optimization method allows Phase 2 to progressively enhance accuracy over multiple iterations, even if initial attempts contain errors. This iterative refinement is demonstrated in Table 2 and Figure 5 (b), where we show the performance improvement of Phase 2 from a single iteration to five iterations. Furthermore, achieving similar iterative improvements with traditional vision foundation models in a zero-shot setting would be challenging. These models may not inherently support progressive accuracy enhancements without explicit re-training or additional mechanisms, which our LLM-Simulator feedback setup seamlessly facilitates.

---

> > ### Author Response · Authors · 2024-11-29
> > **Response to Additional Questions (2)**
> >
> > * *For Phase 1, reinforcement learning could achieve similar results, though its training time might be a concern. However, I see a similar issue with current LLMs, as they also tend to be very slow in such applications.*
> >
> >
> > Thank you for your insightful feedback. We agree that RL may be utilized to address the Phase 1 objective. However, we believe that Bayesian Optimization (BO) or Covariance Matrix Adaptation Evolution Strategy (CMA-ES) may offer faster and more effective solutions for this task. In our experiments, while CMA-ES demonstrates competitive performance, our proposed method achieves superior estimation of physics parameters and exhibits higher accuracy across all conducted experiments. We acknowledge that the processing time of LLMPhy is influenced by the response time of the LLM, which can vary due to factors beyond our control. Nonetheless, we anticipate improvements in this area in the future. Additionally, our findings highlight that integrating an LLM within the optimization loop, alongside other complex modules, provides substantial benefits.

---

> > > ### Comment · Reviewer_THXz · 2024-12-02
> > >
> > > Thank you for your detailed response and efforts to address my concerns. While I appreciate the clarifications, I remain unconvinced about the claimed novelty of the work. Specifically:
> > > 1. EUREKA: HUMAN-LEVEL REWARD DESIGN VIA CODING LARGE LANGUAGE MODELS has already demonstrated the capability of leveraging program synthesis by state-of-the-art LLMs to address both discrete and continuous black-box optimization tasks. This overlaps significantly with the primary claim of novelty in the present work.
> > > 2. Another related work, Learning to Learn Faster from Human Feedback with Language Model Predictive Control, similarly utilizes LLMs for program generation, integrates a simulator, and employs a control feedback loop. While there are some differences in approach, the fundamental concepts appear closely related to those presented here.
> > >
> > > Given these prior works, I am not fully convinced that the claim of novelty holds, particularly regarding the statement that "leveraging the program synthesis capabilities of state-of-the-art LLMs enables our method to effectively handle both discrete and continuous black-box optimization tasks." This claim does not seem sufficiently distinct from prior works to justify its novelty.
> > >
> > > That said, I acknowledge the authors' effort in addressing my earlier concerns. While I will raise my score slightly in recognition of this, the lack of a compelling argument for novelty prevents me from raising it above the acceptance threshold.

---

> > > > ### Author Response · Authors · 2024-12-02
> > > > **Response to Reviewer THXz**
> > > >
> > > > Dear Reviewer THXz,
> > > >
> > > > We are truly thankful to the reviewer for acknowledging our contributions and raising the score. Please see our responses to the Reviewer's new comments below.
> > > >
> > > > *1. EUREKA: HUMAN-LEVEL REWARD DESIGN VIA CODING LARGE LANGUAGE MODELS has already demonstrated the capability of leveraging program synthesis by state-of-the-art LLMs to address both discrete and continuous black-box optimization tasks. This overlaps significantly with the primary claim of novelty in the present work.*
> > > >
> > > > Thanks again for asking us to clarify this part of our response better. As we note in our previous comments to the reviewer, we have referenced and contrasted against Eureka in our revision and described the essential differences.
> > > >
> > > > * We note that Eureka is not claimed as a black-box optimization model and thus do not compare to any of the standard black-box optimization schemes such as BO or CMA-ES as we do. However, it does use program synthesis in an iterative manner as we do, but differently. Specifically, as the Eureka paper writes "given executable reward functions from an earlier iteration, Eureka performs in-context reward mutation, proposing new improved reward functions from the best one in the previous iteration. Concretely, a new EUREKA iteration will take the best-performing reward from the previous iteration, its reward reflection, and the mutation prompt as context and generate K more i.i.d reward outputs from the LLM".
> > > > * Our problem setup is very different from Eureka where we do not use any training within the optimization loop and thus our setup is much simplified offering a cohesive closed-loop LLM-simulator black-box module, the output from which may be used in  a downstream task (e.g., solving the QA task).
> > > > * We use an implicit analysis-by-synthesis of the synthesized program within a simulator for computing the fitness, instead of using a fitness that also potentially incorporate training dynamics of the fitness function as in Eureka.
> > > > * Unlike Eureka, we use the full optimization trajectory in our feedback to LLM, instead of a greedy approach of providing only the best solution from the previous iteration. We also experimented with the setting of providing only the best program thus far to LLMPhy and found inferior results (please see our results reported in Table 5).
> > > > * We also note that a key idea in our setup is that the LLM uses the world knowledge associated with the physics parameters to optimize, while this is not the case with Eureka, where the reward variables are not given any semantic meaning (as their task does not offer this possibility).
> > > >
> > > > We believe these are **important differences with the approach in Eureka**.
> > > >
> > > > *2. Another related work, Learning to Learn Faster from Human Feedback with Language Model Predictive Control, similarly utilizes LLMs for program generation, integrates a simulator, and employs a control feedback loop. While there are some differences in approach, the fundamental concepts appear closely related to those presented here.*
> > > >
> > > > Thank you for suggesting to look at this paper. We will include it in our revision. However, note that this work uses an LLM to convert human natural language feedback to produce reward code, which is then used in a MuJoCo based simulator to produce robot actions using a horizon trajectory optimization algorithm. This appears to be a different problem than what we are considering in our setup, where the LLM-simulator iterations are used to estimate unobservable parameters.
> > > >
> > > > Kindly let us know if we overlooked the contributions in these papers and missed any important details that the reviewer wanted us to make a note of. If so, please let us know and we will consider addressing them explicitly. We truly appreciate the great efforts taken by the reviewer in thoroughly reviewing our work and we will strive to make sure all your comments are clearly addressed.

---

### Official Review · Reviewer_2zB8 · 2024-11-03

**Soundness:** 3
**Presentation:** 3
**Contribution:** 2
**Rating:** 3
**Confidence:** 4

**Summary:**

This paper proposed a new framework combining Large Language Models (LLMs) with physics engines and optimization algorithms for complex physical reasoning. This approach cab infer and simulate multi-object dynamics under physical interactions. The authors propose a new dataset, TraySim, where objects on a tray are impacted by an external force, simulating dynamic, inter-object interactions.

**Strengths:**

This paper addresses an important question in AI by enabling large language models to perform complex physical reasoning, which is essential for real-world applications. The authors introduce a new framework, LLMPhy, that combines large language models with optimization algorithms and a physics simulator to estimate physical parameters and predict interactions. They also present a new dataset, TraySim, designed to benchmark models on multi-object physical reasoning tasks, making it a valuable resource for advancing AI’s capabilities in dynamic environments.

**Weaknesses:**

First, the paper lacks a discussion on its relationship with existing datasets like CoPhy https://arxiv.org/abs/1909.12000 and ComPhy https://comphyreasoning.github.io/, which also aim to infer physical properties of objects. These datasets offer benchmarks widely used in physical reasoning research, and a comparison of the proposed method on them would strengthen the validation of LLMPhy and clarify its distinct contributions.

Second, several established methods already tackle similar tasks, such as the approach presented in the ComPhy paper and methods like https://arxiv.org/abs/2012.08508 The absence of comparisons with these methods and relevant literature weakens the contextual positioning of LLMPhy within existing approaches.

Third, the methodology lacks clarity, especially regarding critical details like how multiview images are utilized to reconstruct object shapes and the specific inputs provided to the LLM. For example, if the LLM samples optimization variables based on physics knowledge from its training data, is object category information also supplied to guide this process?

Lastly, the unclear explanation of how raw RGB video data is converted into simulation inputs makes it difficult to assess how this approach could apply in realistic scenarios beyond controlled simulations. This raises concerns about the generalizability of LLMPhy to real-world applications.

**Questions:**

The authors may provide answers for my clarification questions about the method.

---

> ### Author Response · Authors · 2024-11-23
> **Response to Reviewer 2zB8**
>
> We thank the reviewer for the encouraging comments, identifying the importance of our approach and pointing out the omissions in our initial submission. We apologize that some of the results in the paper could not be incorporated in the previous draft. We have now thoroughly revised the paper catering to the reviewers' recommendations. We sincerely hope you could take a relook at the paper and let us know if you suggest other changes to improve this work.
>
> * *First, the paper lacks a discussion on its relationship with existing datasets like CoPhy and ComPhy, which also aim to infer physical properties of objects.*
>
> We apologize for the omission of these two papers, which we have now cited and contrasted in the Related Works. Note that both these papers consider the physical reasoning task in a supervised setting, while our goal is to bring out the physical reasoning capabilities of an LLM-simulator combination in a zero-shot setting, which is an entirely different goal. We also note that ComPhy uses a setup with two physical parameters (mass and charge) however our setup is inspired by the standard mass-spring-damping dynamical model. Also, given that we need the LLM to interact with the simulator in the optimization setup, unfortunately the datasets proposed in these benchmarks cannot be directly used to validate our setup.
>
> * *Second, several established methods already tackle similar tasks. The absence of comparisons with these methods and relevant literature weakens the contextual positioning of LLMPhy within existing approaches.*
>
> As noted above, ComPhy uses a supervised training framework and does not use a simulator in the optimization loop, as a result the methods are not comparable. We would also like to emphasize that in CoPhy and Comphy, the physical parameters are only implicitly estimated, while in LLMPhy the optimization estimates the exact physical values of these parameters from the implicit analysis-by-synthesis approach through minimizing the dynamical trajectory prediction error (please also see Figure 5c). Further, the key technical focus in LLMPhy is to demonstrate the zero-shot black-box optimization capabilities of LLM-simulator combination, to evaluate for which we compare against other standard black-box methods, demonstrating solid performances.
>
> * *Third, the methodology lacks clarity, especially regarding critical details like how multiview images are utilized to reconstruct object shapes and the specific inputs provided to the LLM. For example, if the LLM samples optimization variables based on physics knowledge from its training data, is object category information also supplied to guide this process?*
>
> We thank the reviewer for asking these details. In Figure 3, we have elaborated on our architecture and provided a simplified program. Further, Figure 7 and 8 in the Appendix provides details of a LLMPhy prompt that is sent to the LLM. In Appendix Sections I, J, K, and L, we have included full optimization traces, feedback, and interactions between the LLM and the simulator. We sincerely hope the reviewer can take a look at our additional results in the Appendix, that we believe could add clarity to understand our method better. Kindly let us know if there are additional details that we could provide to improve the paper further.  *Is object category information also supplied to guide this process?* Yes, the object category is also provided as part of the program in Phase 1.
>
> * *Lastly, the unclear explanation of how raw RGB video data is converted into simulation inputs makes it difficult to assess how this approach could apply in realistic scenarios beyond controlled simulations. This raises concerns about the generalizability of LLMPhy to real-world applications.*
>
> We assume the Phase 1 of LLMPhy allows complete access to the objects, including multiview videos, as well as tracking the object motions. In the real world, this is not necessarily a problem as one may use AprilTags for tracking the objects. Currently, doing real-world experiments is muddled due to the need to estimate the external impact force precisely, which will need programming a robotic controller to interact with a pusher; this will need additional programming efforts and we hope to look at this problem in a separate paper.

---

> > ### Author Response · Authors · 2024-11-30
> > **Rebuttal and Additional Questions?**
> >
> > Dear Reviewer 2zB8,
> >
> > Thank you so much for the feedback you provided on our initial submission. We have thoroughly revised the paper as per the reviewers' comments. We are summarizing below the changes we incorporated into the paper to address your questions.
> >
> > * We have included references to CoPhy and ComPhy in the Related Works, explaining the differences to our method.
> > * We have thoroughly revised the paper to describe the task setup, with illustrations explaining the pipeline better (Figure 3, Appendix / Sec D).
> > * Added a pseudo-realistic experiment (Table 5) to validate the robustness of the method to noise.
> >
> > As the rebuttal window is closing soon, could you kindly take a look at our revision and our responses to your questions above, and let us know if you have additional concerns that we could address?
> >
> > Thank you,
> >
> > Authors of LLMPhy

---

> ### Comment · Reviewer_2zB8 · 2024-12-01
>
> Thank you for the authors’ detailed responses and updates to the manuscript. After carefully reviewing the revised materials, I have decided to maintain my original scoring on this paper for the following reasons.
>
> First, while the authors clarified their relationship to existing work (e.g., ComPhy and CoPhy), one of the contributions of the paper becomes somewhat ambiguous. Specifically, it is unclear whether the introduction of the TraySim dataset—"a synthetic multi-view dataset with 100 scenes for zero-shot evaluation"—should be considered a primary contribution.
>
> Second, the characterization of the approach as “with supervision” now appears to me to be inconsistent with its methodology. The approach requires additional scene annotations at test time (e.g., object shape models) and access to a predefined simulation interface. These requirements indicate a degree of supervision, and the claim of being “unsupervised” seems overstated. Moreover, related work, such as https://arxiv.org/abs/2111.00312 (2021) and http://vda.csail.mit.edu/ (2017), which also leverage known object models and simulations, should be explicitly discussed and compared to contextualize the contribution of this paper.
>
> Overall, while the proposed method is technically sound, it relies on strong assumptions about perception (e.g., full 3D models of objects) and simulation (e.g., alignment between the simulation engine and the test data). These assumptions limit the generalizability and broader applicability of the method. Additionally, as noted by other reviewers, the inference process is slow due to the involvement of a large language model in the reasoning loop. Addressing these limitations or discussing potential solutions would enhance the paper’s impact.

---

> ### Author Response · Authors · 2024-12-02
> **Response to Post-Rebuttal Comments by Reviewer 2zB8**
>
> Dear Reviewer 2zB8,
>
> We are grateful to the reviewer for going through our revision and acknowledging the technical soundness of our contribution. Please find below our responses to the reviewer's additional questions.
>
> *1. It is unclear whether the introduction of the TraySim dataset—"a synthetic multi-view dataset with 100 scenes for zero-shot evaluation"—should be considered a primary contribution.*
>
> We want to emphasize that our **primary contribution is the algorithm for black-box optimization using LLM-simulator iterations**. In order to demonstrate the effectiveness of our algorithm, we make a secondary contribution by introducing the TraySim dataset that offers a very complex scenario for physical parameter estimation and reasoning. The key reason for crafting the new dataset is because our setup uses program synthesis for optimization and as the reviewer might know, the functionalities of the APIs used in such synthesized programs need implementations for their use in the simulator (this is a standard practice, e.g., Visual Programming, CVPR 2023 or ViperGPT, ICCV 2023). Such an API-simulator integration is unavailable in any existing dataset that we are aware of. We will be publicly releasing our dataset and all the associated elements upon acceptance of our paper so that our setup will be useful to the community in extending our approach.
>
> *2. The characterization of the approach as “with supervision” appears to me to be inconsistent with its methodology. The approach requires additional scene annotations at test time (e.g., object shape models) and access to a predefined simulation interface. These requirements indicate a degree of supervision, and the claim of being “unsupervised” seems overstated.*
>
> We explicitly acknowledge that we make the following assumptions when demonstrating the efficacy of our approach, which *we strongly believe are very reasonable assumptions to make* in order to demonstrate our approach. We assume that:
> * The simulator has access to CAD models of the objects (in reality the simulator is stand-in for a robot interacting with the scene)
> * The physics engine is capable of emulating the impact dynamics of each of the object classes
> * The LLM/LVLM is knowledge about the world and its physics
>
> Further, when we say our method is unsupervised or zero-shot, we assume our method does not “learn” the distribution of the inference variables from a training set to be applied to the test set, instead every test sample is independent, which we believe **is consistent with the standard definition of “zero-shot inference”**. If the reviewer thinks these assumptions are unreasonable, we are happy to adjust the wording in the final paper, to what the reviewer consider as appropriate.
>
> *3. Moreover, related work, such as https://arxiv.org/abs/2111.00312 (2021) and http://vda.csail.mit.edu/ (2017), which also leverage known object models and simulations, should be explicitly discussed and compared to contextualize the contribution of this paper.*
>
> Thank you for suggesting these additional references. We have carefully read them and will be citing them in the revised paper. We find that the first paper  [a]  the reviewer mentions deals with the problem of 6D pose estimation from RGB-D images using inverse graphics and does not appear as directly related to our work. For the second paper [b], we are in fact citing a related work [c] in the revision. As we wrote in the related works, our paper is inspired by [c] (and thus to [b] as well), however in our work, we attempt to go beyond these prior works by using more complex physical reasoning setups and leveraging knowledge and reasoning capability of LLM-simulator combinations. Unfortunately, as our method requires implementation of the simulator APIs, we cannot compare to the experimental setup in these prior works, as the reviewer suggests. We also find that [a] uses the YCB-video dataset for evaluation and does not involve a simulator, while [b] uses a simulator, however *we could not find their dataset publicly*.
>
> [a] 3DP3: 3D Scene Perception via Probabilistic Programming, Gothoskar et al., NeurIPS 2021
>
> [b] Learning to see physics via visual de-animation, Wu et al., NeurIPS 2017
>
> [c] Neural scene de-rendering, Wu et al, CVPR 2017
>
> *3. The inference process is slow due to the involvement of a large language model in the reasoning loop. Addressing these limitations or discussing potential solutions would enhance the paper’s impact.*
>
> We acknowledge that the inference process is slow at the moment due to the need to invoke an external closed-source LLM. However, looking at it optimistically, we believe such issues may be handled in the time to come and should not impact the scientific merit and usefulness of our work.

---

> > ### Comment · Reviewer_2zB8 · 2024-12-03
> >
> > A few quick clarifications:
> > 1. Yes, I understand that. However, in the manuscript, the dataset is still listed as one of the contributions in the introduction.
> > 2&3. To clarify, part of the reason many existing papers require supervision for object properties, etc., is precisely because they lack access to ground-truth object shapes at test time (along with other properties). If shape information is available, other Bayesian optimization frameworks could be applied to infer a few physical parameters, similar to the approach in the Wu et al. (2017) paper and 3DP3 (yes, it's for static scenes but the similar framework should work for videos as well). I believe a comparison with a similar framework should be included.
> >
> > I think it’s crucial to describe and address the assumption about object models. First, it is very unnatural to consider a scenario where object shapes are available at test time but their physical properties are not. Second, the current manuscript does not clearly explain how this assumption could be eliminated.
> >
> > For these reasons, I maintain my judgment.

---

### Official Review · Reviewer_sHax · 2024-11-04

**Soundness:** 3
**Presentation:** 3
**Contribution:** 2
**Rating:** 5
**Confidence:** 4

**Summary:**

The paper introduces a framework called LLMPhy, which combines large language models (LLMs) with physics engines to tackle complex physical reasoning tasks. The proposed method is tested on a new dataset, TraySim, where the model predicts object stability and interactions on a tray after an impact. LLMPhy operates in two phases: parameter estimation and simulation, with the LLM synthesizing hypotheses and the physics engine verifying them in a feedback loop.

Key Strengths:

TraySim Dataset: A tailored benchmark for multi-object physical interactions.

Zero-Shot Reasoning: Achieves notable improvements over Bayesian optimization on complex tasks.

Limitations:

Limited Parameter Scope: Currently models only four physical attributes, reducing generalizability.

No Real-World Testing: Results are based solely on simulations, lacking validation in physical settings.

**Strengths:**

The paper proposes a black-box optimization framework that uses In-context learning to enhance the physical reasoning skills of SOTA LLMs.To augment the reasoning, the framework leverages the coding capability of LLMs and give it access to a simulator. This paper is a valuable contribution to physical reasoning using LLMs, presenting an original approach that effectively combines the strengths of language models and physics-based simulations. The results are promising, though some improvements in scalability, dataset credibility, and validation across physical engines would strengthen the framework’s applicability and robustness.

**Weaknesses:**

(1) my first concern is the limited physical parameters, the model currently explores only four physical parameters, which restricts its application to broader and more complex real-world scenarios. This constraint is acknowledged by the authors but remains a significant limitation for potential applications in varied physical environments.

(2) Dataset Scale and Generalization: With only 100 sequences, TraySim may be limited in capturing the full complexity of physical interactions. Expanding this dataset or testing LLMPhy on existing physics-based benchmarks could provide a more comprehensive understanding of the model’s generalization ability.

(3) Absence of Real-World Validation: While the simulation results are promising, real-world validations or experiments in a physical setup would enhance the credibility of the proposed method. Without real-world tests, it remains uncertain how well the model's inferred physical parameters would translate to actual physics.

**Questions:**

(1) how would LLMPhy perform when scaled to more complex environments with additional physical parameters? Besides, since different physics engines can have unique methods for simulating physical interactions, how might the choice of simulator (e.g., using MuJoCo vs. others) affect the model’s predictions and the general applicability of the method?

(2)  the simulator access setup is interesting, if applicable to real-world tasks, we can use LLM augmented with simulators to perform physical reasoning in real-world. However, the paper is assuming a perfect simulator  that can simulate real-world environment, which makes the results less interesting. It would be nice to add such experiments (studying how sim2real gaps will impact the evaluation)
Since the paper only use synthetic dataset to evaluate, this raise the question if the method applies to real-world robotics settings

(3) Reproducibility with Open-Source LLMs: Given that GPT-4o is closed-source, how feasible is it to adapt LLMPhy to open-source alternatives, and what changes would be necessary to maintain accuracy?

(4) Can the authors provide insights into how the framework might handle situations where parameter estimation fails due to high variability in initial conditions or other environmental factors?

---

> ### Author Response · Authors · 2024-11-23
> **Response to Reviewer sHax**
>
> We thank the reviewer for the encouraging comments and pointing out the omissions in our initial submission. We apologize that some of the results in the paper could not be incorporated. We have now thoroughly revised the paper catering to the reviewers' recommendations. We sincerely hope you could take a relook at the paper and let us know if you suggest other changes to improve this work.
>
> * *limited physical parameters, the model currently explores only four physical parameters, which restricts its application to broader and more complex real-world scenarios.*
>
> We would like to emphasize that each object class in our dataset has 4 physical parameters. We have 3 object classes in the TraySim dataset, which amounts to a total of 12 unique parameters estimated in Phase 1. We note that Phase 2 of LLMPhy works with 5-9 object instances, and a significantly larger state space for layout estimation (9 instances, 9 locations, and 3 classes, and 10 colors). That being said, we do understand your concern and were curious to find out how the performance of LLMPhy would be with more object classes.  In the revised paper, Appendix F, we have included additional experiments using 5 different object classes instead of 3, each class having its own physical parameters (and thus 20 parameters to infer instead of 12). Our results using this extended dataset, as reported in Table 5, show that LLMPhy still performs best against alternatives, thus demonstrating the scalability of our approach.
>
> * *Dataset Scale and Generalization: With only 100 sequences, TraySim may be limited ...testing LLMPhy on existing physics-based benchmarks...*
>
> Thanks for this point. We used 100 sequences for two reasons: i) the physical parameters are all randomly sampled for all the object classes and as there is no training involved, this is a sufficiently large set to evaluate a zero-shot task, and ii) the LLMs we use in LLMPhy are expensive, and as you may see from the optimization program traces provided in Appendix/Sections I,J,K,L, the programs grow in size quadratically with each iteration. The LLM that we currently use (o1-mini) does not provide a memory from previous iterations and thus the full optimization trace so far needs to be provided for every step, which increases the number of tokens used and the cost. We hope to consider these aspects in a future version of this paper and keep the focus of this paper to demonstrating the feasibility of black-box optimization using an LLM. *Why not use other physics-based benchmarks?* This is because we use APIs in the synthesized programs that are implemented to talk to our specific simulator; such a setup is not currently available within other benchmarks. While, we have abstracted the APIs sufficiently well (as depicted in Figure 3) making the LLM part of LLMPhy agnostic to the simulator, we still need implementations of these APIs, which is not available in existing benchmarks.
>
> * *Absence of Real-World Validation*
>
> Thanks for this important question. The main difficulty to have a real world validation for our setup is to get the external forces computed effectively, for which we need to program a robotic controller; this will demand significant efforts and is better dealt with in a separate paper. The focus of the current paper that we desire is, to be on the feasibility of conducting black-box optimization for physical reasoning using an LLM, and for which we hope simulation experiments would be an initial step.
>
> That being said, we attempted a pseudo-real-world experiment by adding noise to the simulations, to capture a "noisy" real-world. This experiment is detailed in Appendix/Section F/Table 5, where the feedback trajectory error from the simulator was corrupted by random noise (producing a large ~25% change to the actual error) before being providing as feedback to the LLM. Our goal is to see how robust the LLM reasoning is and how the behavior changes if the simulation is erroneous e.g., if the experiment was done in the real world. Our results in Table 5 show that the LLMPhy performance is sufficiently robust and the performance does not degrade significantly (down by ~6%), and continues to show better results against Bayesian Optimization and CMA-ES. While we agree that this result is still in simulations, we believe our results do provide some confidence towards a real-world validation.

---

> ### Author Response · Authors · 2024-11-23
> **Response to Reviewer sHax (2)**
>
> * *Besides, since different physics engines can have unique methods for simulating physical interactions, how might the choice of simulator (e.g., using MuJoCo vs. others) affect the model’s predictions and the general applicability of the method?*
>
> In Figure 3 in the revised paper, we have shown the structure of the program that is synthesized in LLMPhy. As may be noted, the LLM interacts with the simulator using API calls of the form sim.create_object() where the spatial attributes and the physical parameters of the object instances are supplied. The actual implementation of the create_object() function is entirely abstracted out from the LLM and the LLM is agnostic to the underlying simulator used. Thus, even if we replaced MuJoCo with any other simulator, the interaction of the LLM with the simulator should not be affected. That being said, implementing the entire setup using another simulator is significant work and is not feasible quickly as the actual implementation of these APIs will need to be redone. However, as noted above, we see that the LLMPhy reasoning is quite robust and thus even if a different simulator introduces additional artifacts or noise, our results show that the performance may not be impacted much as long as the simulator is able to produce reasonable physical simulation.
>
> * *The paper is assuming a perfect simulator that can simulate real-world environment, which makes the results less interesting. It would be nice to add such experiments (studying how sim2real gaps will impact the evaluation)*
>
> Thanks for emphasizing this comment. Sim2Real experiments are indeed an interesting direction and may need a dedicated focus as we may need to understand, accommodate, and capture many other physical parameters in the real-world and environment, which is solid scientific problem, and could be considered in a dedicated paper.
>
> * *Reproducibility with Open-Source LLMs: Given that GPT-4o is closed-source, how feasible is it to adapt LLMPhy to open-source alternatives, and what changes would be necessary to maintain accuracy?*
>
> We did try LLMPhy using open-source models, including LlaVa-1.6, however, found that the programs generated were not executable without errors. So far we have seen o1-mini and o1 to produce correct programs without needing many restart steps (we restart the iteration step for LLMPhy when a synthesized program fails), while GPT-4o synthesizes useful programs with some restarts (~15% of time). We hope future open-source LVLMs will have better program synthesis abilities and may be useful in LLMPhy.
>
> * *Can the authors provide insights when parameter estimation fails due to high variability in initial conditions*
>
> As we noted, our experiments in Section F/Table 5 show that the LLMPhy estimations are fairly robust to perturbations. From the LLMPhy Phase 1 optimization traces provided in Sections I and K (Appendix) show that the LLM is essentially using the change in the heights of the object instances as inferred from the object trajectories to find if they are upright or falling after the collision, and adjusting the physics parameters using their physical semantics (e.g., friction, stiffness, etc.). We assume a closed system in our study, and if there are other environmental factors that may influence the solution (e.g., the tray is on a slope or on an uneven surface), we believe the method should still work if these aspects are explicitly captured in the simulator. Please also see the **video provided in the supplementary materials that may provide insights into the complexity of our current setup**.

---

> > ### Author Response · Authors · 2024-11-30
> > **Rebuttal and Additional Questions?**
> >
> > Dear Reviewer sHax,
> >
> > Thank you so much for the feedback you provided on our initial submission. We have thoroughly revised the paper as per the reviewers' comments. We are summarizing below the changes we incorporated into the paper to address your questions:
> > * New experiments with more parameters to be estimated in Phase 1 (Table 5)
> > * A pseudo-realistic experiment to validate the robustness of the method to noise (Table 5)
> > * Added experiments to other LLMs, however using open-source LLMs appear to be infeasible at the moment.
> > * We have also clarified on the practical feasibility of the setup, data size and generalization, as well as the scalability of the approach.
> >
> > As the rebuttal window is closing soon, could you kindly take a look at our revision and our responses to your questions above, and let us know if you have additional concerns that we could address?
> >
> > Thank you,
> >
> > Authors of LLMPhy

---

### Official Review · Reviewer_e7Sm · 2024-11-04

**Soundness:** 2
**Presentation:** 3
**Contribution:** 2
**Rating:** 3
**Confidence:** 4

**Summary:**

The paper proposes a task, dataset, and solution for an LLM-prompting based physics parameter estimation and trajectory estimation using privileged simulator feedback in the loop.

**Strengths:**

The authors attempt a novel task for improving low-level physical reasoning using an LLM, such as predicting physics parameter or object trajectories, instead of most prior works (to the best of my knowledge) do very high level semantic physical reasoning with LLMs. I believe this line of work is quite interesting, but the paper is needs more work both in terms of experimental study and presentation. See comments / suggestions below.

**Weaknesses:**

1. The abstract doesn’t differentiate proposed work with existing literature and justify the need of a new dataset/method.
2. Writing is a slightly unstructured and somewhat informal. For instance, L110: ‘..leads to significant improvements in performance against alternatives, including using Bayesian optimization and solely using only LLM for..’ would read more formal for a research paper as ‘leads to x% improvements over baseline methods that use only LLM or Bayesian optimization for..’. The current statement is unclear as to weather these are two different baselines or not. The use of word ‘significant’ without any concrete number is not standard for a research paper. L123: ‘Our experiments using LLMPhy on the TraySim dataset demonstrate promising results’, same issue with the word ‘promising’ - it is vague and not very concrete. L192: ‘Our attempt to solve the…’ - is grammatically incorrect and not sure what’s the meaning of the statement. These are just a few examples but for me most the paper reads like an initial draft and some writing iterations will significantly improve it’s presentation and understanding. Also, organizing the text in subsections (in Section 2, for instance) will improve the presentation. L325: ‘X is the input data, and Λ is the desired result.’ - very ambiguous notation definition. Major writing revision is needed.
3. There’s a claim that the method is generalizable to any simulation however results for only one simulation is presented. If the claim to true, the method application on other existing simulations should be shown as well. Or the authors should discuss what are the challenges for doing so.

**Questions:**

1. A qualitative example should be provided in the main paper instead of just in supplements to ground the method for the reader.
2. How will this method generalize to real world if ground truth values from the simulator are used?
3. What are plots submitted in supplementary supposed to mean, they are neither referenced in the main of supplementary pdfs.

---

> ### Author Response · Authors · 2024-11-23
> **Response to Reviewer e7Sm**
>
> We thank the reviewer for the encouraging comments and appreciating the novelty. We have thoroughly revised the paper as per the reviewers’ suggestions. We sincerely hope you may be able to have a look at our revision, in which we have incorporated all the suggestions you and other reviewers have provided.
>
> * *The abstract doesn’t differentiate proposed work with existing literature and justify the need of a new dataset/method.*
>
> Thanks for this point. To our knowledge, there is no prior work that attempts to bring out the  LLM/LVLM abilities for solving both continuous and discrete black-box optimization problems like we do in this paper. We have modified the abstract to emphasize this point. We have further elaborated on this aspect in the Related Works section where we have clearly delineated what has been done earlier and how we differ from prior works, and the importance of the differences.
>
> * *Writing is a slightly unstructured and somewhat informal.*
>
> Thanks for mentioning this. We have rephrased the sentence in the Introduction as follows: “​​Our results on TraySim show that LLMPhy leads to clear improvements in performance against alternatives on the QA task (+2.5% accuracy), including using Bayesian optimization, CMA-ES, and solely using an LLM for physical reasoning (+12%), while demonstrating better convergence and estimation of the physical parameters.”
>
> * *Also, organizing the text in subsections ... Major writing revision is needed.*
>
> We did try our best efforts at improving the quality of our work in the revision. We appreciate if you could kindly have a relook at our revision, which has more results, comparisons, ablation studies, qualitative examples, program synthesis examples, and optimization traces, including comparisons to other LLMs.
>
> * *There’s a claim that the method is generalizable to any simulation however results for only one simulation is presented.*
>
> Please have a look at Figure 3 in the revised paper in which we have shown the structure of the program that is synthesized in LLMPhy. As may be noted, the LLM interacts with the simulator using API calls of the form sim.create_object() where the spatial attributes and the physical parameters of the object instances are supplied. The actual implementation of the create_object() function is entirely abstracted out from the LLM and the LLM is agnostic to the underlying simulator used, i.e., it is a true black-box optimization model. However, implementing the setup using another simulator is a significant effort and is not feasible quickly as the implementation of these APIs will need to be redone.
>
> To further understand the impact of a “noisy” underlying simulator, we in fact did an experiment in the Appendix/Section F/Table 5, where the feedback trajectory error from the simulator was corrupted by random noise before being provided to the LLM. Our goal is to see how robust the LLM reasoning is and how its behavior changes if the simulation is erroneous. Our results in Table 5 show that LLMPhy performance is sufficiently robust and do not degrade significantly (- ~6%), and continues to show better results against Bayesian Optimization and CMA-ES.
>
> * *A qualitative example should be provided in the main paper.*
>
> We have included several qualitative examples in Figure 4 and Figure 13. We have also included video sequences showing our dynamical setup in the supplementary materials.
>
> * *How will this method generalize to real world if ground truth values from the simulator are used?*
>
> While the ground truth trajectories (which are the *observable* quantities) from the simulator are used in Phase1 estimation (for the estimation of the unobservable physical quantities), we believe this is not a deterrent to demonstrating the usefulness of our approach in the real world. For example, for real world experiments, one may easily use AprilTags for tracking the center of gravity of the object instances. We also assume to have full access to the Phase 1 auxiliary sequences, including multiple views as well as object segments. Thus, we think our assumption is not weakening our contributions in any manner. The reviewer may ask why then not show real-world experimental results? The main difficulty is to get the external forces computed effectively, for which we need to program a robotic controller; this will demand significant efforts and is better dealt with in a separate paper.
>
> * *What are plots submitted in supplementary supposed to mean?"
>
> We apologize that we did not get a chance to clearly explain the plots in the supplementary material at the time of the initial submission. We have added the explanations in the revised paper. Please see the captions of Figures 10, 11 and 12 in the Appendix. Further, in Appendix D, we have also included details of the prompts to the LLMs and highlighted the key elements in the prompt. We hope you can take a look at the revised explanations and let us know if you need further clarity on any aspect.

---

> > ### Author Response · Authors · 2024-11-30
> > **Rebuttal and Additional Questions?**
> >
> > Dear Reviewer e7Sm,
> >
> > Thank you so much for the feedback you provided on our initial submission. We have thoroughly revised the paper as per the reviewers' comments. As the rebuttal window is closing soon, could you kindly take a look at our revision and our responses to your questions above, and let us know if you have additional concerns that we could address?
> >
> > We are summarizing below the changes we incorporated into the paper to address your questions:
> > * We have updated the abstract and fixed the issues you pointed out in the Introduction.
> > * Added an additional result (Table 5) to validate the robustness of our method, imitating a real-world adaptation of the method.
> > * Added a qualitative example to the main paper (Figure 4) and more examples (Figure 13).
> > * Explained how the trajectories in the real-world could be obtained in a similar manner as in the simulations.
> > * Incorporated detailed explanations of the contents of the supplementary materials into the Appendix, as well as included additional qualitative videos demonstrating the setup and comparisons to other methods.
> >
> > Thank you,
> >
> > Authors of LLMPhy

---

> > > ### Comment · Reviewer_e7Sm · 2024-12-01
> > > **post rebuttal score update**
> > >
> > > Thanks! I have updated my score based on the revision. I still believe one simulation domain is not sufficient to claim a black-box approach - since it may be significant effort or not that straightforward to prompt engineer when applying it to another simulator, hence I will retain my soundness score.

---

> > > > ### Author Response · Authors · 2024-12-01
> > > > **Response to Post-rebuttal Comments**
> > > >
> > > > We thank the reviewer for the comment. While it is certainly a significant effort to study and  build our entire, complex framework in another simulator to demonstrate its efficacy, we note that MuJoCo is one of the best simulators that do capture multi-body rigid and soft-body dynamics with high accuracy, with physics rendered close to the real world. As noted in Appendix A, any physics engine that is capable of computing the forward dynamics of a multi-body system can be integrated within our framework.
> > > >
> > > > * Further, as explained in our revision, our simulator is disconnected from the LLM through API calls, and the LLM does not make any assumption on which simulator is used (note that the name of the simulator is not mentioned in the LLM prompt anywhere). To demonstrate this, we added random noise of large magnitude to the output trajectories produced by the simulator when feeding back the response to the LLM for the next iteration; our results in Table 5 shows that the LLM still behaves in a useful manner, clearly demonstrating that the LLM is not using any knowledge of the physics engine for optimization, and is **truly black-box**.
> > > >
> > > > * In the experiments in Table 5, we have also included additional objects with distinct physics attributes (flute glass and champagne glass); our experiments show that our model continues to demonstrate state-of-the-art performance, again showing that **changes in the physics of the objects do not change the reasoning power of the model**.
> > > >
> > > > * We also want to point out that our task in *Phase 2 is an entirely different setup from Phase 1 of inferring the layout of the scene*, and it may be considered as a **different experiment using a different physics engine from the setup in Phase 1**. For this setup as well, we show that using our iterative approach in LLMPhy (which is our core contribution described in Algorithm 1) shows significantly better results than using a single iteration (Table 2).
> > > >
> > > > Overall, we sincerely believe we have provided enough evidence in our revised paper that clearly demonstrates that our approach is agnostic to the underlying simulator. To accommodate the reviewer’s comments, we can further *incorporate additional experiments using a different simulator in the final revision of this paper*, if the paper is accepted. We hope the reviewer will re-consider the rating provided.

---

### Official Review · Reviewer_4AdL · 2024-11-06

**Soundness:** 2
**Presentation:** 4
**Contribution:** 3
**Rating:** 6
**Confidence:** 4

**Summary:**

This paper addresses the challenge of reasoning about the result of a complex multi-body physical interaction; specifically, the goal is to predict which objects will remain upright on a tray after an external force is applied by a pusher mass. This prediction is done by prompting an LLM to answer a multiple choice question about upright objects given the context of a synthesized video sequence of the multi-body physical interaction produced by a physics simulator. The key contribution of LLMPhy is an iterative framework between an LLM and the physics simulator to infer unknown physical properties of the objects in the scene, thereby allowing the physics simulator to synthesize the aforementioned video sequence.

The physical properties of the object categories are inferred by having the LLM generate programs for the simulator that attempt to model a reference dataset of videos of objects experiencing a different external force than the test set. The generated programs essentially consist of LLM-generated values for the unknown physical parameters, LLM-generated scene layouts that attempt to match those in a given reference video (object class, object location, and object color for each object in the video), and executable code for the physics simulator. This inference process is iterative and uses the error between the reference video's object trajectories and the simulated object trajectories as feedback to the LLM in order to generate updated (and ideally better) values of the unknown physical properties and a better scene layout. Once the values of the physical properties of the objects are estimated through this optimization process, they are applied to new scenarios involving more objects and a different external force; given only the first frame of a video of an interaction and the inferred object properties, an LLM generates a program for the simulator which then synthesizes a video of the interaction. This video is the video mentioned above that is provided as context to an LLM to answer a multiple choice questions about which objects will remain upright.

In addition to the LLMPhy framework, this paper also introduces a novel task and dataset.

**Strengths:**

- The paper is well written and organized, and the method is clearly explained and reasonably illustrated for further clarity.

- Both the problem and the proposed method are thoroughly mathematically formulated, which further helps with making the method understandable.

- The problem being addressed, namely reasoning about the result of complex multi-body physical interactions, is challenging and is relevant for enabling important future embodied capabilities.

- I like the ablation over the source of the physical parameter values (Random and Ground Truth), and I agree with the conclusions drawn from that ablation.

**Weaknesses:**

- The experiments are limited. Only 3 objects categories are included in the experimentation, and the scenarios consist exclusively of a single external force being applied to a tray containing objects; further, the reasoning question that is being answered is relatively simple (which objects remain upright) compared to the kind of information one might want for practical uses, such as the pose of the objects.

- Only GPT4o is used for the LLM, which makes any extrapolation or empirical trend to other models imposible.

**Questions:**

I am recommending a weak rejection (marginally below acceptance) at this point primarily due to the limitations of the experiments. I think the paper is very well written with a robust technical description, but it is hard to draw conclusions from the experiments, preventing the paper from being a sufficiently significant scientific contribution. While I also think that the specific physical reasoning being evaluated is simple (which objects remain upright), more comprehensive experimentation would make this paper acceptable. To be clear, its potential impact is bounded by this simplicity, but I think the problem is still sufficiently interesting and challenging.

Here are some actionable additional experiments that would certainly help improve my recommendation:
- Introduce more object categories that exhibit sufficiently distinct physical behaviors after interaction
- Allow multiple sizes of the tray during testing to investigate having more objects and different configurations of objects.
- Test multiple initial positions and velocities for the pusher; as I understood, only one initial position and velocity was used in the tests (line 403-405).
- It would be better to include more LLMs for evaluations, ideally at least another SOTA black-box model and a SOTA open model. However, this can be prohibitive due to cost, so this is a lower priority than other experimental improvements as long as it is clearly acknowledged that conclusions about LLMs more generally are not possible.

I would also be interested in an ablation that provided ground-truth scene layouts during the optimization process in order to better understand the optimization performance of the LLM with respect to just the physical properties.

Here are some questions that I would appreciate having answered:
- What is the format of the multiple choice answers?
- Does the model exhibit consistency when presented with the inverse question, i.e., which objects will be knocked over?
- Why does each object category have the same allowable tilt \alpha?
- While Figure 3 shows the trajectory error, I wonder about the error for each individual parameter value; it seems like this should be available and might be interesting. Do the individual physical properties pose different difficulty for the LLM to infer?

Here are some minor comments (these had no impact on my recommendation):
- Since images are provided as input in some phases, it would be good to clarify that the LLM in question is multi-modal. Some readers may expect the acronym VLM for models that can also accept images as input. This is stated, but I recommend being proactive about avoiding any confusion about what the input is and how it is encoded.
- Figure 3 x-axis intervals should be integers
- missing space: "atleast" line 396
- double word: "aproaches that that use" line 460
- mispelling: "differnetiability" line 487
- Recommend a general grammatical pass, several other minor errors
- It would be good to write out PSNR (line 427)

---

> ### Author Response · Authors · 2024-11-23
> **Response to Reviewer 4AdL**
>
> We are thankful to the reviewer for the insightful feedback and listing out additional experiments to bring out the advantages of our proposed method. In the revised submission, we have made a sincere and best possible effort in addressing all the questions that you have raised. We hope the revision addresses your concerns and if not please let us know additional experiments if any that we should further provide. Kindly see below the details of how we have addressed your concerns.
>
> * *The experiments are limited. Only 3 objects categories are included in the experimentation, and the scenarios consist exclusively of a single external force being applied to a tray containing objects?*
>
> We had used three objects in the Phase 1 of LLMPhy for three reasons: i) Tto understand and clearly demonstrate the feasibility of our approach towards characterizing the performance to standard methods, ii) reduce the cost involved, as more classes would demand a longer number of LLM optimization steps, that will quickly become very expensive, and iii) we wanted to imitate how a human would solve this physical estimation task; a typical way would be to isolate the objects and estimate the physics parameters for each object through interaction, which in our case will involve working with the three classes individually, which we thought would make the problem too simplified, instead decided to estimate the physics of three objects all together, i.e., our state space consists of 12 parameters. We note that Phase 2 of LLMPhy works with 5-9 object instances and a significantly larger state space for layout estimation, consisting of 9 object instances from 3 classes at 9 locations on the tray and may choose colors from 10 options.
>
> That being said, we do acknowledge your concern and were curious to find out how the performance of LLMPhy would be with more object classes.  In the Appendix F of the revised paper, we have included additional experiments using 5 different object classes instead of 3, each class having its own distinct physical parameters (except for mass, which makes the setup a bit more interesting). Our results using this extended dataset, as reported in Table 5, show that LLMPhy still performs better than alternatives.
>
> While, we use only a single external force to the tray, there are two aspects that we would like to emphasize here: i) there are a multitude of internal forces (friction, collisions, etc.) that are already in play within the closed system that makes the setup very challenging, and ii) once the physical parameters are determined in phase 1, the entire reasoning process on the task images happens within the simulator, which can use any number of external forces, changes to the tray sizes, pusher locations, etc., which as per our problem setup will be given as part of the question, and thus could be used in the simulator to produce the respective Phase 2 video. We request the reviewer to please have a look at the **video in the supplementary materials** that may help to understand the complexity of our problem setup better.
>
> * *The reasoning question that is being answered is relatively simple (which objects remain upright) compared to the kind of information one might want for practical uses, such as the pose of the objects.*
>
> We agree that the question structure we use in the QA task may look simple, however given the complexity of the problem setup, we believe answering the question is not that simple, given the velocity of the pusher and the layout of the objects in the scene vary, albeit significant variations in the unobservable physical parameters. To demonstrate the difficulty in reasoning and solving this task, we have **included video sequences from the TraySim dataset in the supplementary materials** that show the object instance collisions on impact, which we believe clearly show that the outcome is not deducible from the task images alone.
>
> * *Only GPT4o is used for the LLM, which makes any extrapolation or empirical trend to other models impossible.*
>
> Thanks for pointing this out. In the revised paper (Table 4), we have included additional results comparing o1-mini, o1-preview, and GPT-4o. Given the better performances using o1-mini as against GPT-4o in Phase 1 LLMPhy reasoning, we have updated our results using o1-mini in the tables (Table 1, 3, and 4). We attempted to use open-source models for solving the task, such as LlaVa-Next, however we found its program synthesis abilities were significantly inferior to GPT models. We hope future open-source models may be prove better.
>
> * *It is hard to draw conclusions from the experiments.*
>
> We apologize that the previous version of the paper lacked several important results. We have taken significant efforts to address the shortcomings and have thoroughly revised the experimental sections of the paper, including adding many results in the Appendix. We sincerely hope you may find the revised paper more complete.

---

> ### Author Response · Authors · 2024-11-23
> **Response to Reviewer 4AdL (2)**
>
> Additional Comments:
> * *Introduce more object categories that exhibit sufficiently distinct physical behaviors after interaction*
>
> Please see Table 5 in the Appendix.
>
> * *Allow multiple sizes of the tray during testing. Test multiple initial positions and velocities for the pusher.*
>
> As we noted above, in the test setup, we assume that apart from the arrangement of the object instances on the tray, all other physical attributes of the test scene are given. Thus, using multiple sizes of the tray or using more objects are technically not a bottleneck in our setup. That being said, there are practical issues (which may seem quite silly, however are serious limitations of the current state-of-the-art LVLMs) that prevent us from implementing this correctly. Specifically, we assume distinct colors for each of the object instances on the tray and using a larger grid will need more object colors, and each color should be clearly distinguishable from each other for the Phase 2 LLM inference. This limits the number of instances in Phase 2 to about 10. Having more object locations (beyond a 3x3 grid) and having fewer object instances also make the exact position of the instances difficult for the LLM to infer. For example, if there are only 5 instances in a 4x4 grid, with many empty spots, we found that the LLM is unable to distinguish if an empty spot is a gap between the objects or a missing object in the grid. These are limitations of the current state-of-the-art LVLMs and are not directly related to the demonstration of the feasibility or applicability of our optimization method. For a similar reason, the location of the pusher is assumed the same for all the problems, which will not affect generalizability of our approach, as any position for the pusher can be considered since the reasoning is entirely being simulated and pusher location is not an optimization variable.
>
> * *It would be better to include more LLMs for evaluations, ideally at least another SOTA black-box model and a SOTA open model. *
>
> Thanks for this point. In Table 4 in the Appendix, we report results with GPT-4o, o1-mini, and o1-preview.  We have also included in Table 1, results using CMA-ES, which is a state-of-the-art continuous black-box optimization model and show better results using LLMPhy. In Figure 5, we plot the convergence of various LLMs against Bayesian Optimization and CMA-ES. As noted above, we did not see open-source LLMs to produce useful and executable code for our optimization setup, and thus we could not use them.
>
> * *Ablation that provides ground-truth scene layouts during the optimization process*
>
> Thanks for pointing out this important experiment. We have added this result to Table 1 as well as to all other experiments (Table 3, 4). In Figure 5, we compare the actual physical parameters estimated using LLMPhy against their ground truth, and clearly show better estimation.
>
> * *What is the format of the multiple choice answers?*
>
> We have included Figure 4 in the revised paper that shows the precise format of the answers.
>
> * *Does the model exhibit consistency when presented with the inverse question?*
>
> We currently use the simulator to execute the Phase 2 program, which will produce a predicted test sequence, from which we extract the pose of each object instance in the last frame. Thus, as the pose in the simulator is deterministic, the inverse question will not change the accuracy; i.e., there is no reasoning in the extraction of the answer, instead the reasoning is in finding the correct parameters to run the simulation for the test sequence from the test images.
>
> * *Why does each object category have the same allowable tilt \alpha?*
>
> In our setup, we consider only tall and thin rigid objects with a uniformly distributed mass and thus we assume that this is a reasonable and conservative threshold, as beyond 45 degrees the object will fall off.
>
> * *While Figure 3 shows the trajectory error, I wonder about the error for each individual parameter value?*
>
> Thanks for this great question. Indeed, this is a very useful and insightful comparison. In Figure 5(c), we plot the average absolute error between the predicted and the ground truth for all the four physics parameters for the three objects. We find that LLMPhy shows the lowest error in the estimation of friction, stiffness, and inertia parameters, and closely matching with damping values ('damping' range is in (0, 10] as against others which are in (0,1]). We believe this result explains the better accuracy of LLMPhy in the QA task as reported in Table 1. Indeed, different physical parameters pose different reasoning difficulties. In Figure 6, we show the outcome of simulations when these parameters are changed while keeping other factors the same.
>
> We have also fixed all the typos you have pointed out. Thank you much for listing them.

---

> > ### Comment · Reviewer_4AdL · 2024-11-27
> > **Response to Authors' Rebuttal**
> >
> > Thank you for providing the additional experiments and for answering my questions. I have updated my recommendation accordingly.

---

> ### Author Response · Authors · 2024-11-27
> **Thank you!**
>
> Dear Reviewer 4AdL,
> We sincerely appreciate the time you have dedicated to reviewing our paper and providing detailed feedback. We are pleased to learn that you are satisfied with our revision and responses. Please let us know if there are any additional concerns or questions that we should address and any experiments that we should present results on.
>
> Thank you,
> Authors

---

### Author Response · Authors · 2024-11-24
**Summary of Revision**

We sincerely thank all the reviewers for their diligent efforts in reviewing our paper, for patiently identifying its shortcomings, and for providing detailed feedback on the necessary revisions. We are pleased that the reviewers recognized the novelty of our problem setup and the robustness of our approach. We acknowledge that the previous version of our manuscript lacked sufficient detail to facilitate easy review and did not include experiments that effectively demonstrated the performance enhancements provided by LLMPhy.

In response, we have thoroughly addressed all the reviewers’ suggestions to the best of our efforts and made substantial revisions to the paper. We kindly request the reviewers to reconsider our revised submission and provide feedback on whether the raised concerns have been adequately addressed. Below, we summarize the key changes made in this revision.

* We have reorganized the Experiments section with additional comparisons to baselines (CMA-ES), comparisons to other LLMs (Table 4), reporting performances from each Phases (Tables 1 and 2), comparisons between the estimated and ground truth physics parameters (Figure 5c), as well as comparing the optimization convergence (Figure 5-a,b), all the results demonstrating better results than prior works.

* We have also included additional illustrations in Figure 3 depicting our LLMPhy architecture and program synthesis examples demonstrating our APIs.

* We have revised the text in the paper to fix typos, including related prior works pointed out by the reviewers, as well as clarifying details of our task setup.

* In addition to these changes, we have also included extensive empirical results in the Appendix.

- * Qualitative comparisons between methods (Fig 4, Figure 13)
- * Sensitivity of the physics parameters to the simulation outcome (Fig 6)
- * Ablation studies to various choices of the LLMs in Phase 1 (Table 4)
- * Ablation studies using additional object classes with unique physical properties (Table 3)
- * Ablation studies with additional iterations for baseline optimizers (Table 3)
- * Robustness of the approach to noise in the feedback to LLM (Table 5)
- * Ablation studies using ground truth problem layout for Phase 2 (Table 1, Table 5)
- * Ablation studies on LLM feedback (Table 5)
- * Error and variance plots on the convergence of LLMPhy and other methods (Figure 10 - a, b, c, d)
- * Plots of the trajectories of parameter adjustments (Figure 10 e)
- * Qualitative visualizations of the optimization phases (Figures 11 and 12)
- * Supplementary video showing examples from our TraySim dataset and qualitative video results comparing the simulation using the estimated physical parameters against respective ground truth (*Supplementary materials/LLMPhy_comparison_videos_TraySim.mp4*)
- * We have included detailed explanation of the prompt and code examples to the LLM (Appendix/Section D)
- * We have included LLM-Simulator interaction and synthesized programs in Appendix/Section I, J, K, L.

We hope the reviewers can take a look at our revision and let us know if there are more concerns that we should consider including towards another revision.

---

### Author Response · Authors · 2024-11-27
**Rebuttal & Revision comments?**

Dear Reviewers,

We sincerely appreciate you taking the time to read our paper and providing very valuable feedback. We have carefully addressed your concerns in our detailed responses and have thoroughly revised the paper to incorporate the changes you suggested. We would be deeply grateful if you could kindly review our rebuttal and the updated paper and let us know if there are any additional questions or concerns we can address.


Thank you for your time and consideration

Authors

---

### Author Response · Authors · 2024-12-03
**Summary of Discussion**

Dear Reviewers,

We sincerely appreciate the time and effort you have invested in reviewing our paper and its revision, as well as the thorough discussions that ensued. We are delighted to see that several reviewers recognize the novelty of our contributions and the significance of our problem setup, as reflected in the increased scores. We are truly grateful for this positive feedback.

Below, we summarize the key points from the discussions:

**Reviewer 4AdL** acknowledges our challenging setup and the importance of our method for future robotic embodied capabilities. We are thankful for the comprehensive list of missing experiments provided by the reviewer, which we have thoroughly incorporated into the revision.

**Reviewer e7Sm** identified several textual issues, which we have corrected in the revision. The reviewer also requested experiments using an additional simulator. Due to the significant effort required, these experiments could not be completed within the rebuttal window. Instead, we provided indirect experiments by adding noise to the simulations and introducing new objects into the simulation with different physical attributes. Our method continues to perform well under these new conditions, substantiating its generalizability and robustness.

**Reviewer sHax** highlighted concerns regarding limited parameter size, lack of real-world experiments, data size, and the use of a perfect simulator. In response, we have conducted additional experiments with more objects in the scene thereby increasing the number of optimization variables, and provided experiments indirectly addressing robustness to noise and scalability of our method as mentioned above. Including real-world experiments would require additional efforts beyond the scope of the current work and could introduce factors outside the closed system we assume.

**Reviewer 2zB8** pointed out missing references and comparisons to prior works, which we have incorporated into our revision. We note that the suggested prior datasets are incompatible with our framework and thus cannot be compared. During discussions, the reviewer questioned the strength of our assumptions regarding the simulator and CAD models, and whether our setup qualifies as zero-shot. We have clarified our setup and reaffirm that it meets the standard definitions of zero-shot. Additionally, the reviewer noted slow inference times, which we acknowledge; however is caused by extraneous factors unrelated to our algorithm.

**Reviewer THXz** requested clarifications regarding prior works such as Eureka, comparisons to black-box methods like CMA-ES, explanations of the Phase 2 task, and the warm start of the method using prior solutions. We have revised the paper to address all the concerns by adding clarifications to Eureka and including an experimental comparison, providing performance comparisons to CMA-ES across multiple aspects, and enhancing the explanations of the Phase 2 task, while addressing the other concerns pointed out. The reviewer has increased their score but requested further clarifications on the differences to Eureka, which we have provided in our final response.


Overall, after extensive discussions with the reviewers, we believe we have clearly established the novelty of our technical methodology and our physical reasoning setup compared to prior works, as acknowledged by Reviewers 4AdL and e7Sm. The following outstanding concerns remain:
* Real-world experiments
* Use of alternative simulators to demonstrate generalizability
* Comparison to prior physical reasoning datasets and methods

We have made every feasible effort to address these outstanding issues and believe that the revision provides sufficient evidence demonstrating the robustness, adaptability, scalability, and scientific merit of our method. We earnestly hope that the reviewers will recognize the strengths of our work and support its acceptance.

Thank you,

Authors of LLMPhy

---

### Meta-Review · Area_Chair_H37Y · 2024-12-20

**Metareview:**

This paper introduces LLMPhy, which integrates LLMs with a physics simulator to address physics reasoning tasks. The main claim is that LLMPhy can iteratively estimate the physical hyperparameters of a system using an analysis-by-synthesis approach and leverage the inferred properties to solve reasoning tasks.

Strengths:

+ Dataset Contribution: TraySim provides a new benchmark for evaluating multi-object physical interactions.
+ Potential in Advancing Physical Reasoning: The paper demonstrates the promise of using LLMs for physical reasoning tasks.

Weaknesses
- Limited Experimental Scope: Multiple reviewers note that the experiments are limited in scope. Tests are restricted to a few object categories (initially three, later expanded to five). Additionally, the scenarios are simplistic, as they involve only single external forces applied to objects on a tray.
- Real-World Validation: A significant limitation of the paper is the lack of real-world testing. Reviewers also question the real-world applicability of the proposed method and whether it can translate effectively outside the simulation environment.

Overall:
The novelty of LLMPhy and its potential for advancing physical reasoning are notable strengths. However, several critical issues prevent its acceptance in its current form. The experimental scope is too narrow, and additional testing with diverse setups, simulators, and real-world validations is crucial to substantiate its claims.

**Additional Comments On Reviewer Discussion:**

The reviewers asked several clarifying questions and noted issues with the writing. These concerns were adequately addressed by the authors and acknowledged by the reviewers.

Multiple reviewers highlighted that the experimental evaluation was narrow, with only three object categories tested initially, simplistic scenarios, and limited configurations. The authors responded by expanding the number of object categories to five and clarifying that additional object configurations were tested during the rebuttal. While this expansion is an improvement, the fundamental issues of simplistic scenarios and a lack of diverse experimental setups remain partially unresolved.

Several reviewers also pointed out the absence of real-world experiments and questioned whether the proposed method's findings would translate effectively to real-world physics. The authors acknowledged this limitation, explaining that real-world validation requires significant additional effort, such as programming a robotic controller, which they plan to address in future work. While their plans for future work are noted, the lack of current results weakens the paper’s claims of applicability.

---

### Decision · Program_Chairs · 2025-01-22

Reject